# Unlocking the Potential of Weighting Methods in Federated Learning Through Communication Compression

**Valery Parfenov**[1,2,3] *    **Nail Bashirov**[1,2,4]    **Daniil Medyakov**[1,2,4]    **Dmitry Bylinkin**[1,2,4]

**Aleksandr Beznosikov**[1,2,4,5]

[1]Basic Research of Artificial Intelligence Laboratory (BRAIn Lab)
[2]Moscow Independent Research Institute of Artificial Intelligence
[3]HSE University    [4]Federated Learning Problems Laboratory    [5]Innopolis University

DEBUG: final mode ON

## Abstract

Modern machine learning problems are frequently formulated in federated learning domain and incorporate inherently heterogeneous data. Weighting methods operate efficiently in terms of iteration complexity and represent a common direction in this setting. At the same time, they do not address directly one of the main obstacle in federated and distributed learning – communication bottleneck. We tackle this issue by incorporating compression into the weighting scheme. We establish the convergence considering both exact and stochastic oracles. Finally, we evaluate the practical performance of the proposed method on classification problems.

## 1 Introduction

Behind groundbreaking results achieved by new machine learning models lies a carefully constructed optimization process. From the advent of `Stochastic Gradient Descent (SGD)` (Robbins & Monro, 1951) to adaptive methods like `Adam` (Kingma & Ba, 2014) and beyond, new outputs of optimization theory not only accelerated convergence but have, at times, redefined what is possible in entire industries. Contemporary supervised machine learning approaches universally require large-scale training data to reach state-of-the-art results on established benchmarks (Alzubaidi et al., 2021; Hoffmann et al., 2022; Shoeybi et al., 2019). The primary way to process this volume of samples is usage of multiple nodes for computations. This setting poses new challenges for the research community, highlighting once again that the future of the entire field hinges on novel solutions.

To harness the full potential of such data, distributed learning (Verbraeken et al., 2020) has become a domain paradigm, enabling cutting-edge results in computer vision (CV) (Goyal et al., 2017), natural language processing (NLP) (Shoeybi et al., 2019), and recommendation systems (Covington et al., 2016) by leveraging multiple machines working in parallel. Formally, this setting can be characterized by the following formulation of an optimization problem:

$$\min_{\theta \in \mathbb{R}^d} \left\{ f(\theta) = \frac{1}{M} \sum_{i=1}^{M} f_i(\theta) \right\}, \tag{1}$$

where $f_i(\theta)$ represents the empirical risk (Shalev-Shwartz et al., 2010) for data at node $i$. A bottleneck emerges in this distributed setting: communication. During the training process, local model states should be synchronized. This coordination steps can be prohibitively time-expensive and completely offset advantage gained from a parallel processing.

Federated learning offers several major approaches to address this issue: *local steps* techniques (Stich, 2018; Gorbunov et al., 2021b), *partial participation* concept (Li et al., 2019b; Rizk et al., 2021), *data-similarity-based* methods (Hendrikx et al., 2020; Kovalev et al., 2022; Lin et al., 2023). Finally, in our work, we adopt *compression*. The first works in this field were dedicated to one-bit

---

*Email: parfenov.vr@gmail.com

quantization (Seide et al., 2014; Bernstein et al., 2018). Currently, the most widely used techniques include quantization (Alistarh et al., 2017) and sparsification (Alistarh et al., 2018; Beznosikov et al., 2023a) methods such as `RandK` and `TopK`. A key consideration in this context is that increasing the number of nodes enhances the robustness of the training process to inaccuracies in aggregated local gradients. This gives rise to a trade-off between transmission precision and communication cost, which can be exploited by compressing gradients during aggregation. Formally, compression can be described using unbiased and contractive compression operators. In our work, we utilize the former.

**Definition 1.** *We say that a map $\mathcal{Q} : \mathbb{R}^d \to \mathbb{R}^d$ is an unbiased compression operator, or simply unbiased compressor, if there exist a constant $\omega$ such that holds:*

$$\mathbb{E}\left[\mathcal{Q}(x)\right] = x, \ \mathbb{E}\left[\|\mathcal{Q}(x)\|^2\right] \leq \omega \|x\|^2 \text{ for all } x \in \mathbb{R}^d. \tag{2}$$

Contemporary problem formulations often additionally involve heterogeneity, which necessitates the development of federated learning techniques (Konečný et al., 2016; McMahan et al., 2017; Smith et al., 2017; Li et al., 2020; Kairouz et al., 2021). The high cost of transmitting raw samples often makes homogeneous redistribution infeasible. Moreover, in particular settings, observation redistribution is impractical or fundamentally disallowed (Nishio & Yonetani, 2019; Zhang et al., 2020; Diao et al., 2020; Mishchenko et al., 2023; Khirirat et al., 2023; Islamov et al., 2025).

Standard formulation (1) of the objective function treats all devices equally. However, since the data across nodes may inherently differ, the effectiveness of this formulation becomes questionable. To address this issue, various weighting strategies was proposed, alternating the optimization problem into

$$\min_{\theta \in \mathbb{R}^d} \left\{ \sum_{i=1}^{M} \pi_i f_i(\theta) \right\}, \tag{3}$$

where $\pi_i$ represent weights constrained to the simplex $\Delta^{M-1}$, provided by particular weighting method. The idea here is to assign big weights to clients with clean representative or even unique data, and small weights to ones with noisy inappropriate samples. If this is achieved by any means, performance of the model can be improved by effectively training it on higher-quality observations.

Currently, a wide range of weighting methods has been developed (McMahan et al., 2017; Nishio & Yonetani, 2019; Wang et al., 2020; Cao et al., 2020). Each technique offers its own advantages, such as adaptivity or the absence of extra information communication. *Agnostic* reformulation of optimization problem (Mohri et al., 2019; Namkoong & Duchi, 2016; Shalev-Shwartz & Ben-David, 2014; Hashimoto et al., 2018):

$$\min_{\theta \in \mathbb{R}^d} \max_{\pi \in \Lambda} \left\{ \sum_{i=1}^{M} \pi_i f_i(\theta) \right\}, \tag{4}$$

where $\Lambda$ is a convex subset of $\Delta^{M-1}$, combines both of these advantages. The weights are selected automatically during training, while the strategy only requires the server to have additional knowledge of the local losses. Communicating this information is inexpensive and does not exacerbate the communication bottleneck. Intuitively, the method operates as follows: if certain nodes possess unique observations, a brief training phase can lead to a rapid loss reduction on the remaining users' samples. This, in turn, assigns higher weights to the devices holding the unique data, thereby mitigating the problem of data imbalance and reducing model bias.

However, while mitigating the issues of data heterogeneity across nodes, weighting methods do not address another core challenge – the communication bottleneck – which makes them independently nonviable in real-world applications. To address this fundamental problem and unlock the practical potential of weighting methods we aim to investigate the following question.

*Is it possible to effectively combine weighting-based approaches with communication compression techniques?*

## 2 OUR CONTRIBUTION

- We answer the posed question affirmatively by introducing `ADI` (Algorithm 1). It incorporates compression (1) into agnostic weighting scheme (4). Moreover, operating in the saddle

point problem setting, the proposed method never requires the transmission of full gradients, which is further reinforce practical applicability.

- We establish theoretical guarantees under general assumptions for the weighting setup. Our analysis additionally includes practically relevant settings of stochastic local oracles and partial participation.

- We validate `ADI` performance on the classification problems, including a large scale real world task and an experimental study of the interplay between compression and weighting techniques.

## 3 RELATED WORKS

In this section, we intend to survey both classical results and recent developments in the fields of weighting methods, compression, and saddle point problems, with a particular focus on studies that integrate the latter two. These directions are most relevant to our work.

### 3.1 WEIGHTING METHODS

First approach in this field, `FedAvg` (McMahan et al., 2017), suggests to assign weights to clients regarding the size of dataset $m_i$: $\pi_i = \frac{m_i}{m}$, where $m = \sum_{i=1}^{M} m_i$. This approach enables weight determination prior to training initiation, which precludes the need for additional inter-node communication and mitigates associated bottleneck. However, it only addresses data imbalance in terms of quantity rather than quality. Subsequent approaches employ dynamic weight assignment. To estimate client importance, they leverage such information as cross-client weight distribution divergence (Wang et al., 2020), local-global gradient discrepancy (Cao et al., 2020; Nguyen et al., 2020), and local loss (Mohri et al., 2019; Cho et al., 2022). Alternative approaches leverage hardware-aware metrics, including node computation capacity and connection stability, to accelerate training. These methods minimize participation of edge devices with significantly slower compute or communication capabilities (Nishio & Yonetani, 2019; Li et al., 2022; Ribero et al., 2022).

Utilized in this paper technique (4) (Mohri et al., 2019; Namkoong & Duchi, 2016; Shalev-Shwartz & Ben-David, 2014; Hashimoto et al., 2018) offers the benefit of adaptivity while introducing minimal additional communication overhead, as it only requires transmitting local loss values – a single scalar per device. The communication cost of aggregating this exact information is incomparably lower than even that of compressed gradients. This feature is particularly crucial as we aim to address the communication bottleneck. Finally, as can be observed, problem (4) is a saddle-point problem not a classical minimization one. This introduces additional challenges to algorithm design and theoretical analysis.

### 3.2 METHODS FOR SADDLE POINT PROBLEMS

The `Gradient Descent` method can be generalized to `Descent-Ascent` algorithm (Loizou et al., 2021) for saddle point problems (SPP). However, this straightforward generalization may fail to converge even for relatively simple objective functions (Beznosikov et al., 2023b). A more robust alternative, the `Extragradient` method, was introduced in 1976 by Korpelevich and has since become a fundamental paradigm for solving saddle point problems. The original `Extragradient` algorithm requires two gradient evaluations per iteration, but there are modifications that reduce this to a single one, for instance, optimistic approach (Popov, 1980). It is worth noting that alternative techniques for solving SPP also exist (Tseng, 2000; Nesterov, 2007; Malitsky, 2015). At the same time, the research community continues to actively adapt `Extragradient` method to various settings (Nemirovski, 2004; Alacaoglu & Malitsky, 2022), including distributed learning with communication compression (Beznosikov et al., 2022).

### 3.3 COMPRESSION METHODS

`QSGD` (Alistarh et al., 2017) was one of the initial steps toward understanding compression techniques applied to classical minimization problems. It examined the incorporation of quantized communication into `SGD` (Robbins & Monro, 1951). Authors used restrictive assumptions that all nodes have identical functions, and the stochastic gradients have bounded second moment. These assumptions were relaxed in subsequent studies (Khirirat et al., 2018; Mishchenko et al., 2024). Additionally,

`QSGD` suffered from an irreducible term in the theoretical convergence bound, caused by the stochasticity of the compressor, even when full local gradients were computed. The next notable concept in this field was the error feedback technique. Initially introduced as a successful heuristic (Seide et al., 2014; Ström, 2015), later it obtained theoretical support in (Stich et al., 2018; Karimireddy et al., 2019) and enabled the analysis of biased compression. Then, a significant advancement followed with the idea of compressing the difference between successive local gradient estimators, instead of directly compressing the gradients. This concept was first introduced in `DIANA` (Mishchenko et al., 2024) and enabled vanishing irreducible compressor stochasticity term, improved theoretical guarantees and extension of the analysis to new settings. Later, in (Richtárik et al., 2021), it was shown that local state difference compression can be interpreted as a variant of the error feedback technique, which led to the development of the `EF21` algorithm. Subsequently, in `MARINA` (Gorbunov et al., 2021a) the `PAGE` (Li et al., 2021) variance reduction technique was utilized. Using biased local gradient estimators `MARINA` reached state-of-the-art convergence rates. Finally, the authors of `DASHA` (Tyurin & Richtárik, 2022) ultimately combined error feedback with the `EF21` mechanism and achieved optimal oracle complexity while preserving the state-of-the-art communication performance of `MARINA`. Moreover, they eliminated the need for periodic transmission of full gradients, which was required in `MARINA`.

Despite the fundamental importance of variational inequalities including saddle point problems, and their extensive study, methods for them which incorporate the compression remains largely unexplored. Only several algorithms operating in this setting was proposed. `MASHA` (Beznosikov et al., 2022), integrates operator compression with the `Extragradient` concept. An extension of this approach, `Optimistic MASHA` (Beznosikov & Gasnikov, 2022), incorporates the optimistic principle and, through the use of permutation compressor, leverages data similarity to strengthen theoretical guarantees. Finally, `Three Pillars` (Beznosikov et al., 2023c) combines compression, data similarity, and local steps, unifying all three concepts within a single framework and achieving optimal theoretical guarantees. However, despite these theoretical advantages, the practical applicability of `Three Pillars` remains limited. In particular, due to its strong reliance on data similarity across all devices. Moreover, a key practical drawback of all three methods lies in the requirement for periodic transmission of full operator values.

## 4 SETUP

The analysis in this work is conducted relying on further assumptions.

**Assumption 1.** *For all $i = 1, 2, \ldots, M$, let $f_i$ be $\tilde{L}_i$-Lipschitz, i.e., $|f_i(\theta_1) - f_i(\theta_2)| \leq \tilde{L}_i \|\theta_1 - \theta_2\|$ holds for all $\theta_1, \theta_2 \in \mathbb{R}^d$. We denote $\tilde{L} = \max_i \{\tilde{L}_i\}$.*

**Assumption 2.** *For all $i = 1, 2, \ldots, M$, let $f_i$ be $L_i$-smooth, i.e., $\|\nabla f_i(\theta_1) - \nabla f_i(\theta_2)\| \leq L_i \|\theta_1 - \theta_2\|$ holds for all $\theta_1, \theta_2 \in \mathbb{R}^d$. We denote $L = \max_i \{L_i\}$.*

**Assumption 3.** *For all $i = 1, 2, \ldots, M$, let $f_i$ be convex, i.e., $f_i(\theta_1) \geq f_i(\theta_2) + \langle \nabla f_i(\theta_2), \theta_1 - \theta_2 \rangle$ holds for all $\theta_1, \theta_2 \in \mathbb{R}^d$.*

## 5 ALGORITHM AND THEORETICAL ANALYSIS

### 5.1 DESCRIPTION OF THE ALGORITHM

Now we are ready to present our Algorithm 1 `ADI` (**A**gnostic **DIANA**). In suggested approach each iteration begins with the nodes computing weighted loss gradient $\tilde{f}_i$, compressing the difference with their local memory state, and sending the result to the server – Lines 5, 7 and 9, respectively. Additionally, Lines 5 and 7 represents the modifications required in stochastic local oracle and partial participation settings. The nodes then update their local states in Line 8.

All remaining operations are carried out on the server side. Firstly, it aggregate compressed local differences and local losses – Line 11. Then, gradients estimators $g$ and $p$ with respect to $\theta$ and $\pi$ are computed in Lines 12 and 14. After that, the update of $\theta$ is performed using the optimistic version $\hat{g}$ of oracle $g$ – Lines 16 and 13. At the same time, the weights $\pi_i$ are constrained to remain within a subset $\Lambda$ of the simplex. Hence, a `Mirror Descent` step in Line 17 is applied to update them. Where the Kullback–Leibler divergence is used as the Bregman divergence (Bregman, 1967). Additionally, we point out that due to the maximization over $\pi_i$ in agnostic objective formulation (4),

a negative sign precedes the inner product. Finally, server updates the local state $h$ and communicates $\theta$ and $\pi_i$ to each node.

Thus, Lines 16 and 17 – considering Lines 13 and 15 – correspond to a step of `Optimistic Extragradient` method (Yudin, 1983; Popov, 1980). While Lines 7, 8, 11, 12, 18 reflect the idea of difference compression introduced in `DIANA` (Mishchenko et al., 2024). This concept resembles the variance reduction technique (Johnson & Zhang, 2013) and similarly enables the elimination of the irreducible term, caused by the stochasticity, in the convergence analysis. These methods are driven by an intuitive idea: near the optimum of a smooth function, the full gradient tends to zero, while local gradients may remain relatively large.

At the same time, according to Definition 1, the distortion introduced by the compressor scales with the norm of its input. As a result, near the optima compression of local gradient introduce a significant noise and prevent the aggregated estimator from converging to zero. This leads to erratic oscillations near the solution $\theta^*$ in practice and to an irreducible variance term in theoretical analysis. In contrast, the difference between local gradients at nearby points is bounded due to the smoothness of each local objective. Hence, as the algorithm approaches the optimum and the update steps diminish, it becomes necessary to compress progressively finer differences. Consequently, the local estimators $h_i^k$ tend to the local gradients $\nabla f_i(\theta^*)$. It ensure that the aggregated estimator $h^k$ converges to zero. This property allows the method to converge to the optimum itself, rather than to a neighborhood of it.

Let us now justify the choice of `DIANA` as the compression foundation in our method. To this end we consider alternative candidates. Firstly, `DASHA` (Tyurin & Richtárik, 2022), which demonstrates state-of-the-art results in the classical minimization setting, offers theoretical guarantees under non-convex objective functions. At the same time, the theoretical analysis of SPP in such setting remains largely underdeveloped, making the exten-

---

**Algorithm 1** `ADI`

1: **Input:** Starting points $\theta^0 \in \mathbb{R}^d, \pi^0 \in \Lambda$, $\{h_i^0\}_{i=1}^M$, $h_i^0 \in \mathbb{R}^d$ and $h^0 = \sum_{i=1}^M h_i^0$, number of iterations $K$, number of nodes $M$, random variables $\eta_i^k \sim Bern(p)$.

2: **Parameters:** $\alpha, \beta, \gamma_\theta, \gamma_\pi > 0; p \in (0, 1]$.

3: **for** $k = 1, 2, 3, \ldots, K$ **do**

4:     **for all nodes** $i = 1, 2, \ldots, M$ **in parallel do**

5:        $\begin{bmatrix} \tilde{f}_i^k = \pi_i^k \nabla f_i(\theta^k) \text{ exact local gradient} \\ f_i^k = \pi_i^k \nabla f_{i,\xi_i}(\theta^k) \text{ stochastic oracle} \end{bmatrix}$

6:        $\Delta_i^k = \tilde{f}_i^k - h_i^k$

7:        $\begin{bmatrix} \hat{\Delta}_i^k = \mathcal{Q}(\Delta_i^k) \\ \hat{\Delta}_i^k = \frac{\eta_i^k}{p}\mathcal{Q}(\Delta_i^k) \text{ partial participation} \end{bmatrix}$

8:        $h_i^{k+1} = h_i^k + \beta\hat{\Delta}_i^k$

9:        send $\hat{\Delta}_i^k, f_i(\theta^k)$ to server

10:     **end for**

11:     $\hat{\Delta}^k = \sum_{i=1}^M \hat{\Delta}_i^k$

12:     $g^k = h^k + \hat{\Delta}^k$

13:     $\hat{g}^k = (1 + \alpha)g^k - \alpha g^{k-1}$

14:     $p^k = \left(f_i(\theta^k)\right)_{i=1}^M$

15:     $\hat{p}^k = (1 + \alpha)p^k - \alpha p^{k-1}$

16:     $\theta^{k+1} = \theta^k - \gamma_\theta\hat{g}^k$

17:     $\pi^{k+1} = \arg\min_{\pi \in \Lambda}\left\{-\gamma_\pi\langle\hat{p}^k, \pi\rangle + D_{KL}(\pi, \pi^k)\right\}$

18:     $h^{k+1} = h^k + \beta\hat{\Delta}^k$

19:     server send $\pi_i^{k+1}, \theta^{k+1}$ to $i^{th}$ node for all $i$

20: **end for**

---

sion of `DASHA`'s analysis to our scenario intricated. `MASHA` (Beznosikov et al., 2022), on the other hand, operates in the SPP setting and incorporates compression. However, it requires periodic communication of full gradients, which significantly limits its practical applicability. By establishing the analysis of `DIANA` within the SPP setup, we avoid such constraints while leveraging its compression strategy.

Another detail we want to highlight is simplex regularization. In various problem settings, it may be advantageous to impose additional constraints on the weights by restricting the feasible set to a subset $\Lambda$ of the simplex $\Delta^{M-1}$ (Mehta et al., 2024). Let us provide a reasoning, helpful for understanding which regularization can be suitable in our case. Considering optimization problem (4) with $\Lambda = \Delta^{M-1}$ at optimum point the weights take the form of $\pi_{i_0} = 1, \pi_j = 0$ for all $j \neq i_0$, where $i_0 = \arg\max_i f_i(\theta^*)$. At the same time, some clients may possess noisy samples. The model can not – and should not – learn patterns from such data. Even a single device with notable higher noise level can cause an obstacle to effective training. Particularly, since its data is less representative, it experiences a slower decrease in loss. As training progresses, this leads to the weight of that client growing close to one. Further training will only lead to overfitting the model to the noise present in the data of the given device. This potential issue can be mitigated by using $\Lambda = \Delta^{M-1} \cap Q_a^M$, where $Q_a^M = \left\{x \in \mathbb{R}^M \big| 0 \leq x_i \leq \frac{a}{M}\right\}$ and $a \in [1, M]$. The parameter $a$ controls the trade-off

between full flexibility in weight assignment and stronger averaging. Specifically, setting $a = 1$ recovers formulation (1), while $a = M$ imposes no additional constraints on the weights. We employ regularization of the specified form and additionally highlight its role in the theoretical sectrion.

Finally, let us follow all communications in the proposed algorithm. At each iteration, transmissions of $f_i(\theta^k)$, $\hat{\Delta}_i^k$ from the nodes to the server and $\pi_i^k$, $\theta^k$ in the opposite direction are required. As $f_i(\theta^k)$ and $\pi_i^k$ are scalar values, they do not pose major threat to communication efficiency. Then, both $\hat{\Delta}_i^k$ and $\theta^k$ have dimensionality $d$. The vector $\theta^k$ is transmitted from the server to nodes, which poses fewer challenges (Kairouz et al., 2021). In contrast, aggregation of $\hat{\Delta}_i$ on the server constitutes the main obstacle to communication efficiency. ADI address this issue since $\hat{\Delta}_i^k$ is compressed version of $\Delta_i^k$, which makes its transmission significantly cheaper than that of an general vector of dimension $d$. We are now ready to proceed to the theoretical analysis. In the following sections, we provide guaranties for the cases of exact local gradients, stochastic local oracles and partial participation. For a comparison of the ADI rates with those of prior compression methods, we refer to Appendix A.

## 5.2 Convergence guaranties in exact local gradient setting

We establish the convergence with respect to **Gap** function (Definition 2 in Section F). It is standard for convex-concave SPP setup criteria. To initiate the analysis, we introduce the notation $z = (\theta, \pi)^\top$ and $F(z^k) = (g^k, -p^k)^\top$. Descent Lemma 3 (Section F) imposes conditions on operator $F$ evaluation across iterations. Then Lemmas 1 and 2 (Section E) justify the transition to the **Gap**$(z)$ function and further analysis.

Finally, Theorem 1 represents our main theoretical result. We remind that the constants $L$ and $\tilde{L}$ were introduced in Assumptions 2 and 1, respectively.

**Theorem 1.** *Let Assumptions 1, 2, 3 hold and* $\alpha = 1$, $\beta = \frac{1}{\omega}$, $\gamma_\pi = \gamma_\theta = \gamma \leq \gamma_0 = \min\left\{\frac{1}{2\tilde{L}}\sqrt{\frac{1}{96\omega^3 + 14M\omega^2}}, \sqrt{\frac{1}{2}\frac{1}{4M\tilde{L}^2 + 576\frac{a\omega^3}{M}L^2 + 28\omega^2 L^2}}\right\}$, $\Lambda = \Delta^{M-1} \cap Q_a^M$, *where* $Q_a^M = \{x \in \mathbb{R}^M | 0 \leq x_i \leq \frac{a}{M}\}$. *Then, after $K$ iterations of Algorithm 1 with unbiased compressor 1 $\mathcal{Q}$ and exact local gradients solving problem* (4) *the following holds:*

$$\mathbb{E}\left[\mathbf{Gap}(\overline{z}_K)\right] \leq \frac{V}{2\gamma K},$$

*where*

$$
\begin{aligned}
V \;=\; & \mathbb{E}\Bigg[\max_{z \in \mathcal{D}}\Big\{4D_{KL}(\pi, \pi^1) + 2\|\theta^1 - \theta\|^2 \\
& + 2\gamma\langle F(z^1) - F(z^0), z - z^1\rangle\Big\} + 32\gamma^2\omega^2\sum_{k=0}^{1}\sum_{i=1}^{M}\left\|\tilde{f}_i^k - h_i^k\right\|^2 + 7\gamma^2\omega\sum_{k=0}^{1}\left\|\tilde{f}^k - h^k\right\|^2\Bigg]
\end{aligned}
$$

*and $\overline{z}_K = \frac{1}{K}\sum_{k=1}^{K} z^k$.*

This implies the following bounds on the number of communication rounds and the amount of information transmitted from the clients to the server.

**Corollary 1.** *In setting of Theorem 1 with $\gamma = \gamma_0$, Algorithm 1 with exact local gradients needs*

$$\mathcal{O}\left(\frac{1}{\varepsilon}\left[\tilde{L}\omega^{3/2} + \tilde{L}M^{1/2}\omega + L\left(\sqrt{\frac{a\omega^3}{M}} + \omega\right)\right]\right)$$

*iterations in order to reach $\varepsilon$-accuracy with respect to $\mathbb{E}\left[\mathbf{Gap}(\overline{z}_K)\right]$. Additionally, it requires*

$$\mathcal{O}\left(\frac{1}{\varepsilon}\left[\tilde{L}\omega^{1/2} + \tilde{L}M^{1/2} + L\left(\sqrt{\frac{a\omega}{M}} + 1\right)\right]\right)$$

*bits communicated from nodes to the server.*

The first term in both bounds in Corollary 1 originate from the recursion on $\|\pi_i^{k+1} - \pi_i^k\|_1^2$. If the weights are fixed, these terms vanish, and under condition $\omega \leq M$, compression leads to at least no increase in communication complexity. Returning to the analysis of the full result, we must acknowledge that weighting algorithms typically suffer from weak theoretical guarantees. For instance, theoretically FedAvg enjoys only sublinear convergence rate in the strongly convex setting

(Li et al., 2019b). In our setup, the weighting-induced terms deteriorate the theoretical guarantees monotonically with increasing compression rate. Finally, discussing the role of simplex regularization $\Lambda$ in the theoretical analysis, we note that it enables an acceleration by a factor of $\frac{a}{M}$ in square-root terms.

### 5.2.1 EXTENSION TO THE NON-CONVEX SETUP

In this section, we extend our analysis by relaxing the convexity Assumption 3 to Assumption 4. It is inspired by the *minty* assumption, traditionally associated with non-monotonicity (non-convexity) in the respective literature Dang & Lan (2015); Mertikopoulos et al. (2018); Kannan & Shanbhag (2019).

**Assumption 4.** *Let there exists a point $\theta^* \in \mathbb{R}^d$ such that:*

$$\left\langle \sum_{i=1}^{M} \pi_i \nabla f_i(\theta), \theta - \theta^* \right\rangle \geq \sum_{i=1}^{M} \pi_i f_i(\theta) - \sum_{i=1}^{M} \pi_i f_i(\theta^*), \text{ for all } \theta \in \mathbb{R}^d, \pi \in \Delta^{M-1}.$$

For a more detailed discussion of the setting and the proofs for this section, we refer to Appendix G.

Considered setting is special, since the objective function $\sum_{i=1}^{M} \pi_i f_i(\theta)$ is linear in the weights $\pi$ by construction. This is also reflected in the criterion presented in (5) below. Convergence with respect to the weights $\pi$ involves the same term as in the convex setting, whereas convergence with respect to the parameter $\theta$ is now expressed through the mean squared norm of the gradients.

$$W^K = \mathbb{E} \max_{\pi' \in \Lambda} \left[ \sum_{i=1}^{M} f_i(\theta^*)(\overline{\pi}_i^K - \pi_i') + \left\langle \sum_{i=1}^{M} \pi_i' \nabla f_i(\theta^*), \overline{\theta}^K - \theta^* \right\rangle \right]$$

$$+ \frac{1}{8\gamma K} \sum_{k=1}^{K} \mathbb{E} \|\pi^{k+1} - \pi^k\|^2 + \frac{\gamma}{32} \mathbb{E} \left\| \sum_{i=1}^{M} \overline{\pi_i \nabla f_i(\theta)}^K \right\|^2,$$

Where $\overline{\pi}^K = \sum_{k=1}^{K} \frac{1}{K} \pi^{k+1}$, $\overline{\theta}^K = \sum_{k=1}^{K} \frac{1}{K} \pi^{k+1}$ and $\overline{\pi_i \nabla f_i(\theta)}^K \sim U\left[\pi_i^1 \nabla f_i(\theta^1), \dots, \pi_i^K \nabla f_i(\theta^K)\right]$.

The final result of this section, Corollary 2, provides convergence rates with respect to $W^K$ under the relaxed assumptions. We remind that the constants $L$ and $\tilde{L}$ were introduced in Assumptions 2 and 1 respectively.

**Corollary 2.** *In setting of Theorem 3 with $\gamma = \gamma_1$, Algorithm 1 with exact local gradients needs*

$$\mathcal{O}\left( \frac{1}{\varepsilon} \left[ \tilde{L}\omega^{3/2} + \tilde{L}M^{1/2}\omega + L\left( \sqrt{\frac{a\omega^3}{M}} + \omega \right) \right] \right)$$

*iterations in order to reach $\varepsilon$-accuracy with respect to $W^K$. Additionally, it requires*

$$\mathcal{O}\left( \frac{1}{\varepsilon} \left[ \tilde{L}\omega^{1/2} + \tilde{L}M^{1/2} + L\left( \sqrt{\frac{a\omega}{M}} + 1 \right) \right] \right)$$

*bits communicated from nodes to the server.*

### 5.3 CONVERGENCE GUARANTIES IN STOCHASTIC LOCAL ORACLE SETTING

Despite introducing new challenges, federated learning is still subject to classical difficulties of gradient-based optimization. In practice, computing the full even local gradient may be prohibitively expensive, especially in the presence of devices with limited computational capabilities. This makes stochastic optimization (Robbins & Monro, 1951; Bottou et al., 2018) particularly relevant in practical applications, including the context of federated learning. We extend our analysis to cover this setting as well.

Guaranties in this case are provided under assumption that all nodes have access to an unbiased oracle $\nabla f_{i,\xi_i}(x^k)$ with bounded variance, i.e., Assumption 5 holds.

**Assumption 5.** *Let for all $k = 1, 2, \dots, K$ and $i = 1, 2, \dots, M$ $\nabla f_{i,\xi_i}(\theta^k)$ satisfies*

$$i) \quad \mathbb{E}\nabla f_{i,\xi_i}(\theta^k) = \nabla f_i(\theta^k)$$
$$ii) \quad \mathbb{E}\|\nabla f_{i,\xi_i}(\theta^k) - \nabla f_i(\theta^k)\|^2 \leq \sigma^2.$$

`ADI` structure imposes minor modification in this setting. Particularly, Line 5 transforms into *sample $\tilde{f}_i^k = \pi_i^k \nabla f_{i,\xi_i}(\theta^k)$*. The theoretical analysis similarly remains largely unchanged, as stochasticity was already involved in the compression operator, and the oracle is assumed to be independent of it. Thus, we can reformulate Theorem 1 for stochastic oracle setting as follows.

**Theorem 2.** *Let in setting of Theorem 1 additionally Assumption 5 holds. Then, it implies*

$$\mathbb{E}\left[\mathbf{Gap}(\overline{z}_K)\right] \leq \frac{V}{2\gamma K} + \gamma \frac{64a^2\omega^2}{M}\sigma^2$$

*for iterations of Algorithm 1 with stochastic local oracles.*

Choosing $\gamma = \min\left\{\gamma_0, \sqrt{\frac{VM}{128a^2\omega^2\sigma^2 K}}\right\}$ we obtain the further guarantees.

**Corollary 3.** *In setting of Theorem 2 with $\gamma = \min\left\{\gamma_0, \sqrt{\frac{VM}{128a^2\omega^2\sigma^2 K}}\right\}$, Algorithm 1 with stochastic local oracles needs*

$$\mathcal{O}\left(\frac{1}{\varepsilon^2}\left[\frac{a^2\omega^2\sigma^2}{M}\right] + \frac{1}{\varepsilon}\left[\tilde{L}\omega^{3/2} + \tilde{L}M^{1/2}\omega + L\left(\sqrt{\frac{a\omega^3}{M}} + \omega\right)\right]\right)$$

*iterations in order to reach $\varepsilon$-accuracy with respect to $\mathbb{E}\left[\mathbf{Gap}(\overline{z}_K)\right]$. Additionally, it requires*

$$\mathcal{O}\left(\frac{1}{\varepsilon^2}\left[\frac{a^2\omega\sigma^2}{M}\right] + \frac{1}{\varepsilon}\left[\tilde{L}\omega^{1/2} + \tilde{L}M^{1/2} + L\left(\sqrt{\frac{a\omega}{M}} + 1\right)\right]\right)$$

*bits communicated from nodes to the server.*

In this case, guaranties in Theorem 2 are affected by an additional irreducible term $\gamma\frac{64a^2\omega^2}{M}\sigma^2$ induced by the stochasticity of the local oracle. It is general term for analysis in stochastic oracle setup with Assumption 5. In its presence, optimal stepsize $\gamma$ transforms into $\gamma = \min\left\{\gamma_0, \sqrt{\frac{VM}{128a^2\omega^2\sigma^2 K}}\right\}$ and communication complexity bounds include an additional term $\frac{1}{\varepsilon^2}\left[\frac{\omega\sigma^2}{M}\right]$.

## 5.4 Convergence Guaranties in Partial Participation Setting

Another classical direction in federated learning is partial participation (Li et al., 2019b; Rizk et al., 2021). In its context only the subset of all nodes are involved in each computation and communication round. This modification addresses several challenges inherent to federated setting, primarily the periodic unavailability of some devices (Li et al., 2019b; Yang et al., 2021). We establish theoretical guarantees for this setup as well.

**Corollary 4.** *In setting of Theorem 1 with $\beta = \frac{p}{\omega}$, $H = 32\gamma^2\left(\frac{\omega}{p}\right)^2$, $N = 7\gamma^2\frac{\omega}{p}$, $\gamma_\pi = \gamma_\theta = \gamma \leq \gamma_p = \min\left\{\frac{1}{2\tilde{L}}\sqrt{\frac{1}{96\left(\frac{\omega}{p}\right)^3 + 14M\left(\frac{\omega}{p}\right)^2}}, \sqrt{\frac{1}{2}\frac{1}{4M\tilde{L}^2 + 576\frac{a}{M}\left(\frac{\omega}{p}\right)^3 L^2 + 28\left(\frac{\omega}{p}\right)^2 L^2}}\right\}$ it implies*

$$\mathbb{E}\left[\mathbf{Gap}(\overline{z}_K)\right] \leq \frac{V}{2\gamma K}$$

*for iterations of Algorithm 1 with partial participation.*

According Corollary 4, we bound number of communication rounds and the volume of data sent from the clients to the server.

**Corollary 5.** *In setting of Corollary 4 with $\gamma = \gamma_p$, Algorithm 1 with partial participation needs*

$$\mathcal{O}\left(\frac{1}{\varepsilon}\left[\tilde{L}\left(\frac{\omega}{p}\right)^{3/2} + \tilde{L}M^{1/2}\frac{\omega}{p} + L\left(\sqrt{\frac{a\omega^3}{Mp^3}} + \frac{\omega}{p}\right)\right]\right)$$

*iterations in order to reach $\varepsilon$-accuracy with respect to $\mathbb{E}\left[\mathbf{Gap}(\overline{z}_K)\right]$. Additionally, it requires*

$$\mathcal{O}\left(\frac{1}{\varepsilon}\left[\tilde{L}\left(\frac{\omega}{p}\right)^{1/2} + \tilde{L}M^{1/2} + L\left(\sqrt{\frac{a\omega}{Mp}} + 1\right)\right]\right)$$

*bits communicated from nodes to the server.*

Analysis in this setting relies on the observation that multiplying the compression operator by the $\frac{\eta}{p}$, with $\eta \sim \mathrm{Bern}(p)$, yields another valid compression operator. It remains unbiased, while its compression rate $\omega$ is scaled by a factor of $p$. Finally we note that our analysis in stochastic local gradients and partial participation settings can be straightforwardly merged.

## 6 EXPERIMENTS

To validate the performance of our algorithm `ADI` on practical tasks, we compare it in experiments against baseline methods that employ either weighting schemes or communication compression techniques. Specifically, `ADI` with no compression and `EF21`(Richtárik et al., 2021), `DIANA` (Mishchenko et al., 2024) serve as representatives respectively. Although weighting-based approaches are specifically designed to improve performance in heterogeneous settings, we assess the generality of `ADI` by conducting experiments under varying degrees of heterogeneity, including the homogeneous case. It is also important to note that classical approaches and weighting-based methods formally solve different optimization problems (1) and (3). Consequently, comparing them in terms of loss is not valid, and we instead rely on model quality metrics such as accuracy.

We conduct a comparative evaluation on image classification tasks using CIFAR-10 (Krizhevsky et al., 2009) dataset and RESNET-18 (Meng et al., 2019) neural network architecture, which is considered to be a standard benchmark for optimizers performance. We set number of clients $M$ equal to 10 and evaluate optimizers under 2 major data distribution setups: **i.i.d.** distribution, where each client has the same number of data samples, and class labels are uniformly distributed across clients; and **non-i.i.d.** distribution (namely Dirichlet one with the parameter $\alpha = 0.5$) with different amount of data samples per client.

The first set of experiments, presented in Figure 1, compares `ADI` with compression-based methods under different setups of data heterogeneity and parameter $K = 10\%, 50\%$ for

(a) Rand50% compressor.

(b) Rand10% compressor.

Figure 1: Performance comparison for `ADI` across different heterogeneity levels.

`RandK` compressor. To ensure a fair comparison, we run the experiments for 10k communication rounds with stochastic oracle for each method and tune theirs hyperparameters.

As illustrated in the plots presented in Figures 1a, 1b, the weighting mechanism plays a crucial role in the convergence behavior of our method. By effectively mitigating the impact of data heterogeneity, `ADI` demonstrates superior convergence properties compared to baseline approaches. Furthermore, the accumulated weight adjustments significantly influence the later stages of training, contributing to enhanced model accuracy and overall performance. Notably, comparing `ADI` and `ADI (no comp.)`, we observe that incorporating compression only marginally slows down the convergence of the algorithm in terms of iterations, while yielding significant savings in communication bits.

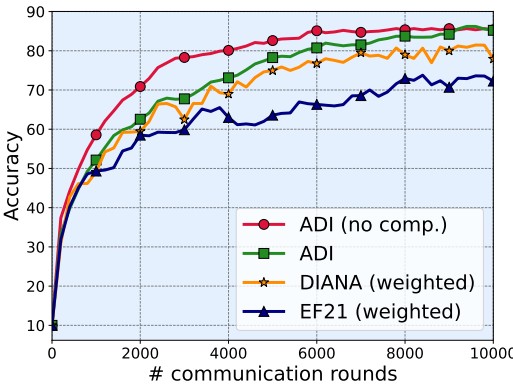

Figure 2: Weighting approach comparison.

With the identity compressor, `ADI` reduces to an `Optimistic Extragradient` (Popov, 1980) method for problem (4), effectively representing a standalone weighting-based optimization approach.

**Ablation study.**   The second experiment (see Figure 2) compares the same compressed baselines, where we additionally incorporate agnostic weighting via a heuristic descent-ascent-style modification, in order to isolate and assess the specific role of our joint compression-weighting design. We use `RandK` with `K` $= 10\%$ and **non-i.i.d.** data distribution. We observe consistent improvements in convergence across all methods – demonstrating that while weighting itself enhances robustness in highly heterogeneous settings, our principled joint design yields the strongest gains over these naive integrations. We further analyze the evolution of client weights under Algorithm 1 in a heterogeneous setting, with full results shown in Figure 3. At initialization, all clients are assigned equal weights, reflecting no prior knowledge of their data quality or relevance. As training progresses, the weights rapidly diverge, adapting to the statistical heterogeneity of local datasets. Over time, each client's weight converges to a distinct, stable plateau – indicating that the system learns a consistent, data-driven importance score for every participant.

This convergence behavior reveals two key phases of the optimization process:

(i)   an early exploration phase, during which substantial weight adjustments occur;

(ii)   a later stabilization phase, where weights remain nearly constant once the global model approaches an optimum.

Notably, significant reweighting ceases once the optimizer enters a neighborhood of a (local or global) minimum, suggesting that the weighting mechanism primarily acts during transient, high-gradient stages of training — precisely when client contributions are most discriminative.

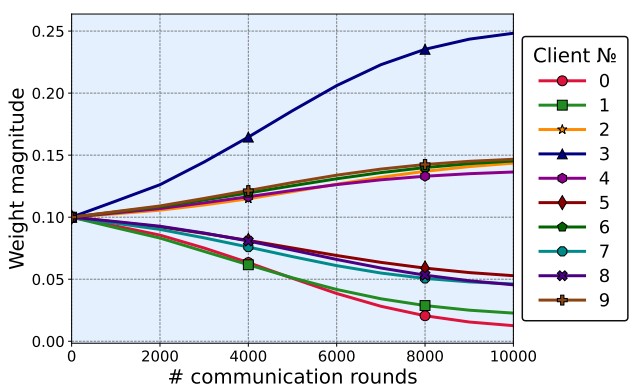

Figure 3: Weights magnitudes for Algorithm 1 in **non-i.i.d.** data distribution setup.

Experimental setup details, additional experiments, including large scale problems, and more detailed weighting-compression interaction experimental study can be found in the Appendix B and C.

## 7   DISCUSSION

This study has introduced a method for federated learning, supported by comprehensive theoretical analysis and empirical validation. Theoretical guarantees were established for a range of relevant scenarios, including setups with exact local gradients, stochastic local oracles, and partial client participation. Experimental results demonstrated that the superiority of the proposed method over the baselines becomes more pronounced as the level of compression and data heterogeneity increases. This allows us conclude that two of the most important problems in federated learning – the communication bottleneck and data heterogeneity – can be addressed concurrently, offering new potential for specific federated learning formulations. Additionally, the developed approach maintains performance comparable to baseline algorithms in homogeneous data settings and never requires the transmission of full gradients, thus further supporting its practical utility.

As a natural direction for future research, we consider the adaptation of accelerated methods for saddle-point problems to our setting, as well as the extension of our analysis to the case of contractive compression operators (Beznosikov et al., 2023a).

## ACKNOWLEDGMENTS

The work was supported by the Ministry of Economic Development of the Russian Federation (agreement No. 139-15-2025-013, dated June 20, 2025, IGK 000000C313925P4B0002).

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

## A  ADDITIONAL COMPARISON WITH PRIOR WORKS

In this section we provide complexity comparison of `ADI` against baselines.

Note that while previous compression algorithms address Problem (1), `ADI` operates with the objective function (4). The difference in objective functions makes a direct formal comparison of the rates less transparent. Additional difficulties in theoretically comparing `ADI` with classical compression methods arise from the distinct convergence criterion inherent to saddle-point problems, which should be taken into consideration as well. Nevertheless, for completeness of presentation, we provide the comparative Table 1 below. Comparison is conducted for a smooth nonconvex setup, which in case of `ADI` and `MASHA`, corresponds to the minty assumption. For the sake of clarity, constants specific to the problem have been omitted from the estimates.

Table 1: Comparison of complexity across algorithms with compression.

| Algorithm | Communication rounds | Bits of communication |
|---|---|---|
| ADI (this work) | $\frac{1}{\varepsilon}\left[\omega^{3/2} + M^{1/2}\omega + \sqrt{\omega^3/M}\right]$ | $\frac{1}{\varepsilon}\left[\omega^{1/2} + M^{1/2} + \sqrt{\omega/M}\right]$ |
| DIANA (Horvóth et al., 2022) | $\frac{1}{\varepsilon}\left[1 + (1+\omega)\sqrt{\omega/M}\right]$ | $\frac{1}{\varepsilon}\left[1/\omega + \frac{1+\omega}{\sqrt{\omega M}}\right]$ |
| DASHA (Tyurin & Richtárik, 2022) | $\frac{1}{\varepsilon}[1 + \omega/\sqrt{M}]$ | $\frac{1}{\varepsilon}[1/\omega + 1/\sqrt{M}]$ |
| EF21 (Richtárik et al., 2021) | $\frac{1}{\varepsilon\alpha}$ | $\frac{1}{\varepsilon}$ |
| MASHA (Beznosikov et al., 2022) | $\frac{1}{\varepsilon}\left[\omega^2/M + \omega\right]$ | $\frac{1}{\varepsilon}\left[\omega/M + 1\right]$ |

*Notation:* $\varepsilon$ = accuracy of the solution, $\omega$ = compression rate introduced in Definition 1, $M$ = number of nodes, $\alpha$ = parameter of contractive compressor.

Additionally, we note that `EF21` was originally designed to be used with a contractive compressor with constant $\alpha$. Since this is a well-known and practically significant method, we include it in our experimental baselines and present its convergence rate for completeness.

## B  ADDITIONAL CLARIFICATION ON IMAGE CLASSIFICATION

Our experiments are conducted on the CIFAR-10 (Krizhevsky et al., 2009) dataset using a RESNET-18 (Meng et al., 2019) architecture, with $M = 10$ clients for federated training. We evaluate each sampling strategy under three representative data partitioning schemes: (**homo**) an *i.i.d.* homogeneous split, where each client receives a statistically identical sample of the data; (**hetero**) a heterogeneous configuration in which clients are assigned disjoint class subsets, simulating non-*i.i.d.* label distributions; and (**pathological**) a strongly heterogeneous regime, reflecting real-world imbalances through uneven data quantities and skewed class distributions across clients. This controlled setup enables a rigorous comparison of Algorithm 1 under increasingly realistic and challenging federated learning conditions. In all experiments we do not use simplex regularization, i.e. $\Lambda = \Delta^{M-1}$.

### B.1  HYPERPARAMETERS DETAILS

In our experiments, we employed the default partitioning utility provided by the `Flower` (`flwr`) framework (Beutel et al., 2020) to generate a non-i.i.d. (heterogeneous) data distribution across clients. To calibrate the hyperparameters of the `ADI` optimization method, we conducted a systematic grid search over the following ranges:

- Learning rate for model parameters ($\gamma_\theta$):
  $\{1\times10^{-4}, 5\times10^{-4}, 1\times10^{-3}, 5\times10^{-3}, 1\times10^{-2}, 2\times10^{-2}\}$
- Learning rate for client weights ($\gamma_\pi$):
  $\{1\times10^{-3}, 5\times10^{-3}, 1\times10^{-2}, 5\times10^{-2}, 1\times10^{-1}\}$

- Momentum decay coefficient for the model update ($\alpha$):
  $\{0.70, 0.75, 0.80, 0.85, 0.90, 0.95\}$

- Momentum decay coefficient for the weight adaptation ($\beta$):
  $\{0.05, 0.10, 0.15, 0.20\}$

The optimal configuration (selected based on validation performance (e.g., final accuracy and convergence stability)) was identified as:

$$\gamma_\theta = 0.01, \quad \gamma_\pi = 0.01, \quad \alpha = 0.90, \quad \beta = 0.10.$$

Furthermore, across all compared methods, we employed a staged learning rate decay schedule to promote convergence stability. Specifically, the initial learning rate was reduced by a factor of 5 after the 2000 communication rounds and subsequently by an additional factor of 10 (i.e., $50\times$ relative to the initial value) after the 7500th communication round. Formally, for an initial learning rate $\gamma_\theta$, the schedule is defined as:

$$\gamma_\theta(k) = \begin{cases} \gamma_\theta, & k < 2000, \\ \gamma_\theta/5, & 2000 \leq k < 7500, \\ \gamma_\theta/50, & k \geq 7500, \end{cases}$$

where $k$ denotes the round number.

## C  ADDITIONAL EXPERIMENTS

### C.1  LINEAR REGRESSION

To evaluate the performance of proposed method under tightly controlled conditions, we conduct additional experiments on the simplest task. We use the `diabets_scaled` (Chang & Lin, 2011) dataset for linear regression task consisting of 768 samples with 8 features and two classes. As baselines, we select the communication compression algorithm `DIANA` (Mishchenko et al., 2024); for uncompressed weighting method, we use `ADI` with identical compressor as `Optimistic Extragradient` (Popov, 1980) for formulation (4). Additionally, we compare `ADI` with `MASHA` (Beznosikov et al., 2022) for problem (4), which is an analogous method combining both weighting and compression. For all algorithms with compression we utilize `RandK` compressor.

To model different degrees of heterogeneity, we introduce parameter $\alpha_h \in [0, 1]$. While emulating training on $M = 4$ devices, we distribute data across clients as follows: the first node receives $\frac{1}{M} + \alpha_h \frac{M-1}{M}$ observations from the negative class and $\frac{1-\alpha_h}{M}$ positive observations. The remaining data is distributed uniformly across the other $M - 1$ devices. Thus, $\alpha_h = 1$ corresponds to complete heterogeneity where the negative class appears only on one device while the other devices contain exclusively positive class observations. Accordingly, $\alpha_h = 0$ corresponds to complete data homogeneity.

The first series of experiments (Figure 4) compares `ADI` with the specified baselines under different levels of data heterogeneity ($\alpha_h$ equal to 0, 0.5, and 1). For all compression methods, we use `RandK` with `K = 1`. `MASHA` additionally transmits full gradients every 8 iterations. These experiments confirm the superiority of weighting methods: while showing comparable performance on homogeneous data, `ADI` gains significant advantage over `DIANA` as heterogeneity increases. By comparing `Optimistic Extragradient` with other methods, we demonstrate the effectiveness of compression, particularly in combination with weighting approaches across varying heterogeneity levels. Finally, we present the evolution of `ADI` algorithm's weights across iterations. We observe that their dynamics can be unpredictable, particularly in the homogeneous setup. Yet this does not lead to performance degradation.

Experiments in Figure 5 compares `ADI` and `DIANA` at $\alpha_h = 0.5$ with different compressor constants `K` $(8, 5, 2, 1)$. This comparison highlights that the advantage of weighting remains independent of the compression level even under aggressive compression as `Rand1`.

Finally, in Figure 6 we verified that `ADI` weights ultimately stabilize in both complete homogeneity and heterogeneity cases. Notably, in the homogeneous scenario, their pre-stabilization evolution does not affect performance.

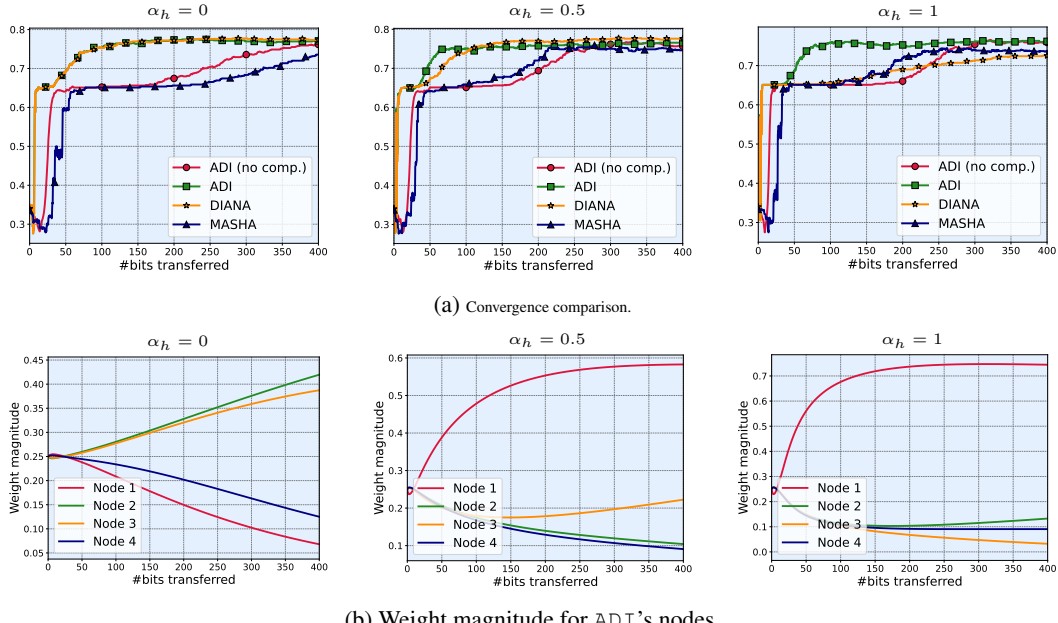

(a) Convergence comparison.

(b) Weight magnitude for `ADI`'s nodes.

Figure 4: Performance comparison for `ADI` across different heterogeneity levels.

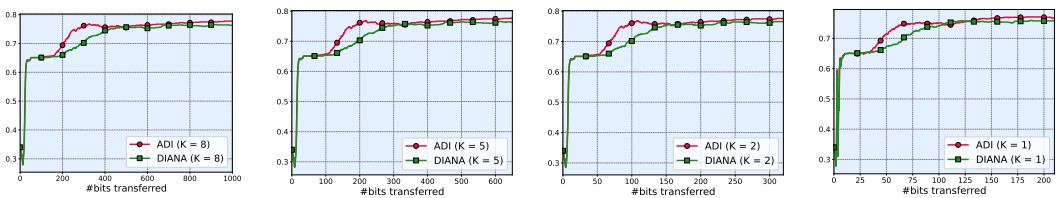

Figure 5: `ADI` and `DIANA` with `RandK` across different K with $\alpha_h = 0.5$.

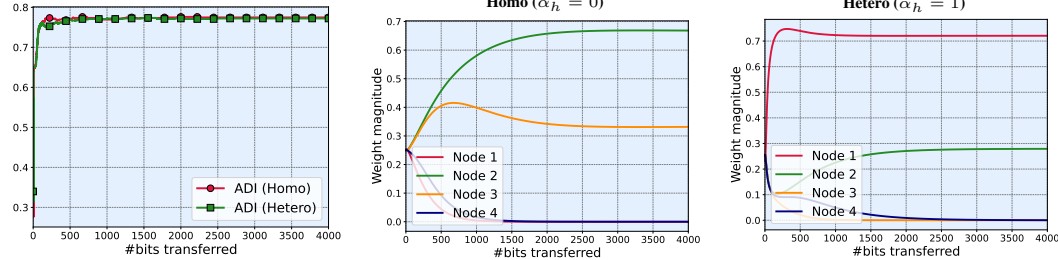

Figure 6: Weights stabilization in homogeneous and heterogeneous setups.

## C.2    ADDITIONAL DATA PARTITIONING

For greater completeness of the experimental evaluation, we conducted an additional experiment using an alternative heterogeneity modeling setup. Specifically, we evaluate the same RESNET-18 (Meng et al., 2019) backbone on the CIFAR-10 (Krizhevsky et al., 2009) dataset partitioned across 10 clients according to a Dirichlet distribution with parameter $\alpha = 0.3$, introducing a stronger degree of non-i.i.d. data heterogeneity. The corresponding results are presented in the Table 2 below. These results report the maximum accuracy (mean $\pm$ standard deviation over three independent runs) achieved by the methods under the same number of communication rounds and an equal compression level.

Table 2: Comparison of ADI with baselines under Dirichlet ($\alpha = 0.3$) partition

| Method | Rand 50% Acc. | Rand 10% Acc. | Rand 5% Acc. |
|--------|---------------|---------------|--------------|
| ADI | $82.6 \pm 0.24$ | $81.9 \pm 0.24$ | $82.4 \pm 0.28$ |
| DIANA | $76.7 \pm 0.26$ | $77.1 \pm 0.32$ | $75.4 \pm 0.40$ |
| EF21 | $64.5 \pm 0.43$ | $64.4 \pm 0.41$ | $61.6 \pm 0.43$ |
| MASHA1 | $63.7 \pm 0.20$ | $63.4 \pm 0.21$ | $63.5 \pm 0.21$ |

This experimental validation demonstrates that even under highly heterogeneous data distribution and a high power of compression (5%), our method achieves strong performance and delivers improved results compared to the baselines.

### C.3 WEIGHTS DISTRIBUTION ANALYSIS

To investigate the sensitivity of client-specific aggregation weights to the severity of model compression, we conduct an ablation study on RESNET-18 (Meng et al., 2019) as the backbone architecture and the CIFAR-10 (Krizhevsky et al., 2009) dataset, partitioned across 10 clients using a Dirichlet distribution with parameter $\alpha = 0.5$ to induce data heterogeneity.

Three compression levels - Rand 50%, Rand 10%, and Rand 5% - are evaluated. For each setting, we report the learned client aggregation weights (mean $\pm$ standard deviation over three independent runs) in Table 3 below.

Table 3: Final client weights assigned by ADI under different compression levels

| Client no. | Weight (Rand 50%) | Weight (Rand 10%) | Weight (Rand 5%) |
|------------|-------------------|-------------------|------------------|
| 1 | $0.019 \pm 0.005$ | $0.019 \pm 0.006$ | $0.014 \pm 0.005$ |
| 2 | $0.026 \pm 0.007$ | $0.025 \pm 0.007$ | $0.027 \pm 0.009$ |
| 3 | $0.141 \pm 0.007$ | $0.139 \pm 0.008$ | $0.139 \pm 0.007$ |
| 4 | $0.248 \pm 0.011$ | $0.244 \pm 0.012$ | $0.247 \pm 0.014$ |
| 5 | $0.136 \pm 0.007$ | $0.137 \pm 0.007$ | $0.134 \pm 0.007$ |
| 6 | $0.054 \pm 0.008$ | $0.055 \pm 0.007$ | $0.057 \pm 0.011$ |
| 7 | $0.145 \pm 0.011$ | $0.143 \pm 0.016$ | $0.144 \pm 0.013$ |
| 8 | $0.047 \pm 0.008$ | $0.062 \pm 0.013$ | $0.056 \pm 0.014$ |
| 9 | $0.043 \pm 0.009$ | $0.052 \pm 0.008$ | $0.048 \pm 0.011$ |
| 10 | $0.147 \pm 0.010$ | $0.124 \pm 0.013$ | $0.134 \pm 0.011$ |

Let us briefly describe and interpret obtained results.

(i) For the majority of clients (e.g., Clients 3, 4, 5, 7), the assigned aggregation weights remain remarkably stable across compression regimes, with variations typically within the margin of statistical uncertainty. This suggests convergence toward a data-informed equilibrium - consistent with theoretical expectations that optimal client weights reflect local data representativeness and utility, rather than being artifacts of compression-induced noise.

(ii) Notably, Clients 8 and 9 exhibit non-monotonic weight adjustments under aggressive compression with Rand 5%, diverging from their trends at 50% and 10% compression. Post-hoc data inspection reveals that these clients possess highly skewed local distributions: each holds samples from only three classes, with two dominant classes constituting over 93% of their local datasets. Under severe sparsification, the reduced model capacity likely amplifies the impact of such distributional bias, leading the weight adaptation mechanism to dynamically re-calibrate contribution levels.

These findings underscore that while global aggregation weights are generally robust to moderate compression, extreme compression intensifies sensitivity to local data pathology, highlighting the interplay between model compression, client heterogeneity, and adaptive weighting strategies in federated optimization.

## C.4 Large scale problem

To further validate the scalability and robustness of ADI, we conduct experiments on the CIFAR-100 (Krizhevsky et al., 2009) dataset, which presents a more challenging, large-scale classification task with 100 classes. We evaluate the RESNET-34 (Koonce, 2021) backbone across 10 clients, with data heterogeneity modeled using a Dirichlet distribution with parameter $\alpha = 0.5$, under three compression regimes: Rand 50%, Rand 10%, and Rand 5%.

In the table below, we report the final accuracy (mean $\pm$ standard deviation over three independent runs with $10^4$ communication rounds) for ADI and several prior approaches.

Table 4: Comparison of ADI with baselines on CIFAR-100 under different compression levels

| Method | Rand 50% Acc. | Rand 10% Acc. | Rand 5% Acc. |
|---|---|---|---|
| ADI | $71.2 \pm 0.21$ | $71.7 \pm 0.22$ | $70.9 \pm 0.26$ |
| DIANA | $69.8 \pm 0.19$ | $70.1 \pm 0.22$ | $71.1 \pm 0.27$ |
| EF21 | $62.2 \pm 0.41$ | $62.3 \pm 0.38$ | $59.2 \pm 0.43$ |
| MASHA1 | $61.7 \pm 0.22$ | $63.2 \pm 0.17$ | $62.5 \pm 0.21$ |

These results demonstrate that ADI maintains stable and competitive performance even on a large-scale, highly heterogeneous dataset, consistently outperforming prior approaches. The findings underscore the effectiveness of ADI's adaptive weighting mechanism in challenging, real-world federated learning scenarios.

## C.5 Additional weighting baselines and partial client participation

To assess the effectiveness of the selected compression strategy, we conduct additional experiments incorporating direct compression of transmitted gradients (Alistarh et al., 2017) as naive baseline into AFL (Mohri et al., 2019) and q-FFL (Li et al., 2019a) traditional weighting algorithms. To further investigate the effects associated with partial client participation, we extended the experimental setup to the corresponding setting. Table 5 presents comparison results on CIFAR-10 (Krizhevsky et al., 2009) using the RESNET-18 (Meng et al., 2019) architecture, while Table 6 reports results on CIFAR-100 (Krizhevsky et al., 2009) with the RESNET-34 (Koonce, 2021) architecture across varying compression rates. Data heterogeneity is induced via a Dirichlet distribution with parameter $\alpha = 0.5$ across 10 clients, while client availability is sampled from a Bernoulli distribution with parameters $p = 0.5, 0.7, 1.0$. For every setup 3 runs was conducted with $10^4$ communication rounds.

Table 5: Comparison of methods on CIFAR-10 under partial client participation setting

| Method ($p$) | Rand 50% Acc. | Rand 10% Acc. | Rand 5% Acc. |
|---|---|---|---|
| ADI ($p = 1.0$) | $85.2 \pm 0.21$ | $85.7 \pm 0.22$ | $84.9 \pm 0.26$ |
| AFL ($p = 1.0$) | $67.2 \pm 0.37$ | $52.3 \pm 0.38$ | $47.2 \pm 0.43$ |
| q-FFL ($p = 1.0$) | $68.7 \pm 0.34$ | $53.2 \pm 0.37$ | $46.5 \pm 0.41$ |
| ADI ($p = 0.7$) | $82.2 \pm 0.20$ | $81.6 \pm 0.22$ | $81.9 \pm 0.22$ |
| AFL ($p = 0.7$) | $65.7 \pm 0.31$ | $49.9 \pm 0.35$ | $44.2 \pm 0.40$ |
| q-FFL ($p = 0.7$) | $65.9 \pm 0.32$ | $50.2 \pm 0.37$ | $45.7 \pm 0.41$ |
| ADI ($p = 0.5$) | $79.6 \pm 0.22$ | $78.9 \pm 0.21$ | $79.3 \pm 0.22$ |
| AFL ($p = 0.5$) | $64.5 \pm 0.31$ | $48.1 \pm 0.39$ | $48.2 \pm 0.41$ |
| q-FFL ($p = 0.5$) | $65.7 \pm 0.29$ | $49.2 \pm 0.34$ | $45.5 \pm 0.41$ |

The table reveals several clear patterns. ADI is almost insensitive to the compression level, even under extreme compression of 5%. However, a decrease in the client availability probability $p$ leads to a slight deterioration in performance. The opposite trend is observed for the baselines: they are relatively insensitive to partial client participation, but their performance drops substantially as the compression level increases.

Overall, the results demonstrate a significant advantage of **ADI** over the baselines: in the most extreme setup ($p = 0.5$, Rand 5%), the algorithm maintains a substantial performance lead over the baselines even under their most favorable conditions ($p = 1$, Rand 50%). This highlights the

Table 6: Comparison of methods on CIFAR-100 under partial client participation setting

| Method ($p$) | Rand 50% Acc. | Rand 10% Acc. | Rand 5% Acc. |
|---|---|---|---|
| ADI ($p = 1.0$) | $71.2 \pm 0.21$ | $71.7 \pm 0.22$ | $70.9 \pm 0.26$ |
| AFL ($p = 1.0$) | $58.2 \pm 0.32$ | $46.3 \pm 0.31$ | $41.2 \pm 0.33$ |
| q-FFL ($p = 1.0$) | $59.7 \pm 0.31$ | $46.2 \pm 0.30$ | $41.5 \pm 0.31$ |
| ADI ($p = 0.7$) | $69.4 \pm 0.20$ | $68.9 \pm 0.20$ | $69.0 \pm 0.19$ |
| AFL ($p = 0.7$) | $58.2 \pm 0.31$ | $46.9 \pm 0.32$ | $41.4 \pm 0.33$ |
| q-FFL ($p = 0.7$) | $58.2 \pm 0.32$ | $46.2 \pm 0.31$ | $41.6 \pm 0.31$ |
| ADI ($p = 0.5$) | $66.9 \pm 0.20$ | $66.9 \pm 0.20$ | $66.2 \pm 0.21$ |
| AFL ($p = 0.5$) | $58.4 \pm 0.30$ | $45.7 \pm 0.28$ | $41.6 \pm 0.30$ |
| q-FFL ($p = 0.5$) | $59.5 \pm 0.29$ | $45.8 \pm 0.29$ | $41.3 \pm 0.31$ |

effectiveness of the chosen compression strategy for achieving a strong performance in real-world applications.

## D    GENERAL INEQUALITIES AND NOTATION

Suppose $x, y \in \mathbb{R}^d$, $\pi_1, \pi_2 \in \Delta$ and $D_{KL}$ is Kullback–Leibler divergence. Then, following inequality holds:

$$\langle x, y \rangle \leq \frac{\beta}{2} \|x\|^2 + \frac{1}{2\beta} \|y\|^2, \tag{Fen}$$

$$\|x + y\|^2 \leq (1 + \alpha) \|x\|^2 + \left(1 + \alpha^{-1}\right) \|y\|^2, \tag{CS}$$

$$D_{KL}(\pi_1, \pi_2) \geq \frac{1}{2} \|\pi_1 - \pi_2\|_1^2 \geq \frac{1}{2} \|\pi_1 - \pi_2\|^2. \tag{Pi}$$

**Definition 2.** *Let $F : \mathbb{R}^d \to \mathbb{R}^d$ and $\mathcal{D}$ be a compact subset of $\mathbb{R}^d$. Then, for any $z \in \mathbb{R}^d$ we define*

$$\mathbf{Gap}(z) = \max_{z' \in \mathcal{D}} \left\{ \langle F(z'), z - z' \rangle \right\}.$$

**Definition 3.** *Let $\|z\|_{bits}$ represents the amount of bits required to encode the vector $z \in \mathbb{R}^d$, $b$ denotes the number of bits per floating point value, and $d$ is the dimensionality of the problem (i.e., $bd = \|z\|_{bits}$). Then for compression operator $\mathcal{Q}$ we define the expected density of compressed vector*

$$q_\omega = \frac{\mathbb{E}\|\mathcal{Q}(z)\|_{bits}}{bd}.$$

## E    AUXILIARY LEMMAS

Lemma 1 reflects the general fact from the theory of saddle point problems.

**Lemma 1.** *If a function $f(x, y) : \mathcal{X} \times \mathcal{Y} \to \mathbb{R}$ is convex w.r.t. $x$ and concave w.r.t. $y$, then target operator $F$ for the min-max problem $\min_{x \in \mathcal{X}} \max_{y \in \mathcal{Y}} \{f(x, y)\}$ of the form*

$$F(z) = \begin{pmatrix} \nabla_x f(x, y) \\ -\nabla_y f(x, y) \end{pmatrix}$$

*is monotone e.i.,*

$$\langle F(z_1) - F(z_2), z_1 - z_2 \rangle \geq 0 \text{ for all } z_1, z_2 \in \mathcal{Z} = \mathcal{X} \times \mathcal{Y}.$$

*Proof.* We start from the definition of monotonicity, given in the statement, and utilize the convexity and concavity of $f$:

$$
\begin{aligned}
\langle F(z_1) - F(z_2), z_1 - z_2 \rangle &= \langle \nabla_x f(x_1, y_1) - \nabla_x f(x_2, y_2), x_1 - x_2 \rangle \\
&\quad - \langle \nabla_y f(x_1, y_1) - \nabla_y f(x_2, y_2), y_1 - y_2 \rangle \\
&= \langle \nabla_x f(x_1, y_1), x_1 - x_2 \rangle + \langle -\nabla_y f(x_1, y_1), y_1 - y_2 \rangle \\
&\quad + \langle \nabla_x f(x_2, y_2), x_2 - x_1 \rangle + \langle -\nabla_y f(x_2, y_2), y_2 - y_1 \rangle \\
&\geq f(x_1, y_1) - f(x_2, y_1) + f(x_1, y_2) - f(x_1, y_1)
\end{aligned}
$$

$$+f(x_2, y_2) - f(x_1, y_2) + f(x_2, y_1) - f(x_2, y_2) = 0.$$

$\square$

The following Lemma 2 (Lemma 3 in (Alacaoglu & Malitsky, 2022)) justifies the interchange of the maximum and the expectation operators, which is crucial for transitioning from the descent lemma to the actual convergence criterion in the main theorem.

**Lemma 2.** *Let $\mathcal{F} = \{\mathcal{F}_k\}_{k\geq}$ be a filtration and $u_k$ a stochastic process adopted to $\mathcal{F}$ with $\mathbb{E}[u_{k+1}|\mathcal{F}_k] = 0$. Then for any $K \in \mathbb{N}, z_a \in \mathcal{Z}$ and compact set $\mathcal{D} \subset \mathcal{Z}$ the following holds:*

$$\mathbb{E}\left[\max_{z \in \mathcal{D}} \sum_{k=0}^{K-1} \langle u_{k+1}, z \rangle\right] \leq \max_{z \in \mathcal{D}}\left(\frac{1}{2}\|z_a - z\|^2 + \frac{1}{2}\sum_{k=0}^{K-1} \mathbb{E}\|u_{k+1}\|^2\right). \tag{5}$$

*Proof.* Let $v_0 = z_a$, $v_{k+1} = v_k + u_{k+1}$. Since $u_k - \mathcal{F}_k$-measurable, $v_k - \mathcal{F}_k$-measurable as well. Then we write

$$\|v_{k+1} - z\|^2 = \|v_k - z\|^2 + 2\langle u_{k+1}, v_k - z \rangle + \|u_{k+1}\|^2.$$

Summing over $k = 0, 1, \ldots, K - 1$ we get

$$\sum_{k=0}^{K-1} 2\langle u_{k+1}, z - v_k \rangle \leq \|v_0 - z\|^2 + \sum_{k=0}^{K-1} \|u_{k+1}\|^2.$$

Maximizing and taking expectation we obtain

$$\mathbb{E}\left[\max_{z \in \mathcal{D}} \sum_{k=0}^{K-1} \langle u_{k+1}, z \rangle - \sum_{k=0}^{K-1} \langle u_{k+1}, v_k \rangle\right] \leq \frac{1}{2}\max_{z \in \mathcal{D}}\|v_0 - z\|^2 + \mathbb{E}\left[\frac{1}{2}\sum_{k=0}^{K-1} \|u_{k+1}\|^2\right].$$

Finally, due to $\mathcal{F}$-measurability of $v_k$ and by the tower property of conditional expectation, the second sum on the left-hand side vanishes. It concludes the proof. $\square$

## F  MISSING PROOFS

Now we are ready to start the main analysis. We proceed with the descent Lemma 3.

**Lemma 3.** *Let $\gamma_\pi = \gamma_\theta = \gamma$. Then, after $K$ iterations of Algorithm 1 solving problem* (4) *the following holds:*

$$
\begin{aligned}
2\gamma\langle F(z^{k+1}), z^{k+1} - z \rangle &\leq& \left(2D_{KL}(\pi, \pi^k) - 2D_{KL}(\pi, \pi^{k+1})\right) \\
&& + \left(\|\theta^k - \theta\|^2 - \|\theta^{k+1} - \theta\|^2\right) \\
&& + \left(2\gamma\alpha\langle F(z^k) - F(z^{k-1}), z - z^k \rangle\right. \\
&& \left. - 2\gamma\langle F(z^{k+1}) - F(z^k), z - z^{k+1} \rangle\right) \\
&& - \frac{1}{2}\|\pi^{k+1} - \pi^k\|^2 - \frac{1}{2}\|\theta^{k+1} - \theta^k\|^2 \\
&& + 2\gamma^2\|p^k - p^{k-1}\|^2 + 2\gamma^2\|g^k - g^{k-1}\|^2,
\end{aligned}
$$

*where $z = \begin{pmatrix} \theta \\ \pi \end{pmatrix}$ and $F(z^k) = \begin{pmatrix} g^k \\ -p^k \end{pmatrix}$.*

*Proof.* We proceed with algorithm steps evaluation.

Mirror descent step provides:

$$
\begin{aligned}
0 &\leq& \langle -\gamma\hat{p}^k + \nabla\psi(\pi^{k+1}) - \nabla\psi(\pi^k), \pi - \pi^{k+1} \rangle \\
&=& -\gamma\langle \hat{p}^k, \pi - \pi^{k+1} \rangle + D_{KL}(\pi, \pi^k) - D_{KL}(\pi, \pi^{k+1}) - D_{KL}(\pi^{k+1}, \pi^k).
\end{aligned}
$$

Rearranging it we reach:

$$
\begin{aligned}
D_{KL}(\pi, \pi^{k+1}) &\leq& D_{KL}(\pi, \pi^k) - D_{KL}(\pi^{k+1}, \pi^k) - \gamma\langle \hat{p}^k, \pi - \pi^{k+1} \rangle \\
&=& D_{KL}(\pi, \pi^k) - D_{KL}(\pi^{k+1}, \pi^k) - \gamma(1 + \alpha)\langle p^k, \pi - \pi^{k+1} \rangle
\end{aligned}
$$

$$+\gamma\alpha\langle p^{k-1}, \pi - \pi^{k+1}\rangle$$
$$= D_{KL}(\pi, \pi^k) - D_{KL}(\pi^{k+1}, \pi^k) - \gamma\langle p^k, \pi - \pi^{k+1}\rangle$$
$$-\gamma\alpha\langle p^k - p^{k-1}, \pi - \pi^{k+1}\rangle$$
$$= D_{KL}(\pi, \pi^k) - D_{KL}(\pi^{k+1}, \pi^k) - \gamma\langle p^k - p^{k+1}, \pi - \pi^{k+1}\rangle$$
$$-\gamma\langle p^{k+1}, \pi - \pi^{k+1}\rangle - \gamma\alpha\langle p^k - p^{k-1}, \pi - \pi^k\rangle$$
$$-\gamma\alpha\langle p^k - p^{k-1}, \pi^k - \pi^{k+1}\rangle. \tag{6}$$

$\theta$ update rule implies:

$$\|\theta^{k+1} - \theta\|^2 \leq \|\theta^k - \theta\|^2 + \|\theta^{k+1} - \theta^k\|^2 + 2\langle\theta^{k+1} - \theta^k, \theta^k - \theta\rangle$$
$$= \|\theta^k - \theta\|^2 - \|\theta^{k+1} - \theta^k\|^2 + 2\langle\theta^{k+1} - \theta^k, \theta^{k+1} - \theta\rangle$$
$$= \|\theta^k - \theta\|^2 - \|\theta^{k+1} - \theta^k\|^2 + 2\gamma\langle\hat{g}^k, \theta - \theta^{k+1}\rangle$$
$$= \|\theta^k - \theta\|^2 - \|\theta^{k+1} - \theta^k\|^2 + 2\gamma(1+\alpha)\langle g^k, \theta - \theta^{k+1}\rangle$$
$$-2\gamma\alpha\langle g^{k-1}, \theta - \theta^{k+1}\rangle$$
$$= \|\theta^k - \theta\|^2 - \|\theta^{k+1} - \theta^k\|^2 + 2\gamma\langle g^k, \theta - \theta^{k+1}\rangle$$
$$+2\gamma\alpha\langle g^k - g^{k-1}, \theta - \theta^{k+1}\rangle$$
$$= \|\theta^k - \theta\|^2 - \|\theta^{k+1} - \theta^k\|^2 + 2\gamma\langle g^k - g^{k+1}, \theta - \theta^{k+1}\rangle$$
$$+2\gamma\langle g^{k+1}, \theta - \theta^{k+1}\rangle + 2\gamma\alpha\langle g^k - g^{k-1}, \theta - \theta^k\rangle$$
$$+2\gamma\alpha\langle g^k - g^{k-1}, \theta^k - \theta^{k+1}\rangle. \tag{7}$$

Summing 2(6) and (7) we get:

$$2D_{KL}(\pi, \pi^{k+1}) + \|\theta^{k+1} - \theta\|^2$$
$$\leq 2D_{KL}(\pi, \pi^k) + \|\theta^k - \theta\|^2 - 2D_{KL}(\pi^{k+1}, \pi^k) - \|\theta^{k+1} - \theta^k\|^2$$
$$-2\gamma\langle p^k - p^{k+1}, \pi - \pi^{k+1}\rangle + 2\gamma\langle g^k - g^{k+1}, \theta - \theta^{k+1}\rangle$$
$$-2\gamma\langle p^{k+1}, \pi - \pi^{k+1}\rangle + 2\gamma\langle g^{k+1}, \theta - \theta^{k+1}\rangle$$
$$-2\gamma\alpha\langle p^k - p^{k-1}, \pi - \pi^k\rangle + 2\gamma\alpha\langle g^k - g^{k-1}, \theta - \theta^k\rangle$$
$$-2\gamma\alpha\langle p^k - p^{k-1}, \pi^k - \pi^{k+1}\rangle + 2\gamma\alpha\langle g^k - g^{k-1}, \theta^k - \theta^{k+1}\rangle.$$

Now we rewrite last inequality using $z = \begin{pmatrix}\theta\\\pi\end{pmatrix}$ and $F(z^k) = \begin{pmatrix}g^k\\-p^k\end{pmatrix}$.

$$2D_{KL}(\pi, \pi^{k+1}) + \|\theta^{k+1} - \theta\|^2$$
$$\leq 2D_{KL}(\pi, \pi^k) + \|\theta^k - \theta\|^2 - 2D_{KL}(\pi^{k+1}, \pi^k) - \|\theta^{k+1} - \theta^k\|^2$$
$$+2\gamma\langle F(z^k) - F(z^{k+1}), z - z^{k+1}\rangle + 2\gamma\langle F(z^{k+1}), z - z^{k+1}\rangle$$
$$+2\gamma\alpha\langle F(z^k) - F(z^{k-1}), z - z^k\rangle$$
$$-2\gamma\alpha\langle p^k - p^{k-1}, \pi^k - \pi^{k+1}\rangle + 2\gamma\alpha\langle g^k - g^{k-1}, \theta^k - \theta^{k+1}\rangle.$$

Pinsker's inequality (Pi) and (CS) with $\beta = 2\gamma$ provides

$$2D_{KL}(\pi, \pi^{k+1}) + \|\theta^{k+1} - \theta\|^2$$
$$\leq 2D_{KL}(\pi, \pi^k) + \|\theta^k - \theta\|^2 - \|\pi^{k+1} - \pi^k\|^2 - \|\theta^{k+1} - \theta^k\|^2$$
$$+2\gamma\langle F(z^k) - F(z^{k+1}), z - z^{k+1}\rangle + 2\gamma\langle F(z^{k+1}), z - z^{k+1}\rangle$$
$$+2\gamma\alpha\langle F(z^k) - F(z^{k-1}), z - z^k\rangle$$
$$+2\alpha\gamma^2\|p^k - p^{k-1}\|^2 + \frac{\alpha}{2}\|\pi^k - \pi^{k+1}\|_1^2$$
$$+2\alpha\gamma^2\|g^k - g^{k-1}\|^2 + \frac{\alpha}{2}\|\theta^k - \theta^{k+1}\|^2.$$

Finally, rearranging brings us to

$$2\gamma\langle F(z^{k+1}), z^{k+1} - z\rangle$$
$$\leq \left(2D_{KL}(\pi, \pi^k) - 2D_{KL}(\pi, \pi^{k+1})\right) + \left(\|\theta^k - \theta\|^2 - \|\theta^{k+1} - \theta\|^2\right)$$
$$+ \left(2\gamma\alpha\langle F(z^k) - F(z^{k-1}), z - z^k\rangle - 2\gamma\langle F(z^{k+1}) - F(z^k), z - z^{k+1}\rangle\right)$$
$$- \left(1 - \frac{\alpha}{2}\right)\|\pi^{k+1} - \pi^k\|_1^2 - \left(1 - \frac{\alpha}{2}\right)\|\theta^{k+1} - \theta^k\|^2$$
$$+ 2\alpha\gamma^2\|p^k - p^{k-1}\|^2 + 2\alpha\gamma^2\|g^k - g^{k-1}\|^2.$$

□

## F.1 ANALYSIS IN EXACT LOCAL GRADIENTS SETTING

For the subsequent analysis, we need recursive relations for the oracle distortion terms. For notational convenience we introduce $v^k = \mathbb{E}\left\|\tilde{f}^k - h^k\right\|^2, w^k = \sum_{i=1}^M \mathbb{E}\left\|\tilde{f}_i^k - h_i^k\right\|^2$.

**Lemma 4.** *Let Assumptions 1 and 2 hold. Then for iterations of Algorithm 1 with unbiased compressor 1 $\mathcal{Q}$ and exact local gradients holds:*

$$w^k \leq (1 + c_2^{-1})\left[6\frac{aL^2}{M}\mathbb{E}\left\|\theta^k - \theta^{k-1}\right\|^2 + 2\tilde{L}^2\mathbb{E}\left\|\pi^k - \pi^{k-1}\right\|_1^2\right]$$
$$+ (1 + c_2)(1 + \beta^2\omega - 2\beta)w^{k-1}. \tag{8}$$

*Proof.* We begin by using the explicit update rule of clients' local state.

$$w^k = \sum_{i=1}^M \mathbb{E}\|\tilde{f}_i^k - h_i^k\|^2 = \sum_{i=1}^M \mathbb{E}\|\tilde{f}_i^k - h_i^{k-1} - \beta\mathcal{Q}(\tilde{f}_i^{k-1} - h_i^{k-1})\|^2$$

$$= \sum_{i=1}^M \mathbb{E}\left\|\left(\tilde{f}_i^k - \tilde{f}_i^{k-1}\right) + \left(\tilde{f}_i^{k-1} - h_i^{k-1} - \beta\mathcal{Q}(\tilde{f}_i^{k-1} - h_i^{k-1})\right)\right\|^2$$

$$\overset{\text{(CS)}}{\leq} (1 + c_2^{-1})\sum_{i=1}^M \mathbb{E}\left\|\tilde{f}_i^k - \tilde{f}_i^{k-1}\right\|^2$$

$$+ (1 + c_2)\sum_{i=1}^M \mathbb{E}\left\|\tilde{f}_i^{k-1} - h_i^{k-1} - \beta\mathcal{Q}(\tilde{f}_i^{k-1} - h_i^{k-1})\right\|^2. \tag{9}$$

We estimate the first term:

$$\sum_{i=1}^M \mathbb{E}\left\|\tilde{f}_i^k - \tilde{f}_i^{k-1}\right\|^2 = \sum_{i=1}^M \mathbb{E}\left\|\pi_i^k\nabla f_i(\theta^k) - \pi_i^{k-1}\nabla f_i(\theta^{k-1})\right\|^2$$

$$\overset{\text{(CS)}}{\leq} 2\sum_{i=1}^M \mathbb{E}\left\|\pi_i^k\nabla f_i(\theta^k) - \pi_i^k\nabla f_i(\theta^{k-1})\right\|^2$$

$$+ 2\sum_{i=1}^M \mathbb{E}\left\|(\pi_i^k - \pi_i^{k-1})\nabla f_i(\theta^{k-1})\right\|^2$$

$$= 2\sum_{i=1}^M \mathbb{E}\pi_i^{k^2}\left\|\nabla f_i(\theta^k) - \nabla f_i(\theta^{k-1})\right\|^2$$

$$+ 2\sum_{i=1}^M \mathbb{E}\left|\pi_i^k - \pi_i^{k-1}\right|^2\left\|\nabla f_i(\theta^{k-1})\right\|^2$$

$$\overset{(i)}{\leq} 2\sum_{i=1}^M \mathbb{E}\pi_i^{k^2}L_i^2\left\|\theta^k - \theta^{k-1}\right\|^2 + 2\mathbb{E}\sum_{i=1}^M \left|\pi_i^k - \pi_i^{k-1}\right|^2\tilde{L}^2$$

$$
\begin{aligned}
&= \quad 2\mathbb{E}\left\|\theta^k - \theta^{k-1}\right\|^2 \sum_{i=1}^M \pi_i^{k^2} L_i^2 + 2\tilde{L}^2 \mathbb{E}\left\|\pi^k - \pi^{k-1}\right\|^2 \\
&\leq \quad 2\mathbb{E}\left\|\theta^k - \theta^{k-1}\right\|^2 \sum_{i=1}^M \pi_i^{k^2} L_i^2 + 2\tilde{L}^2 \mathbb{E}\left\|\pi^k - \pi^{k-1}\right\|_1^2. \quad (10)
\end{aligned}
$$

Where *(i)* holds due to Assumptions 1 and 2. We can bound $\sum_{i=1}^M \mathbb{E}\pi_i^{k^2} L_i^2$ using condition $\pi \in \Delta^{M-1} \cap Q_a^M$, where $Q_a^M = \left\{x \in \mathbb{R}^M \,\middle|\, 0 \leq x_i \leq \frac{a}{M}\right\}$ and $a \in [1, M]$:

$$
\begin{aligned}
\sum_{i=1}^M \pi_i^{k^2} L_i^2 &\leq \quad L^2 \sum_{i=1}^M \pi_i^{k^2} \leq L^2 \max_{x \in (\Delta^{M-1} \cap Q_a^M)} \|x\|^2 \\
&= \quad L^2 \left[\left(\frac{a}{M}\right)^2 \left\lceil \frac{M}{a} \right\rceil + \left(1 - \frac{a}{M}\left\lceil \frac{M}{a} \right\rceil\right)^2\right] \\
&\leq \quad L^2 \left[\left(\frac{a}{M}\right)^2 \left(\left\lceil \frac{M}{a} \right\rceil + 1\right)\right] \\
&\leq \quad L^2 \left[\left(\frac{a}{M}\right)^2 \left(\frac{M}{a} + 2\right)\right] \\
&\leq \quad L^2 \left[\left(\frac{a}{M}\right)^2 \frac{3M}{a}\right] = L^2 \frac{3a}{M}. \quad (11)
\end{aligned}
$$

Substitution of (11) into (10) gives

$$
\begin{aligned}
\sum_{i=1}^M \mathbb{E}\left\|\tilde{f}_i^k - \tilde{f}_i^{k-1}\right\|^2 &= \quad \sum_{i=1}^M \mathbb{E}\left\|\pi_i^k \nabla f_i(x^k) - \pi_i^{k-1} \nabla f_i(x^{k-1})\right\|^2 \\
&\leq \quad 6\frac{aL^2}{M}\mathbb{E}\left\|\theta^k - \theta^{k-1}\right\|^2 + 2\tilde{L}^2 \mathbb{E}\left\|\pi^k - \pi^{k-1}\right\|_1^2. \quad (12)
\end{aligned}
$$

Then we evaluate the second term of (9) RHS:

$$
\begin{aligned}
&\mathbb{E}\left\|\tilde{f}_i^{k-1} - h_i^{k-1} - \beta\mathcal{Q}(\tilde{f}_i^{k-1} - h_i^{k-1})\right\|^2 \\
&= \mathbb{E}\left\|\tilde{f}_i^{k-1} - h_i^{k-1}\right\|^2 + \beta^2 \mathbb{E}\left\|\mathcal{Q}(\tilde{f}_i^{k-1} - h_i^{k-1})\right\|^2 \\
&\quad - 2\mathbb{E}\left\langle \tilde{f}_i^{k-1} - h_i^{k-1}, \beta\mathcal{Q}(\tilde{f}_i^{k-1} - h_i^{k-1})\right\rangle \\
&\overset{1}{\leq} \mathbb{E}\left\|\tilde{f}_i^{k-1} - h_i^{k-1}\right\|^2 + \beta^2 \omega \mathbb{E}\left\|\tilde{f}_i^{k-1} - h_i^{k-1}\right\|^2 - 2\beta\mathbb{E}\|\tilde{f}_i^{k-1} - h_i^{k-1}\|^2 \\
&= (1 + \beta^2\omega - 2\beta)\mathbb{E}\left\|\tilde{f}_i^{k-1} - h_i^{k-1}\right\|^2. \quad (13)
\end{aligned}
$$

Finally, combining (9) with (12) and (13) we obtain:

$$
\begin{aligned}
w^k &\leq \quad (1 + c_2^{-1})\left[6\frac{aL^2}{M}\mathbb{E}\left\|\theta^k - \theta^{k-1}\right\|^2 + 2\tilde{L}^2 \mathbb{E}\left\|\pi^k - \pi^{k-1}\right\|_1^2\right] \\
&\quad + (1 + c_2)(1 + \beta^2\omega - 2\beta)\mathbb{E}\sum_{i=1}^M \left\|\tilde{f}_i^{k-1} - h_i^{k-1}\right\|^2 \\
&= \quad (1 + c_2^{-1})\left[6\frac{aL^2}{M}\mathbb{E}\left\|\theta^k - \theta^{k-1}\right\|^2 + 2\tilde{L}^2 \mathbb{E}\left\|\pi^k - \pi^{k-1}\right\|_1^2\right] \\
&\quad + (1 + c_2)(1 + \beta^2\omega - 2\beta)w^{k-1}.
\end{aligned}
$$

$\square$

We continue by examining the global oracle distortion evolution.

**Lemma 5.** *Let Assumptions 1 and 2 hold. Then for iterations of Algorithm 1 with unbiased compressor 1 $\mathcal{Q}$ and exact local gradients holds:*

$$
\begin{aligned}
v^k &\leq \left(1 + c_1^{-1}\right)\left(2L^2 \mathbb{E}\left\|\theta^k - \theta^{k-1}\right\|^2 + 2M\tilde{L}^2 \mathbb{E}\left\|\pi^k - \pi^{k-1}\right\|_1^2\right) \\
&\quad + (1 + c_1)\left(v^{k-1}(1 + \beta^2 - 2\beta) + \beta^2 w^{k-1}(\omega - 1)\right).
\end{aligned} \tag{14}
$$

*Proof.* We begin with the explicit global estimator update rule.

$$
\begin{aligned}
v^k &= \mathbb{E}\left\|\tilde{f}^k - h^k\right\|^2 = \mathbb{E}\left\|\tilde{f}^k - h^{k-1} - \beta\sum_{i=1}^M \mathcal{Q}(\tilde{f}_i^{k-1} - h_i^{k-1})\right\|^2 \\
&= \mathbb{E}\left\|\left(\tilde{f}^k - \tilde{f}^{k-1}\right) + \left(\tilde{f}^{k-1} - h^{k-1} - \beta\sum_{i=1}^M \mathcal{Q}(\tilde{f}_i^{k-1} - h_i^{k-1})\right)\right\|^2 \\
&\overset{(CS)}{\leq} \left(1 + c_1^{-1}\right)\mathbb{E}\left\|\tilde{f}^k - \tilde{f}^{k-1}\right\|^2 \\
&\quad + (1 + c_1)\mathbb{E}\left\|\tilde{f}^{k-1} - h^{k-1} - \beta\sum_{i=1}^M \mathcal{Q}(\tilde{f}_i^{k-1} - h_i^{k-1})\right\|^2.
\end{aligned} \tag{15}
$$

Then we examine first term on the (15) RHS.

As all $f_i$ are $L$-Lipschitz continuous (Assumption 2) the weighted sum $\sum_{i=1}^M \pi_i f_i$ is $L$-Lipschitz continuous as well. It justifies $(i)$ in following inequality sequence.

$$
\begin{aligned}
\mathbb{E}\left\|\tilde{f}^k - \tilde{f}^{k-1}\right\|^2 &= \mathbb{E}\left\|\sum_{i=1}^M \pi_i^k \nabla f_i(\theta^k) - \pi_i^{k-1}\nabla f_i(\theta^{k-1})\right\|^2 \\
&= \mathbb{E}\left\|\sum_{i=1}^M \pi_i^k\left(\nabla f_i(\theta^k) - \nabla f_i(\theta^{k-1})\right) + \left(\pi_i^k - \pi_i^{k-1}\right)\nabla f_i(\theta^{k-1})\right\|^2 \\
&\overset{(CS)}{\leq} 2\mathbb{E}\left\|\sum_{i=1}^M \pi_i^k\left(\nabla f_i(\theta^k) - \nabla f_i(\theta^{k-1})\right)\right\|^2 \\
&\quad + 2\mathbb{E}\left\|\sum_{i=1}^M \left(\pi_i^k - \pi_i^{k-1}\right)\nabla f_i(\theta^{k-1})\right\|^2 \\
&\overset{(i)}{\leq} 2L^2\mathbb{E}\left\|\theta^k - \theta^{k-1}\right\|^2 + 2\mathbb{E}\left(\sum_{i=1}^M \left|\pi_i^k - \pi_i^{k-1}\right|\left\|\nabla f_i(\theta^{k-1})\right\|\right)^2 \\
&\overset{1}{\leq} 2L^2\mathbb{E}\left\|\theta^k - \theta^{k-1}\right\|^2 + 2\mathbb{E}\left(\sum_{i=1}^M \left|\pi_i^k - \pi_i^{k-1}\right|\tilde{L}\right)^2 \\
&\leq 2L^2\mathbb{E}\left\|\theta^k - \theta^{k-1}\right\|^2 + 2\tilde{L}^2\mathbb{E}\left\|\pi^k - \pi^{k-1}\right\|_1^2.
\end{aligned} \tag{16}
$$

Now we concentrate on the second term on the (15) RHS:

$$
\begin{aligned}
&\mathbb{E}\left\|\tilde{f}^{k-1} - h^{k-1} - \beta\sum_{i=1}^M \mathcal{Q}(\tilde{f}_i^{k-1} - h_i^{k-1})\right\|^2 \\
&= \mathbb{E}\left\|\tilde{f}^{k-1} - h^{k-1}\right\|^2 + \beta^2\mathbb{E}\left\|\sum_{i=1}^M \mathcal{Q}(\tilde{f}_i^{k-1} - h_i^{k-1})\right\|^2 \\
&\quad - 2\beta\mathbb{E}\left\langle \tilde{f}^{k-1} - h^{k-1}, \sum_{i=1}^M \mathcal{Q}(\tilde{f}_i^{k-1} - h_i^{k-1})\right\rangle
\end{aligned}
$$

$$\stackrel{\substack{(1)\\(20)}}{\leq} \mathbb{E}\left\|\tilde{f}^{k-1} - h^{k-1}\right\|^2 + \beta^2\left(v^{k-1} + \omega w^{k-1}\right) - 2\beta\mathbb{E}\left\|\tilde{f}^{k-1} - h^{k-1}\right\|^2$$
$$= v^{k-1}(1 + \beta^2 - 2\beta) + \beta^2\omega w^{k-1}. \tag{17}$$

Plugging (17) and (16) into (15) yields:

$$
\begin{aligned}
v^k &\leq \left(1 + c_1^{-1}\right)\left(2L^2\mathbb{E}\left\|\theta^k - \theta^{k-1}\right\|^2 + 2M\tilde{L}^2\mathbb{E}\left\|\pi^k - \pi^{k-1}\right\|_1^2\right)\\
&\quad + (1 + c_1)\left(v^{k-1}(1 + \beta^2 - 2\beta) + \beta^2 w^{k-1}(\omega - 1)\right).
\end{aligned}
$$

$\square$

The last preparation before proceeding to the main theorem is evaluation of global state dynamics.

**Lemma 6.** *For iterations of Algorithm 1 with unbiased compressor 1 $\mathcal{Q}$, the following holds:*

$$2\gamma^2\mathbb{E}\|g^k - g^{k-1}\|^2 = 4\gamma^2(\omega - 1)\left(w^k + (1 - \beta)^2 w^{k-1}\right) + 4\gamma^2\left(v^k + (1 - \beta)^2 v^{k-1}\right). \tag{18}$$

*Proof.* Let us again begin with the explicit global estimator update rule.

$$
\begin{aligned}
&2\gamma^2\mathbb{E}\|g^k - g^{k-1}\|^2\\
&= 2\gamma^2\mathbb{E}\|h^k + \hat{\Delta}^k - h^{k-1} - \hat{\Delta}^{k-1}\|^2\\
&= 2\gamma^2\mathbb{E}\left\|\sum_{i=1}^M\left(h_i^k - h_i^{k-1}\right) + \sum_{i=1}^M\mathcal{Q}\left(\tilde{f}_i^k - h_i^k\right) - \sum_{i=1}^M\mathcal{Q}\left(\tilde{f}_i^{k-1} - h_i^{k-1}\right)\right\|^2\\
&= 2\gamma^2\mathbb{E}\left\|\beta\sum_{i=1}^M\mathcal{Q}\left(\tilde{f}_i^{k-1} - h_i^{k-1}\right) + \sum_{i=1}^M\mathcal{Q}\left(\tilde{f}_i^k - h_i^k\right) - \sum_{i=1}^M\mathcal{Q}\left(\tilde{f}_i^{k-1} - h_i^{k-1}\right)\right\|^2\\
&= 2\gamma^2\mathbb{E}\left\|\sum_{i=1}^M\mathcal{Q}\left(\tilde{f}_i^k - h_i^k\right) - (1 - \beta)\sum_{i=1}^M\mathcal{Q}\left(\tilde{f}_i^{k-1} - h_i^{k-1}\right)\right\|^2\\
&\stackrel{\text{(CS)}}{=} 4\gamma^2\mathbb{E}\left\|\sum_{i=1}^M\mathcal{Q}\left(\tilde{f}_i^k - h_i^k\right)\right\|^2 + 4\gamma^2(1 - \beta)^2\mathbb{E}\left\|\sum_{i=1}^M\mathcal{Q}\left(\tilde{f}_i^{k-1} - h_i^{k-1}\right)\right\|^2. \tag{19}
\end{aligned}
$$

Terms differ only in their indices, which makes it convenient to analyze them separately. Here we utilize cross-device compressor independence and unbiasedness once again:

$$
\begin{aligned}
\mathbb{E}\left\|\sum_{i=1}^M\mathcal{Q}\left(\tilde{f}_i^k - h_i^k\right)\right\|^2 &= \sum_{i=1}^M\mathbb{E}\left\|\mathcal{Q}\left(\tilde{f}_i^k - h_i^k\right)\right\|^2 + \sum_{i\neq j}\mathbb{E}\left\langle\mathcal{Q}\left(\tilde{f}_i^k - h_i^k\right), \mathcal{Q}\left(\tilde{f}_j^k - h_j^k\right)\right\rangle\\
&= \sum_{i=1}^M\omega\mathbb{E}\left\|\tilde{f}_i^k - h_i^k\right\|^2 + \sum_{i\neq j}\mathbb{E}\left\langle\mathcal{Q}\left(\tilde{f}_i^k - h_i^k\right), \mathcal{Q}\left(\tilde{f}_j^k - h_j^k\right)\right\rangle\\
&\leq \omega\sum_{i=1}^M\mathbb{E}\left\|\tilde{f}_i^k - h_i^k\right\|^2 + \sum_{i\neq j}\mathbb{E}\left\langle\tilde{f}_i^k - h_i^k, \tilde{f}_j^k - h_j^k\right\rangle\\
&= (\omega - 1)\sum_{i=1}^M\mathbb{E}\left\|\tilde{f}_i^k - h_i^k\right\|^2 + \mathbb{E}\left\|\tilde{f}^k - h^k\right\|^2\\
&= (\omega - 1)w^k + v^k. \tag{20}
\end{aligned}
$$

Substituting (20) into (19) we reach

$$2\gamma^2\mathbb{E}\|g^k - g^{k-1}\|^2 = 4\gamma^2(\omega - 1)\left(w^k + (1 - \beta)^2 w^{k-1}\right) + 4\gamma^2\left(v^k + (1 - \beta)^2 v^{k-1}\right).$$

$\square$

Finally, let us introduce the convergence criterion. In saddle point problems under the convex-concave setting convergence measures in term of the **Gap** function (Definition 2). Since ADI

incorporates possibly randomized compression operator $\mathcal{Q}$, the convergence guaranties for it is based on $\mathbb{E}\left[\mathbf{Gap}(z)\right]$. This guaranties are provided by Theorem 1.

**Theorem 1.** *Let Assumptions 1, 2, 3 hold and $\alpha = 1$, $\beta = \frac{1}{\omega}$, $H = 32\gamma^2\omega^2$, $N = 7\gamma^2\omega$,*
$\gamma_\pi = \gamma_\theta = \gamma \leq \gamma_0 = \min\left\{\frac{1}{2\tilde{L}}\sqrt{\frac{1}{96\omega^3 + 14M\omega^2}}, \sqrt{\frac{1}{2}\frac{1}{4M\tilde{L}^2 + 576\frac{a\omega^3}{M}L^2 + 28\omega^2 L^2}}\right\}$, $\Lambda = \Delta^{M-1} \cap Q_a^M$,
*where $Q_a^M = \{x \in \mathbb{R}^M | 0 \leq x_i \leq \frac{a}{M}\}$. Then, after $K$ iterations of Algorithm 1 with unbiased compressor 1 $\mathcal{Q}$ and exact local gradients solving problem* (4) *the following holds:*

$$\mathbb{E}\left[\mathbf{Gap}(\overline{z}_K)\right] \leq \frac{V}{2\gamma K},$$

*where*

$$V = \mathbb{E}\left[\max_{z \in \mathcal{D}}\left\{4D_{KL}(\pi, \pi^1) + 2\|\theta^1 - \theta\|^2 + 2\gamma\langle F(z^1) - F(z^0), z - z^1\rangle\right\}\right.$$

$$\left. + H\sum_{k=0}^{1}\sum_{i=1}^{M}\left\|\tilde{f}_i^k - h_i^k\right\|^2 + N\sum_{k=0}^{1}\left\|\tilde{f}^k - h^k\right\|^2\right] \quad and \quad \overline{z}_K = \frac{1}{K}\sum_{k=1}^{K}z^k.$$

*Proof.* We proceed with using the unbiasedness (2) of compressor $\mathcal{Q}$:

$$\mathbb{E}\left[F(z^k)|z^k\right] = \mathbb{E}\left[\begin{pmatrix}g^k \\ p^k\end{pmatrix}\Big| z^k\right] = \mathbb{E}\left[\begin{pmatrix}h^k + \sum_{i=1}^{M}\mathcal{Q}(\tilde{f}_i^k - h_i^k) \\ p^k\end{pmatrix}\Big| z^k\right] = \begin{pmatrix}\tilde{f}^k \\ p^k\end{pmatrix} \overset{def}{=} \overline{F}(z^k),$$

where $\tilde{f}^k = \sum_{i=1}^{M}\tilde{f}_i^k = \sum_{i=1}^{M}\pi_i^k\nabla f_i(\theta^k)$. Considering $f(\theta, \pi) = \sum_{i=1}^{M}\pi_i f_i(\theta)$ we note that it is convex with respect to $\theta$ due to convexity of all $f_i$. At the same time, $f$ is linear, and therefore concave with respect to all $\pi_i$. Then, noting that $\overline{F}(z) = \begin{pmatrix}\nabla_\theta f(\theta, \pi) \\ -\nabla_\pi f(\theta, \pi)\end{pmatrix}$, we invoke Lemma 1 to establish its monotonicity.

Our objective is to obtain convergence with respect to $\mathbf{Gap}(z) = \max_{x \in \mathcal{D}}\left\{\langle \overline{F}(x), z - x\rangle\right\}$. Hence, the next step is conditioning the result of Lemma 3 on $z^{k+1}$, using $\alpha = 1$ and summing over $k = 1$ to $K$,

$$2\gamma\sum_{k=1}^{K}\langle\overline{F}(z^{k+1}), z^{k+1} - z\rangle$$

$$\leq \sum_{k=1}^{K}\left[\left(2D_{KL}(\pi, \pi^k) - 2D_{KL}(\pi, \pi^{k+1})\right) + \left(\|\theta^k - \theta\|^2 - \|\theta^{k+1} - \theta\|^2\right)\right.$$

$$+ \left(2\gamma\langle F(z^k) - F(z^{k-1}), z - z^k\rangle - 2\gamma\langle\overline{F}(z^{k+1}) - F(z^k), z - z^{k+1}\rangle\right)$$

$$- \frac{1}{2}\|\pi^{k+1} - \pi^k\|_1^2 - \frac{1}{2}\|\theta^{k+1} - \theta^k\|^2$$

$$\left. + 2\gamma^2\|p^k - p^{k-1}\|^2 + 2\gamma^2\|g^k - g^{k-1}\|^2\right]$$

$$= \left(2D_{KL}(\pi, \pi^1) - 2D_{KL}(\pi, \pi^{K+1})\right) + \left(\|\theta^1 - \theta\|^2 - \|\theta^{K+1} - \theta\|^2\right)$$

$$+ \left(2\gamma\langle F(z^1) - F(z^0), z - z^1\rangle - 2\gamma\langle\overline{F}(z^{K+1}) - F(z^K), z - z^{K+1}\rangle\right)$$

$$+ \sum_{k=1}^{K-1}\left[2\gamma\langle\overline{F}(z^{k+1}) - F(z^{k+1}), z - z^{k+1}\rangle\right]$$

$$+ \sum_{k=1}^{K}\left[-\frac{1}{2}\|\pi^{k+1} - \pi^k\|_1^2 - \frac{1}{2}\|\theta^{k+1} - \theta^k\|^2\right.$$

$$\left. + 2\gamma^2\|p^k - p^{k-1}\|^2 + 2\gamma^2\|g^k - g^{k-1}\|^2\right].$$

Maximizing obtained inequality over compact set $z \in \mathcal{D}$ and taking full expectation, we get

$$2\gamma\,\mathbb{E}\left[\max_{z \in \mathcal{D}}\left\{\sum_{k=1}^{K}\langle\overline{F}(z^{k+1}), z^{k+1} - z\rangle\right\}\right] \leq \mathbb{E}\left[\max_{z \in \mathcal{D}}\left\{\left(2D_{KL}(\pi, \pi^1) - 2D_{KL}(\pi, \pi^{K+1})\right)\right.\right.$$

$$+ \left( \|\theta^1 - \theta\|^2 - \|\theta^{K+1} - \theta\|^2 \right)$$
$$+ \left( 2\gamma \langle F(z^1) - F(z^0), z - z^1 \rangle \right.$$
$$- 2\gamma \langle \overline{F}(z^{K+1}) - F(z^K), z - z^{K+1} \rangle )$$
$$+ \sum_{k=1}^{K-1} 2\gamma \langle \overline{F}(z^{k+1}) - F(z^{k+1}), z - z^{k+1} \rangle \Big\}$$
$$+ \sum_{k=1}^{K} \Big[ -\frac{1}{2} \|\pi^{k+1} - \pi^k\|_1^2 - \frac{1}{2} \|\theta^{k+1} - \theta^k\|^2$$
$$+ 2\gamma^2 \|p^k - p^{k-1}\|^2 + 2\gamma^2 \|g^k - g^{k-1}\|^2 \Big] \Big]. \quad (21)$$

Several next steps evaluate different terms of (21), starting with the LHS.

Due to monotonicity of $\overline{F}$,

$$\sum_{k=1}^{K} \langle \overline{F}(z^k), z^k - z \rangle \geq \sum_{k=1}^{K} \langle \overline{F}(z), z^k - z \rangle = K \left\langle \overline{F}(z), \frac{1}{K} \sum_{k=1}^{K} z^k - z \right\rangle.$$

Maximizing over $z$ we get.

$$K \mathbf{Gap}\left( \overline{z}_K \right) \leq \max_{z \in \mathcal{D}} \left\{ \sum_{k=1}^{K} \langle \overline{F}(z^k), z^k - z \rangle \right\}. \quad (22)$$

We apply Lemma 2 to bound the first sum on the RHS of (21):

$$2\mathbb{E} \left[ \max_{z \in \mathcal{D}} \left\{ \sum_{k=1}^{K-1} \left\langle \gamma \left( \overline{F}(z^{k+1}) - F(z^{k+1}) \right), z - z^{k+1} \right\rangle \right\} \right]$$
$$= 2\mathbb{E} \left[ \max_{z \in \mathcal{D}} \left\{ \sum_{k=1}^{K-1} \left\langle \gamma \left( \overline{F}(z^{k+1}) - F(z^{k+1}) \right), z \right\rangle \right\} \right]$$
$$- \sum_{k=1}^{K-1} \mathbb{E} \left\langle \gamma \left( \overline{F}(z^{k+1}) - F(z^{k+1}) \right), z^{k+1} \right\rangle$$
$$= 2\mathbb{E} \left[ \max_{z \in \mathcal{D}} \left\{ \sum_{k=1}^{K-1} \left\langle \gamma \left( \overline{F}(z^{k+1}) - F(z^{k+1}) \right), z \right\rangle \right\} \right] - 0$$
$$\leq \max_{z \in \mathcal{D}} \left( \|z_a - z\|^2 \right) + \gamma^2 \sum_{k=1}^{K-1} \mathbb{E} \left\| \overline{F}(z^{k+1}) - F(z^{k+1}) \right\|^2. \quad (23)$$

We continue with evaluating of the last term applying properties (2) of unbiased compressor $\mathcal{Q}$:

$$\mathbb{E} \left\| \overline{F}(z^k) - F(z^k) \right\|^2 = \mathbb{E} \left\| \begin{pmatrix} \tilde{f}^k \\ p^k \end{pmatrix} - \begin{pmatrix} h^k + \sum_{i=1}^{M} \mathcal{Q}(\tilde{f}_i^k - h_i^k) \\ p^k \end{pmatrix} \right\|^2$$
$$= \mathbb{E} \left\| \sum_{i=1}^{M} \mathcal{Q}(\tilde{f}_i^k - h_i^k) - (\tilde{f}^k - h^k) \right\|^2$$
$$= \sum_{i=1}^{M} \mathbb{E} \left\| \mathcal{Q}(\tilde{f}_i^k - h_i^k) \right\|^2 + \mathbb{E} \left\| \tilde{f}^k - h^k \right\|^2$$
$$- 2 \sum_{i=1}^{M} \mathbb{E} \left\langle \mathcal{Q}(\tilde{f}_i^k - h_i^k), \tilde{f}^k - h^k \right\rangle$$
$$\leq \omega \sum_{i=1}^{M} \mathbb{E} \left\| \tilde{f}_i^k - h_i^k \right\|^2 - \mathbb{E} \left\| \tilde{f}^k - h^k \right\|^2. \quad (24)$$

Finally, using (Pi), the definition of $z = \begin{pmatrix} \theta \\ \pi \end{pmatrix}$ and choosing $z_a = z^1 - z$, we estimate

$$
\max_{z \in \mathcal{D}} \left\{ 2D_{KL}(\pi, \pi^1) + \|\theta^1 - \theta\|^2 \right\} \geq \max_{z \in \mathcal{D}} \left\{ \|\pi^1 - \pi\|^2 + \|\theta^1 - \theta\|^2 \right\}
$$
$$
= \max_{z \in \mathcal{D}} \left\{ \|z_a - z\|^2 \right\}. \tag{25}
$$

Using notation $v^k = \mathbb{E} \left\| \tilde{f}^k - h^k \right\|^2$, $w^k = \sum_{i=1}^M \mathbb{E} \left\| \tilde{f}_i^k - h_i^k \right\|^2$ and combining (23),(24) with (25) we derive:

$$
2\mathbb{E} \left[ \max_{z \in \mathcal{D}} \left\{ \sum_{k=1}^{K-1} \left\langle \gamma \left( \overline{F}(z^{k+1}) - F(z^{k+1}) \right), z - z^{k+1} \right\rangle \right\} \right]
$$
$$
\leq \max_{z \in \mathcal{D}} \left\{ 2D_{KL}(\pi, \pi^1) + \|\theta^1 - \theta\|^2 \right\} + \gamma^2 \sum_{k=2}^K \left( \omega w^k - v^k \right). \tag{26}
$$

After that, we estimate $2\gamma^2 \|p^k - p^{k-1}\|^2$ in (21) via Assumption 1:

$$
2\gamma^2 \|p^k - p^{k-1}\|^2 \leq 2\gamma^2 M \tilde{L}^2 \|\theta^k - \theta^{k-1}\|^2. \tag{27}
$$

Finally, we use (18) to evaluate the sum:

$$
2\gamma^2 \sum_{k=1}^K \mathbb{E}\|g^k - g^{k-1}\|^2
$$
$$
= 4\gamma^2 \sum_{k=1}^K \left[ (\omega - 1)w^k + v^k \right] + 4\gamma^2 (1 - \beta)^2 \sum_{k=1}^K \left[ (\omega - 1)w^{k-1} + v^{k-1} \right]
$$
$$
= 4\gamma^2 \sum_{k=1}^K \left[ (\omega - 1)w^k + v^k \right] + 4\gamma^2 (1 - \beta)^2 \sum_{k=0}^{K-1} \left[ (\omega - 1)w^k + v^k \right]
$$
$$
= 4\gamma^2 (1 + (1 - \beta)^2) \left[ \sum_{k=0}^K (\omega - 1)w^k + v^k \right]
$$
$$
- 4\gamma^2 (1 - \beta)[(\omega - 1)w^K + v^K]
$$
$$
- 4\gamma^2 [(\omega - 1)w^0 + v^0]. \tag{28}
$$

Substituting (28), (27), (26) and (22) into (21) we get

$$
2\gamma K \mathbb{E}\left[ \mathbf{Gap}(\overline{z}_K) \right] \leq \mathbb{E}\Bigg[ \max_{z \in \mathcal{D}} \Big\{ \left( 4D_{KL}(\pi, \pi^1) - 2D_{KL}(\pi, \pi^{K+1}) \right)
$$
$$
+ \left( 2\|\theta^1 - \theta\|^2 - \|\theta^{K+1} - \theta\|^2 \right)
$$
$$
+ \left( 2\gamma \langle F(z^1) - F(z^0), z - z^1 \rangle - 2\gamma \langle \overline{F}(z^{K+1}) - F(z^K), z - z^{K+1} \rangle \right) \Big\}
$$
$$
- 4\gamma^2 (1 - \beta)[(\omega - 1)w^K + v^K] - 4\gamma^2 [(\omega - 1)w^0 + v^0]
$$
$$
+ \gamma^2 \sum_{k=2}^K \left( \omega w^k - v^k \right) + 4\gamma^2 (1 + (1 - \beta)^2) \sum_{k=0}^K \left[ (\omega - 1)w^k + v^k \right]
$$
$$
+ \sum_{k=1}^K \left[ -\frac{1}{2}\|\pi^{k+1} - \pi^k\|_1^2 - \left( \frac{1}{2} - 2\gamma^2 M \tilde{L}^2 \right) \|\theta^{k+1} - \theta^k\|^2 \right] \Bigg].
$$

Using (24) as $\omega w^k - v^k \geq 0$ and $0 < \beta < 1$, and introducing

$$
\Xi_K = \max_{z \in \mathcal{D}} \Big\{ \left( 4D_{KL}(\pi, \pi^1) - 2D_{KL}(\pi, \pi^{K+1}) \right)
$$
$$
+ \left( 2\|\theta^1 - \theta\|^2 - \|\theta^{K+1} - \theta\|^2 \right)
$$
$$
+ \left( 2\gamma \langle F(z^1) - F(z^0), z - z^1 \rangle
$$

$$-2\gamma\langle \overline{F}(z^{K+1}) - F(z^K), z - z^{K+1})\rangle\Big\}$$
$$-4\gamma^2(1-\beta)[(\omega-1)w^K + v^K] - 4\gamma^2[(\omega-1)w^0 + v^0], \tag{29}$$

we can rewrite:

$$2\gamma K \mathbb{E}\left[\mathbf{Gap}(\overline{z}_K)\right] \leq \mathbb{E}\Big[\Xi_K + \sum_{k=0}^{K}\left[9\omega\gamma^2 w^k + 7\gamma^2 v^k\right]$$
$$+ \sum_{k=1}^{K}\left[-\frac{1}{2}\|\pi^{k+1} - \pi^k\|_1^2 - \left(\frac{1}{2} - 2\gamma^2 M\tilde{L}^2\right)\|\theta^{k+1} - \theta^k\|^2\right]\Big]. \tag{30}$$

The next step is summing (30) $+ \sum_{k=2}^{K+1}[H \cdot (8) + N \cdot (14)]$:

$$2\gamma K \mathbb{E}\left[\mathbf{Gap}(\overline{z}_K)\right] + \sum_{k=2}^{K+1}(Hw^k + Nv^k)$$

$$\leq \mathbb{E}\Big[\Xi_K + \sum_{k=0}^{K}\left[9\omega\gamma^2 w^k + 7\gamma^2 v^k\right] + \sum_{k=1}^{K}\left[H(1+c_2)(1+\beta^2\omega - 2\beta)w^k\right]$$

$$+ \sum_{k=1}^{K}\left[N(1+c_1)\left(v^k(1+\beta^2 - 2\beta) + \beta^2 w^k(\omega-1)\right)\right]$$

$$+ \sum_{k=1}^{K}\left[-\left(\frac{1}{2} - 2H\tilde{L}^2(1+c_2^{-1}) - 2NM\tilde{L}^2(1+c_1^{-1})\right)\|\pi^{k+1} - \pi^k\|_1^2\right.$$

$$\left. - \left(\frac{1}{2} - 2\gamma^2 M\tilde{L}^2 - H(1+c_2^{-1})\frac{6aL^2}{M} - N\left(1+c_1^{-1}\right)2L^2\right)\|\theta^{k+1} - \theta^k\|^2\right]\Big].$$

By rearranging the terms, we obtain

$$2\gamma K \mathbb{E}\left[\mathbf{Gap}(\overline{z}_K)\right] + \sum_{k=2}^{K+1}(Hw^k + Nv^k)$$

$$\leq \mathbb{E}\Big[\Xi_K + \sum_{k=0}^{K}\left[7\gamma^2 + N(1+c_1)(1+\beta^2 - 2\beta)\right]v^k$$

$$+ \sum_{k=0}^{K}\left[9\omega\gamma^2 + H(1+c_2)(1+\beta^2\omega - 2\beta) + N(1+c_1)\beta^2(\omega-1)\right]w^k$$

$$+ \sum_{k=1}^{K}\left[-\left(\frac{1}{2} - 2H\tilde{L}^2(1+c_2^{-1}) - 2NM\tilde{L}^2(1+c_1^{-1})\right)\|\pi^{k+1} - \pi^k\|_1^2\right.$$

$$\left. - \left(\frac{1}{2} - 2\gamma^2 M\tilde{L}^2 - H(1+c_2^{-1})\frac{6aL^2}{M} - N\left(1+c_1^{-1}\right)2L^2\right)\|\theta^{k+1} - \theta^k\|^2\right]\Big]. \tag{31}$$

Considering the respective coefficients of $\|\theta^{k+1} - \theta^k\|^2$, $\|\pi^{k+1} - \pi^k\|_1^2$, $w^k$ and $v^k$, we derive the following restrictions:

$$\begin{cases} \frac{1}{2} \geq 4\gamma^2 M\tilde{L}^2 + H(1+c_2^{-1})\frac{6aL^2}{M} + N\left(1+c_1^{-1}\right)2L^2, \\ \frac{1}{2} \geq 2H\tilde{L}^2(1+c_2^{-1}) + 2NM\tilde{L}^2(1+c_1^{-1}), \\ H \geq 9\omega\gamma^2 + H(1+c_2)(1+\beta^2\omega - 2\beta) + N(1+c_1)\beta^2(\omega-1), \\ N \geq 7\gamma^2 + N(1+c_1)(1+\beta^2 - 2\beta). \end{cases} \tag{32}$$

We now turn to selecting the free coefficients to satisfy conditions (32). Beginning with the last inequality on $N$, we set

$$c_1 = \beta, \text{ which yields } N = \frac{7\gamma^2}{\beta} \tag{33}$$

is sufficient.

With this selection the third restriction in (32) transforms into

$$H\left(1 - (1+c_2)(1+\beta^2\omega - 2\beta)\right) \geq 9\omega\gamma^2 + \frac{7\gamma^2}{\beta}(1+\beta)\beta^2(\omega - 1).$$

The choice

$$c_2 = \frac{\beta}{2}, \ \beta = \frac{1}{\omega} \text{ guarantees sufficiency of } H = \frac{32\gamma^2}{\beta^2}. \tag{34}$$

Then, utilizing (34) and (33) we rewrite the second inequality in (32):

$$\frac{1}{2} \geq 2\frac{32\gamma^2}{\beta^2}\tilde{L}^2(1 + 2\beta^{-1}) + 2\frac{7\gamma^2}{\beta}M\tilde{L}^2(1 + \beta^{-1}).$$

This poses constrain on $\gamma$:

$$\gamma \leq \sqrt{\frac{1}{2}\frac{1}{192\omega^3\tilde{L}^2 + 28M\omega^2\tilde{L}^2}} = \frac{1}{2\tilde{L}}\sqrt{\frac{1}{96\omega^3 + 14M\omega^2}}. \tag{35}$$

Finally, we examine the first inequality in (32). Using (33) and (34) we derive:

$$\frac{1}{2} \geq 4\gamma^2M\tilde{L}^2 + \frac{32\gamma^2}{\beta^2}(1 + 2\beta^{-1})\frac{6aL^2}{M} + \frac{7\gamma^2}{\beta}\left(1 + \beta^{-1}\right)2L^2,$$

$$\gamma \ \leq \ \sqrt{\frac{1}{2}\frac{1}{4M\tilde{L}^2 + 576L^2\beta^{-3}\frac{a}{M} + 28L^2\beta^{-2}}}$$

$$= \ \sqrt{\frac{1}{2}\frac{1}{4M\tilde{L}^2 + 576\frac{a\omega^3}{M}L^2 + 28\omega^2L^2}}. \tag{36}$$

By choosing

$$\gamma = \min\left\{\frac{1}{2\tilde{L}}\sqrt{\frac{1}{96\omega^3 + 14M\omega^2}}, \sqrt{\frac{1}{2}\frac{1}{4M\tilde{L}^2 + 576\frac{a\omega^3}{M}L^2 + 28\omega^2L^2}}\right\}, \tag{37}$$

and taking (35), (36) into account, we satisfy (32). Consequently, with the definition $\Xi_K$ (29) substitution, (31) transforms into

$$2\gamma K\mathbb{E}\left[\mathbf{Gap}(\overline{z}_K)\right] + (Hw^{K+1} + Nv^{K+1})$$
$$\leq \mathbb{E}\Bigg[\max_{z\in\mathcal{D}}\Big\{\left(4D_{KL}(\pi, \pi^1) - 2D_{KL}(\pi, \pi^{K+1})\right)$$
$$+ \left(2\|\theta^1 - \theta\|^2 - \|\theta^{K+1} - \theta\|^2\right)$$
$$+ \left(2\gamma\langle F(z^1) - F(z^0), z - z^1\rangle - 2\gamma\langle\overline{F}(z^{K+1}) - F(z^K), z - z^{K+1}\rangle\right)\Big\}$$
$$- 4\gamma^2(1-\beta)[(\omega - 1)w^K + v^K] - 4\gamma^2[(\omega - 1)w^0 + v^0]$$
$$+ H\sum_{k=0}^{1}w^k + N\sum_{k=0}^{1}v^k - \sum_{k=1}^{K}2\gamma^2M\tilde{L}^2\|\theta^{k+1} - \theta^k\|. \tag{38}$$

To proof the convergence we need to eliminate the $-2\gamma\langle\overline{F}(z^{K+1}) - F(z^K), z - z^{K+1}\rangle$ term.

$$-2\gamma\langle\overline{F}(z^{K+1}) - F(z^K), z - z^{K+1}\rangle \overset{\text{(Fen)}}{\leq} \gamma^2\|\overline{F}(z^{K+1}) - F(z^K)\|^2 + \|z - z^{K+1}\|^2$$
$$\overset{(i)}{\leq} 2\gamma^2\|\overline{F}(z^{K+1}) - F(z^{K+1})\|^2 + 2\gamma^2\|F(z^{K+1}) - F(z^K)\|^2$$
$$+ \|\theta - \theta^{K+1}\|^2 + 2D_{KL}(\pi, \pi^{K+1})$$
$$\overset{(ii)}{\leq} \|\theta - \theta^{K+1}\|^2 + 2D_{KL}(\pi, \pi^{K+1}) + 2\tilde{L}^2\gamma^2\|\theta^{K+1} - \theta^K\|^2$$

$$+4\gamma^2(\omega-1)\left(w^{K+1}+(1-\beta)^2 w^K\right)+4\gamma^2\left(v^{K+1}+(1-\beta)^2 v^K\right)$$
$$+2\gamma^2\omega w^{K+1}-2\gamma^2 v^{K+1} \tag{39}$$

Where $(i)$ holds by (CS) and (Pi), and $(ii)$ follows from (24), (27), and (18).

Then, the choice of $H$ (34) and $N$ (33) along with (39) provides

$$-2\gamma\langle\overline{F}(z^{K+1})-F(z^K),z-z^{K+1}\rangle-\sum_{k=1}^{K}2\gamma^2 M\tilde{L}^2\|\theta^{k+1}-\theta^k\|^2$$
$$-(Hw^{K+1}+Nv^{K+1})-2D_{KL}(\pi,\pi^{K+1})-\|\theta^{K+1}-\theta\|^2\le 0. \tag{40}$$

The substitution of (40) into (38) concludes the proof. $\qquad\square$

Theorem 1 yields further bounds on the number of communication rounds and the amount of information transmitted from the clients to the server.

**Remark 1.** *In our analysis we assume that compression does not reduce the size of $\hat{\Delta}_i$ below that of the scalar $f_i$ (i.e., $q_\omega\ge\frac{1}{d}$). Hence, the cost of transmitting $f_i$ can be upper bounded by that of $\hat{\Delta}_i$. This allows us to ignore the communication of $f_i$ in the $\mathcal{O}$ notation.*

**Corollary 1** *In setting of Theorem 1 with $\gamma=\gamma_0$, Algorithm 1 with exact local gradients needs*

$$\mathcal{O}\left(\frac{1}{\varepsilon}\left[\tilde{L}\omega^{3/2}+\tilde{L}M^{1/2}\omega+L\left(\sqrt{\frac{a\omega^3}{M}}+\omega\right)\right]\right)$$

*iterations in order to reach $\varepsilon$-accuracy with respect to $\mathbb{E}\left[\mathbf{Gap}(\overline{z}_K)\right]$. Additionally, it requires*

$$\mathcal{O}\left(\frac{1}{\varepsilon}\left[\tilde{L}\omega^{1/2}+\tilde{L}M^{1/2}+L\left(\sqrt{\frac{a\omega}{M}}+1\right)\right]\right)$$

*bits communicated from nodes to the server.*

*Proof.* The result of Theorem 1 directly provides the first bound.

Given Remark 1, to obtain the second estimate from the first, we consider transmitting $\hat{\Delta}_i^k$ from the nodes to the server for $i=1,2,\ldots,M$. This corresponds to sending $Mdbq_\omega$ bits at every iteration. We omit constants $M,d$ and $b$ under the $\mathcal{O}$ notation. As for $q_\omega$, we note that for practically relevant compressors (Beznosikov et al., 2023a) it holds $q_\omega\le\frac{1}{\omega}$. It concludes the proof. $\qquad\square$

### F.2 ANALYSIS IN STOCHASTIC LOCAL ORACLES SETTING

The convergence proof in the stochastic setting largely mirrors that of Theorem 1. Nevertheless, for the sake of completeness, we present it below.

To streamline the exposition, we slightly modify the notation: $v^k=\mathbb{E}\left\|\tilde{f}^k-h^k\right\|^2$, $w^k=\sum_{i=1}^{M}\mathbb{E}\left\|\tilde{f}_{i,\xi_i}^k-h_i^k\right\|^2$. Lemma 6 remains unchanged under the new notation. Lemma 5 undergoes only minor modifications in the proof and takes the following form.

**Lemma 7.** *Let Assumptions 1, 2 and 5 hold. Then for iterations of Algorithm 1 with unbiased compressor 1 $\mathcal{Q}$ and stochastic local gradients holds:*

$$w^k \le (1+c_2^{-1})\left[6\frac{aL^2}{M}\mathbb{E}\left\|\theta^k-\theta^{k-1}\right\|^2+2\tilde{L}^2\mathbb{E}\left\|\pi^k-\pi^{k-1}\right\|_1^2\right]$$
$$+(1+c_2)(1+\beta^2\omega-2\beta)w^{k-1}+\frac{4a^2}{M}\sigma^2. \tag{41}$$

*Proof.* We begin by using the explicit update rule of clients' local state.

$$w^k = \sum_{i=1}^{M}\mathbb{E}\|\tilde{f}_{i,\xi_i}^k-h_i^k\|^2=\sum_{i=1}^{M}\mathbb{E}\|\tilde{f}_{i,\xi_i}^k-h_i^{k-1}-\beta\mathcal{Q}(\tilde{f}_{i,\xi_i}^{k-1}-h_i^{k-1})\|^2$$

$$= \sum_{i=1}^{M} \mathbb{E} \left\| \left( \tilde{f}_{i,\xi_i}^k - \tilde{f}_{i,\xi_i}^{k-1} \right) + \left( \tilde{f}_{i,\xi_i}^{k-1} - h_i^{k-1} - \beta \mathcal{Q}(\tilde{f}_{i,\xi_i}^{k-1} - h_i^{k-1}) \right) \right\|^2$$

$$\overset{\text{(CS)}}{\leq} (1 + c_2^{-1}) \sum_{i=1}^{M} \mathbb{E} \left\| \tilde{f}_{i,\xi_i}^k - \tilde{f}_{i,\xi_i}^{k-1} \right\|^2$$

$$+ (1 + c_2) \sum_{i=1}^{M} \mathbb{E} \left\| \tilde{f}_{i,\xi_i}^{k-1} - h_i^{k-1} - \beta \mathcal{Q}(\tilde{f}_{i,\xi_i}^{k-1} - h_i^{k-1}) \right\|^2. \tag{42}$$

We now estimate the first term:

$$\sum_{i=1}^{M} \mathbb{E} \left\| \tilde{f}_{i,\xi_i}^k - \tilde{f}_{i,\xi_i}^{k-1} \right\|^2 = \sum_{i=1}^{M} \mathbb{E} \left\| \pi_i^k \nabla f_{i,\xi_i}(\theta^k) - \pi_i^{k-1} \nabla f_{i,\xi_i}(\theta^{k-1}) \right\|^2$$

$$= \sum_{i=1}^{M} \mathbb{E} \left\| \left( \pi_i^k \nabla f_{i,\xi_i}(\theta^k) - \pi_i^k \nabla f_i(\theta^k) \right) - \left( \pi_i^{k-1} \nabla f_{i,\xi_i}(\theta^{k-1}) - \pi_i^{k-1} \nabla f_i(\theta^{k-1}) \right) \right.$$

$$\left. + \left( \pi_i^k \nabla f_i(\theta^k) - \pi_i^{k-1} \nabla f_i(\theta^{k-1}) \right) \right\|^2$$

$$= \sum_{i=1}^{M} \mathbb{E} \left\| \left( \pi_i^k \nabla f_{i,\xi_i}(\theta^k) - \pi_i^k \nabla f_i(\theta^k) \right) - \left( \pi_i^{k-1} \nabla f_{i,\xi_i}(\theta^{k-1}) - \pi_i^{k-1} \nabla f_i(\theta^{k-1}) \right) \right\|^2$$

$$+ \sum_{i=1}^{M} \mathbb{E} \left\| \left( \pi_i^k \nabla f_i(\theta^k) - \pi_i^{k-1} \nabla f_i(\theta^{k-1}) \right) \right\|^2. \tag{43}$$

The second sum of (43) was evaluated in (10), (11) and (11). We proceed with estimating the first term of 43.

$$\sum_{i=1}^{M} \mathbb{E} \left\| \left( \pi_i^k \nabla f_{i,\xi_i}(\theta^k) - \pi_i^k \nabla f_i(\theta^k) \right) - \left( \pi_i^{k-1} \nabla f_{i,\xi_i}(\theta^{k-1}) - \pi_i^{k-1} \nabla f_i(\theta^{k-1}) \right) \right\|^2$$

$$\overset{\text{(CS)}}{\leq} 2 \sum_{i=1}^{M} \mathbb{E} \left\| \pi_i^k \nabla f_{i,\xi_i}(\theta^k) - \pi_i^k \nabla f_i(\theta^k) \right\|^2$$

$$+ 2 \sum_{i=1}^{M} \mathbb{E} \left\| \pi_i^{k-1} \nabla f_{i,\xi_i}(\theta^{k-1}) - \pi_i^{k-1} \nabla f_i(\theta^{k-1}) \right\|^2$$

$$\overset{5}{\leq} 2 \sum_{i=1}^{M} (\pi_i^{k^2} + \pi_i^{k-1^2}) \sigma^2 \leq \sum_{i=1}^{M} \frac{4a^2}{M^2} \sigma^2 = \frac{4a^2}{M} \sigma^2. \tag{44}$$

Substitution of (11) and (44) into (43) gives

$$\sum_{i=1}^{M} \mathbb{E} \left\| \tilde{f}_{i,\xi_i}^k - \tilde{f}_{i,\xi_i}^{k-1} \right\|^2 \leq 6 \frac{aL^2}{M} \mathbb{E} \left\| \theta^k - \theta^{k-1} \right\|^2 + 2\tilde{L}^2 \mathbb{E} \left\| \pi^k - \pi^{k-1} \right\|_1^2 + \frac{4a^2}{M} \sigma^2. \tag{45}$$

Then we evaluate the second term of (42) RHS:

$$\mathbb{E} \left\| \tilde{f}_{i,\xi_i}^{k-1} - h_i^{k-1} - \beta \mathcal{Q}(\tilde{f}_{i,\xi_i}^{k-1} - h_i^{k-1}) \right\|^2$$

$$= \mathbb{E} \left\| \tilde{f}_{i,\xi_i}^{k-1} - h_i^{k-1} \right\|^2 + \beta^2 \mathbb{E} \left\| \mathcal{Q}(\tilde{f}_{i,\xi_i}^{k-1} - h_i^{k-1}) \right\|^2$$

$$- 2 \mathbb{E} \left\langle \tilde{f}_{i,\xi_i}^{k-1} - h_i^{k-1}, \beta \mathcal{Q}(\tilde{f}_{i,\xi_i}^{k-1} - h_i^{k-1}) \right\rangle$$

$$\overset{1}{\leq} \mathbb{E} \left\| \tilde{f}_{i,\xi_i}^{k-1} - h_i^{k-1} \right\|^2 + \beta^2 \omega \mathbb{E} \left\| \tilde{f}_{i,\xi_i}^{k-1} - h_i^{k-1} \right\|^2 - 2\beta \mathbb{E} \| \tilde{f}_{i,\xi_i}^{k-1} - h_i^{k-1} \|^2$$

$$= (1 + \beta^2 \omega - 2\beta)\mathbb{E} \left\| \tilde{f}_{i,\xi_i}^{k-1} - h_i^{k-1} \right\|^2. \tag{46}$$

Finally, combining (42) with (45) and (46) we obtain

$$
\begin{aligned}
w^k &\leq (1 + c_2^{-1}) \left[ 6 \frac{aL^2}{M} \mathbb{E} \left\| \theta^k - \theta^{k-1} \right\|^2 + 2\tilde{L}^2 \mathbb{E} \left\| \pi^k - \pi^{k-1} \right\|_1^2 \right] \\
&\quad + (1 + c_2)(1 + \beta^2 \omega - 2\beta) w^{k-1} + \frac{4a^2}{M} \sigma^2.
\end{aligned}
$$

$\square$

Lemma 5 likewise undergoes a minor modifications.

**Lemma 8.** *Let Assumptions 1 and 2 hold. Then for iterations of Algorithm 1 with unbiased compressor 1 $\mathcal{Q}$ and stochastic local gradients holds:*

$$
\begin{aligned}
v^k &\leq \left(1 + c_1^{-1}\right) \left( 2L^2 \mathbb{E} \left\| \theta^k - \theta^{k-1} \right\|^2 + 2M\tilde{L}^2 \mathbb{E} \left\| \pi^k - \pi^{k-1} \right\|_1^2 \right) \\
&\quad + (1 + c_1) \left( v^{k-1}(1 + \beta^2 - 2\beta) + \beta^2 \omega w^{k-1} \right). \tag{47}
\end{aligned}
$$

*Proof.* We begin with the explicit global estimator update rule.

$$
\begin{aligned}
v^k &= \mathbb{E} \left\| \tilde{f}^k - h^k \right\|^2 = \mathbb{E} \left\| \tilde{f}^k - h^{k-1} - \beta \sum_{i=1}^M \mathcal{Q}(\tilde{f}_{i,\xi_i}^{k-1} - h_i^{k-1}) \right\|^2 \\
&= \mathbb{E} \left\| \left( \tilde{f}^k - \tilde{f}^{k-1} \right) + \left( \tilde{f}^{k-1} - h^{k-1} - \beta \sum_{i=1}^M \mathcal{Q}(\tilde{f}_{i,\xi_i}^{k-1} - h_i^{k-1}) \right) \right\|^2 \\
&\overset{(CS)}{\leq} \left(1 + c_1^{-1}\right) \mathbb{E} \left\| \tilde{f}^k - \tilde{f}^{k-1} \right\|^2 \\
&\quad + (1 + c_1) \mathbb{E} \left\| \tilde{f}^{k-1} - h^{k-1} - \beta \sum_{i=1}^M \mathcal{Q}(\tilde{f}_{i,\xi_i}^{k-1} - h_i^{k-1}) \right\|^2.
\end{aligned}
$$

For the first term on the (48) RHS (16) remains unchanged and we concentrate on the second term of the (48) RHS:

$$
\begin{aligned}
&\mathbb{E} \left\| \tilde{f}^{k-1} - h^{k-1} - \beta \sum_{i=1}^M \mathcal{Q}(\tilde{f}_{i,\xi_i}^{k-1} - h_i^{k-1}) \right\|^2 \\
&= \mathbb{E} \left\| \tilde{f}^{k-1} - h^{k-1} \right\|^2 + \beta^2 \mathbb{E} \left\| \sum_{i=1}^M \mathcal{Q}(\tilde{f}_{i,\xi_i}^{k-1} - h_i^{k-1}) \right\|^2 \\
&\quad - 2\beta \mathbb{E} \left\langle \tilde{f}^{k-1} - h^{k-1}, \sum_{i=1}^M \mathcal{Q}(\tilde{f}_{i,\xi_i}^{k-1} - h_i^{k-1}) \right\rangle \\
&\overset{(20)}{\underset{1}{\leq}} \mathbb{E} \left\| \tilde{f}^{k-1} - h^{k-1} \right\|^2 + \beta^2 \left( v^{k-1} + (\omega - 1)w^{k-1} \right) - 2\beta \mathbb{E} \left\| \tilde{f}^{k-1} - h^{k-1} \right\|^2 \\
&= v^{k-1}(1 + \beta^2 - 2\beta) + \beta^2 w^{k-1}(\omega - 1). \tag{48}
\end{aligned}
$$

Plugging (48) and (16) into (48) yields

$$
\begin{aligned}
v^k &\leq \left(1 + c_1^{-1}\right) \left( 2L^2 \mathbb{E} \left\| \theta^k - \theta^{k-1} \right\|^2 + 2M\tilde{L}^2 \mathbb{E} \left\| \pi^k - \pi^{k-1} \right\|_1^2 \right) \\
&\quad + (1 + c_1) \left( v^{k-1}(1 + \beta^2 - 2\beta) + \beta^2 \omega w^{k-1} \right).
\end{aligned}
$$

$\square$

We are now ready to proceed to the proof of Theorem 2, which is the main result for the stochastic local oracles case. The structure of the reasoning remains the same as in the proof of Theorem 1.

**Theorem 2** *Let in setting of Theorem 1 additionally Assumption 5 holds. Then, it implies*

$$\mathbb{E}\left[\mathbf{Gap}(\bar{z}_K)\right] \leq \frac{V}{2\gamma K} + \gamma\frac{64a^2\omega^2}{M}\sigma^2$$

*for iterations of Algorithm 1 with stochastic local oracles.*

*Proof.* For the sake of consistency with previous notation, we relabel the full weighted local gradient $\tilde{f}_i^k := \pi_i^k \nabla f_i(\theta^k)$ and its stochastic estimator $\tilde{f}_{i,\xi_i}^k = \pi_i^k \nabla f_{i,\xi_i}(\theta^k)$.

As in Theorem 1 we proceed with using the unbiasedness (2) of compressor $\mathcal{Q}$:

$$\mathbb{E}\left[F(z^k)|z^k\right] = \mathbb{E}\left[\left(\begin{array}{c}g^k\\p^k\end{array}\right)\Big|z^k\right] = \mathbb{E}\left[\left(\begin{array}{c}h^k + \sum_{i=1}^M \mathcal{Q}(\tilde{f}_{i,\xi_i}^k - h_i^k)\\p^k\end{array}\right)\Big|z^k\right] = \left(\begin{array}{c}\tilde{f}^k\\p^k\end{array}\right) \stackrel{\text{def}}{=} \overline{F}(z^k),$$

where $\tilde{f}^k = \sum_{i=1}^M \tilde{f}_i^k = \sum_{i=1}^M \pi_i^k \nabla f_i(\theta^k)$. And Lemma 1 again justifies the monotonicity of $\overline{F}$.

The next step is conditioning the result of Lemma 3 on $z^{k+1}$, using $\alpha = 1$ and summing over $k = 1$ to $K$:

$$2\gamma\sum_{k=1}^K\langle\overline{F}(z^{k+1}), z^{k+1}-z\rangle \leq \sum_{k=1}^K\Big[\left(2D_{KL}(\pi, \pi^k) - 2D_{KL}(\pi, \pi^{k+1})\right) + \left(\|\theta^k-\theta\|^2 - \|\theta^{k+1}-\theta\|^2\right)$$
$$+ \left(2\gamma\langle F(z^k) - F(z^{k-1}), z-z^k\rangle - 2\gamma\langle\overline{F}(z^{k+1}) - F(z^k), z-z^{k+1}\rangle\right)$$
$$- \frac{1}{2}\|\pi^{k+1}-\pi^k\|_1^2 - \frac{1}{2}\|\theta^{k+1}-\theta^k\|^2$$
$$+ 2\gamma^2\|p^k-p^{k-1}\|^2 + 2\gamma^2\|g^k-g^{k-1}\|^2\Big]$$
$$= \left(2D_{KL}(\pi, \pi^1) - 2D_{KL}(\pi, \pi^{K+1})\right) + \left(\|\theta^1-\theta\|^2 - \|\theta^{K+1}-\theta\|^2\right)$$
$$+ \left(2\gamma\langle F(z^1) - F(z^0), z-z^1\rangle - 2\gamma\langle\overline{F}(z^{K+1}) - F(z^K), z-z^{K+1}\rangle\right)$$
$$+ \sum_{k=1}^{K-1}\Big[2\gamma\langle\overline{F}(z^{k+1}) - F(z^{k+1}), z-z^{k+1}\rangle\Big]$$
$$+ \sum_{k=1}^K\Big[-\frac{1}{2}\|\pi^{k+1}-\pi^k\|_1^2 - \frac{1}{2}\|\theta^{k+1}-\theta^k\|^2$$
$$+ 2\gamma^2\|p^k-p^{k-1}\|^2 + 2\gamma^2\|g^k-g^{k-1}\|^2\Big].$$

Maximizing obtained inequality over compact set $z \in \mathcal{D}$ and taking full expectation, we get

$$2\gamma\,\mathbb{E}\left[\max_{z\in\mathcal{D}}\Big\{\sum_{k=1}^K\langle\overline{F}(z^{k+1}), z^{k+1}-z\rangle\Big\}\right] \leq \mathbb{E}\Big[\max_{z\in\mathcal{D}}\Big\{\left(2D_{KL}(\pi, \pi^1) - 2D_{KL}(\pi, \pi^{K+1})\right)$$
$$+ \left(\|\theta^1-\theta\|^2 - \|\theta^{K+1}-\theta\|^2\right)$$
$$+ \left(2\gamma\langle F(z^1) - F(z^0), z-z^1\rangle\right.$$
$$\left. -2\gamma\langle\overline{F}(z^{K+1}) - F(z^K), z-z^{K+1}\rangle\right)$$
$$+ \sum_{k=1}^{K-1}2\gamma\langle\overline{F}(z^{k+1}) - F(z^{k+1}), z-z^{k+1}\rangle\Big\}$$
$$+ \sum_{k=1}^K\Big[-\frac{1}{2}\|\pi^{k+1}-\pi^k\|_1^2 - \frac{1}{2}\|\theta^{k+1}-\theta^k\|^2$$
$$+ 2\gamma^2\|p^k-p^{k-1}\|^2 + 2\gamma^2\|g^k-g^{k-1}\|^2\Big]\Big]. \quad (49)$$

Several next steps evaluate different terms of (49). Inequalities (22) and (23) remain valid. We continue with evaluating of the last term applying properties (2) of unbiased compressor $\mathcal{Q}$ and its

independence from stochastic local oracles:

$$
\begin{aligned}
\mathbb{E}\left\|\overline{F}(z^k) - F(z^k)\right\|^2 &= \mathbb{E}\left\|\begin{pmatrix}\tilde{f}^k \\ p^k\end{pmatrix} - \begin{pmatrix}h^k + \sum_{i=1}^{M}\mathcal{Q}(\tilde{f}_{i,\xi_i}^k - h_i^k) \\ p^k\end{pmatrix}\right\|^2 \\
&= \mathbb{E}\left\|\sum_{i=1}^{M}\mathcal{Q}(\tilde{f}_{i,\xi_i}^k - h_i^k) - (\tilde{f}^k - h^k)\right\|^2 \\
&= \sum_{i=1}^{M}\mathbb{E}\left\|\mathcal{Q}(\tilde{f}_{i,\xi_i}^k - h_i^k)\right\|^2 + \mathbb{E}\left\|\tilde{f}^k - h^k\right\|^2 \\
&\quad -2\sum_{i=1}^{M}\mathbb{E}\left\langle\mathcal{Q}(\tilde{f}_{i,\xi_i}^k - h_i^k), \tilde{f}^k - h^k\right\rangle \\
&\leq \omega\sum_{i=1}^{M}\mathbb{E}\left\|\tilde{f}_{i,\xi_i}^k - h_i^k\right\|^2 - \mathbb{E}\left\|\tilde{f}^k - h^k\right\|^2.
\end{aligned}
\tag{50}
$$

Finally, using (Pi), definition of $z = \begin{pmatrix}\theta \\ \pi\end{pmatrix}$ and choosing $z_a = z^1 - z^{k+1}$ we estimate

$$
\begin{aligned}
\max_{z\in\mathcal{D}}\left\{2D_{KL}(\pi,\pi^1) + \|\theta^1 - \theta\|^2\right\} &\geq \max_{z\in\mathcal{D}}\left\{\|\pi^1 - \pi\|^2 + \|\theta^1 - \theta\|^2\right\} \\
&= \max_{z\in\mathcal{D}}\left\{\|z_a - z\|^2\right\}.
\end{aligned}
\tag{51}
$$

Slightly changing old notation $v^k = \mathbb{E}\left\|\tilde{f}^k - h^k\right\|^2$, $w^k = \sum_{i=1}^{M}\mathbb{E}\left\|\tilde{f}_{i,\xi_i}^k - h_i^k\right\|^2$ and combining (23),(50) with (51) we derive

$$
\begin{aligned}
2\mathbb{E}&\left[\max_{z\in\mathcal{D}}\left\{\sum_{k=1}^{K-1}\left\langle\gamma\left(\overline{F}(z^{k+1}) - F(z^{k+1})\right), z - z^{k+1}\right\rangle\right\}\right] \\
&\leq \max_{z\in\mathcal{D}}\left\{2D_{KL}(\pi,\pi^1) + \|\theta^1 - \theta\|^2\right\} + \gamma^2\sum_{k=2}^{K}\left(\omega w^k - v^k\right).
\end{aligned}
\tag{52}
$$

After that, we estimate $2\gamma^2\|p^k - p^{k-1}\|^2$ in (49) via Assumption 1:
$$
2\gamma^2\|p^k - p^{k-1}\|^2 \leq 2\gamma^2 M\tilde{L}^2\|\theta^k - \theta^{k-1}\|^2.
\tag{53}
$$

Finally, we use (18) to evaluate the sum:

$$
\begin{aligned}
2\gamma^2&\sum_{k=1}^{K}\mathbb{E}\|g^k - g^{k-1}\|^2 \\
&= 4\gamma^2\sum_{k=1}^{K}\left[\omega w^k + v^k\right] + 4\gamma^2(1-\beta)^2\sum_{k=1}^{K}\left[\omega w^{k-1} + v^{k-1}\right] \\
&= 4\gamma^2\sum_{k=1}^{K}\left[\omega w^k + v^k\right] + 4\gamma^2(1-\beta)^2\sum_{k=0}^{K-1}\left[\omega w^k + v^k\right] \\
&= 4\gamma^2(1 + (1-\beta)^2)\left[\sum_{k=0}^{K}\omega w^k + v^k\right] \\
&\quad -4\gamma^2(1-\beta)[\omega w^K + v^K] \\
&\quad -4\gamma^2[\omega w^0 + v^0].
\end{aligned}
\tag{54}
$$

Substituting (54), (53), (52) and (22) into (49),

$$
2\gamma K\mathbb{E}\left[\mathbf{Gap}(\overline{z}_K)\right] \leq \mathbb{E}\left[\max_{z\in\mathcal{D}}\left\{\left(4D_{KL}(\pi,\pi^1) - 2D_{KL}(\pi,\pi^{K+1})\right)\right.\right.
$$

$$+ \left(2\|\theta^1 - \theta\|^2 - \|\theta^{K+1} - \theta\|^2\right)$$

$$+ \left(2\gamma\langle F(z^1) - F(z^0), z - z^1\rangle - 2\gamma\langle \overline{F}(z^{K+1}) - F(z^K), z - z^{K+1}\rangle\right)\Big\}$$

$$- 4\gamma^2(1-\beta)[\omega w^K + v^K] - 4\gamma^2[\omega w^0 + v^0]$$

$$+ \gamma^2 \sum_{k=2}^{K} \left(\omega w^k - v^k\right) + 4\gamma^2(1 + (1-\beta)^2) \sum_{k=0}^{K} \left[\omega w^k + v^k\right]$$

$$+ \sum_{k=1}^{K} \left[ -\frac{1}{2}\|\pi^{k+1} - \pi^k\|_1^2 - \left(\frac{1}{2} - 2\gamma^2 M\tilde{L}^2\right)\|\theta^{k+1} - \theta^k\|^2\right]\right].$$

Using (50) as $\omega w^k - v^k \geq 0$ and $0 < \beta < 1$, and recalling

$$\begin{aligned}
\Xi_K &= \max_{z \in \mathcal{D}} \Big\{ \left(4D_{KL}(\pi, \pi^1) - 2D_{KL}(\pi, \pi^{K+1})\right) \\
&\quad + \left(2\|\theta^1 - \theta\|^2 - \|\theta^{K+1} - \theta\|^2\right) \\
&\quad + \left(2\gamma\langle F(z^1) - F(z^0), z - z^1\rangle \\
&\quad - 2\gamma\langle \overline{F}(z^{K+1}) - F(z^K), z - z^{K+1}\rangle\right)\Big\} \\
&\quad - 4\gamma^2(1-\beta)[\omega w^K + v^K] - 4\gamma^2[\omega w^0 + v^0],
\end{aligned} \tag{55}$$

we can rewrite:

$$\begin{aligned}
2\gamma K \mathbb{E}\left[\mathbf{Gap}(\overline{z}_K)\right] &\leq \mathbb{E}\Big[\Xi_K + \sum_{k=0}^{K}\left[9\omega\gamma^2 w^k + 7\gamma^2 v^k\right] \\
&\quad + \sum_{k=1}^{K}\left[-\frac{1}{2}\|\pi^{k+1} - \pi^k\|_1^2 - \left(\frac{1}{2} - 2\gamma^2 M\tilde{L}^2\right)\|\theta^{k+1} - \theta^k\|^2\right]\Big].
\end{aligned} \tag{56}$$

The next step is summing (56) $+ \sum_{k=2}^{K+1}\left[H \cdot (41) + N \cdot (47)\right]$:

$$\begin{aligned}
&2\gamma K \mathbb{E}\left[\mathbf{Gap}(\overline{z}_K)\right] + \sum_{k=2}^{K+1}\left(Hw^k + Nv^k\right) \\
&\leq \mathbb{E}\Big[\Xi_K + \sum_{k=0}^{K}\left[9\omega\gamma^2 w^k + 7\gamma^2 v^k\right] + \sum_{k=1}^{K}\left[H(1+c_2)(1+\beta^2\omega - 2\beta)w^k\right] \\
&\quad + \sum_{k=1}^{K}\left[N(1+c_1)\left(v^k(1+\beta^2 - 2\beta) + \beta^2\omega w^k\right)\right] \\
&\quad + \sum_{k=1}^{K}\left[-\left(\frac{1}{2} - 2H\tilde{L}^2(1+c_2^{-1}) - 2NM\tilde{L}^2(1+c_1^{-1})\right)\|\pi^{k+1} - \pi^k\|_1^2 \right. \\
&\quad \left. - \left(\frac{1}{2} - 2\gamma^2 M\tilde{L}^2 - H(1+c_2^{-1})\frac{6aL^2}{M} - N\left(1+c_1^{-1}\right)2L^2\right)\|\theta^{k+1} - \theta^k\|^2\right] \Big] \\
&\quad + \frac{4KHa^2}{M}\sigma^2.
\end{aligned}$$

The subsequent analysis is unaffected by the additive term $\frac{4KHa^2}{M}\sigma^2$ introduced in the stochastic setting and fully mirrors the reasoning of Theorem 1 starting from Equation (31). $\qquad\square$

Corollary 3 provides bounds on number of communication rounds and transmitted from nodes to the server information in stochastic local oracles setup.

**Corollary 3** *In setting of Theorem 2 with* $\gamma = \min\left\{\gamma_0, \sqrt{\frac{VM}{128a^2\omega^2\sigma^2 K}}\right\}$, *Algorithm 1 with stochastic local oracles needs*

$$\mathcal{O}\left(\frac{1}{\varepsilon^2}\left[\frac{a^2\omega^2\sigma^2}{M}\right] + \frac{1}{\varepsilon}\left[\tilde{L}\omega^{3/2} + \tilde{L}M^{1/2}\omega + L\left(\sqrt{\frac{a\omega^3}{M}} + \omega\right)\right]\right)$$

*iterations in order to reach* $\varepsilon$-*accuracy with respect to* $\mathbb{E}\left[\mathbf{Gap}(\overline{z}_K)\right]$. *Additionally, it requires*

$$\mathcal{O}\left(\frac{1}{\varepsilon^2}\left[\frac{a^2\omega\sigma^2}{M}\right] + \frac{1}{\varepsilon}\left[\tilde{L}\omega^{1/2} + \tilde{L}M^{1/2} + L\left(\sqrt{\frac{a\omega}{M}} + 1\right)\right]\right)$$

*bits communicated from nodes to the server.*

*Proof.* In stochastic local oracles case, guaranties in the Theorem 2 are affected by an additional irreducible term $\gamma\frac{64a^2\omega^2}{M}\sigma^2$. In its presence, optimal stepsize $\gamma$ transforms into $\gamma = \min\left\{\gamma_0, \sqrt{\frac{VM}{128a^2\omega^2\sigma^2 K}}\right\}$. This choice yields $\frac{V}{2\gamma K} \geq \gamma\frac{64a^2\omega^2}{M}\sigma^2$ and makes further analysis similar to the proof of Corollary 1. $\square$

## F.3 Analysis in partial participation setting

We reduce this case to analysis in exact local oracles settings in Section F.1 by claiming that multiplying the compression operator by the $\frac{\eta}{p}$, with $\eta \sim \mathrm{Bern}(p)$, yields another valid compression operator. It remains unbiased, while its compression rate $\omega$ is scaled by a factor of $p$.

**Corollary 4** *In setting of Theorem 1 with* $\beta = \frac{p}{\omega}$, $H = 32\gamma^2\left(\frac{\omega}{p}\right)^2$, $N = 7\gamma^2\frac{\omega}{p}$, $\gamma_\pi = \gamma_\theta = \gamma \leq$

$$\gamma_p = \min\left\{\frac{1}{2\tilde{L}}\sqrt{\frac{1}{96\left(\frac{\omega}{p}\right)^3 + 14M\left(\frac{\omega}{p}\right)^2}}, \sqrt{\frac{1}{2}\frac{1}{4M\tilde{L}^2 + 576\frac{a}{M}\left(\frac{\omega}{p}\right)^3 L^2 + 28\left(\frac{\omega}{p}\right)^2 L^2}}\right\}$$ *it implies*

$$\mathbb{E}\left[\mathbf{Gap}(\overline{z}_K)\right] \leq \frac{V}{2\gamma K}$$

*for iterations of Algorithm 1 with partial participation.*

*Proof.* We consider $\mathcal{Q}' = \frac{\eta}{p}\mathcal{Q}$ and utilize independence of $\eta$ and $\mathcal{Q}$ to write

$$\mathbb{E}\left\|\frac{\eta}{p}\mathcal{Q}(z)\right\|^2 = \mathbb{E}\left(\frac{\eta}{p}\right)^2\|\mathcal{Q}(z)\|^2 = \mathbb{E}\left(\frac{\eta}{p}\right)^2\mathbb{E}\|\mathcal{Q}(z)\|^2 \overset{(2)}{\leq} \frac{1}{p}\omega\|z\|^2. \tag{57}$$

Independence along with (2) guaranties the unbiasedness of $\mathcal{Q}'$ as well:

$$\mathbb{E}\frac{\eta}{p}\mathcal{Q}(z) = \mathbb{E}\frac{\eta}{p}\mathbb{E}\mathcal{Q}(z) \overset{(2)}{=} z. \tag{58}$$

The established properties implies that operator $\mathcal{Q}'$ is unbiased compressor with compression rate $\omega' = \frac{\omega}{p}$. This enables the application of Theorem 1 and finishing the proof. $\square$

Given Corollary 4 in partial participation setting we establish the following bounds.

**Corollary 5** *In setting of Corollary 4 with* $\gamma = \gamma_p$, *Algorithm 1 with partial participation needs*

$$\mathcal{O}\left(\frac{1}{\varepsilon}\left[\tilde{L}\left(\frac{\omega}{p}\right)^{3/2} + \tilde{L}M^{1/2}\frac{\omega}{p} + L\left(\sqrt{\frac{a\omega^3}{Mp^3}} + \frac{\omega}{p}\right)\right]\right)$$

*iterations in order to reach* $\varepsilon$-*accuracy with respect to* $\mathbb{E}\left[\mathbf{Gap}(\overline{z}_K)\right]$. *Additionally, it requires*

$$\mathcal{O}\left(\frac{1}{\varepsilon}\left[\tilde{L}\left(\frac{\omega}{p}\right)^{1/2} + \tilde{L}M^{1/2} + L\left(\sqrt{\frac{a\omega}{Mp}} + 1\right)\right]\right)$$

*bits communicated from nodes to the server.*

*Proof.* Proof in this setting completely coincide with the proof of Corollary 1. $\square$

## G ANALYSIS IN NON-CONVEX SETUP

In this section we conduct the convergence analysis under relaxed convexity Assumption 3. Introduced further Assumption 4 is inspired by the *minty* assumption (i.e. existence of such $z^* \in \mathbb{R}^d$ that $\langle F(z), z - z^* \rangle \geq 0$ for all $z \in \mathbb{R}^d$), traditionally associated with non-monotonicity in respective literature Dang & Lan (2015); Mertikopoulos et al. (2018); Kannan & Shanbhag (2019).

**Assumption 4.** *Let there exists a point $\theta^* \in \mathbb{R}^d$ such that:*

$$\left\langle \sum_{i=1}^{M} \pi_i \nabla f_i(\theta), \theta - \theta^* \right\rangle \geq \sum_{i=1}^{M} \pi_i f_i(\theta) - \sum_{i=1}^{M} \pi_i f_i(\theta^*), \text{ for all } \theta \in \mathbb{R}^d, \pi \in \Delta^{M-1}.$$

We note that in our setting due to the linearity of objective $\sum_{i=1}^{M} \pi_i f_i(\theta)$ with respect to weights $\pi$ transition to the minty assumption is complicated. Instead of it we use Lemma 9.

**Lemma 9.** *Let Assumption 4 holds, then for operator $\overline{F}(z) = \overline{F}(\theta, \pi) = \left( \sum_{i=1}^{M} \pi_i \nabla f_i(\theta), p \right)^\top$, the following holds:*

$$\langle \overline{F}(z) - \overline{F}(z'^*), z - z'^* \rangle \geq 0, \text{ for all } \theta \in \mathbb{R}^d \text{ and } \pi, \pi' \in \Delta^{M-1}, \tag{59}$$

*where $z'^* = (\theta^*, \pi')^\top, z = (\theta, \pi)^\top$.*

*Proof.* We explicitly expand the expression $\langle F(z) - F(z'^*), z - z'^* \rangle$.

$$
\begin{aligned}
\langle F(z) - F(z'^*), z - z'^* \rangle &= \left\langle \sum_{i=1}^{M} \pi_i \nabla f_i(\theta), \theta - \theta^* \right\rangle - \sum_{i=1}^{M} (f_i(\theta) - f_i(\theta^*))(\pi_i - \pi_i') \\
&\overset{(4)}{\geq} \sum_{i=1}^{M} f_i(\theta) \pi_i - \sum_{i=1}^{M} f_i(\theta^*) \pi_i - \sum_{i=1}^{M} (f_i(\theta) - f_i(\theta^*))(\pi_i - \pi_i') \\
&= \sum_{i=1}^{M} (f_i(\theta) - f_i(\theta^*)) \pi_i' \geq 0
\end{aligned}
$$

$\square$

Lemma 9 allows to pass to the analysis with non-convex function $f(\theta)$ and consequently a non-monotone operator $\overline{F}$. Moreover, it enables to take into account naturally convex structure of the objective function $\sum_{i=1}^{M} \pi_i f_i(\theta)$ with respect to the weights $\pi$, which is also reflected in the convergence criterion presented in (5) and written below. In (5) we recognize the part of the **Gap** operator corresponding to $\pi$, while with respect to the parameter $\theta$ the criterion involves an averaged gradient norm.

$$
\begin{aligned}
W^K &= \mathbb{E} \max_{\pi' \in \Lambda} \left[ \sum_{i=1}^{M} f_i(\theta^*)(\overline{\pi}_i^K - \pi_i') + \left\langle \sum_{i=1}^{M} \pi_i' \nabla f_i(\theta^*), \overline{\theta}^K - \theta^* \right\rangle \right] \\
&\quad + \frac{1}{8\gamma K} \sum_{k=1}^{K} \mathbb{E} \| \pi^{k+1} - \pi^k \|^2 + \frac{\gamma}{32} \mathbb{E} \left\| \overline{\sum_{i=1}^{M} \pi_i \nabla f_i(\theta)}^K \right\|^2,
\end{aligned}
$$

Where $\overline{\pi}^K = \sum_{k=1}^{K} \frac{1}{K} \pi^{k+1}, \overline{\theta}^K = \sum_{k=1}^{K} \frac{1}{K} \pi^{k+1}$ and $\overline{\pi_i \nabla f_i(\theta)}^K \sim U \left[ \pi_i^1 \nabla f_i(\theta^1), \dots, \pi_i^K \nabla f_i(\theta^K) \right]$.

Theorem 3 provides convergence guaranties with respect to $W^K$ under relaxed convexity Assumption 4.

**Theorem 3.** *Let Assumptions 1, 2, 4 hold and $\alpha = 1$, $\beta = \frac{1}{\omega}$, $\gamma_\pi = \gamma_\theta = \gamma \leq \gamma_1 = \min \left\{ \tilde{L}^{-1} \left( 48(17\omega^3 - 2M\omega^2) \right)^{-\frac{1}{2}}, \left( 2448 \frac{a}{M} \omega^3 L^2 + 96\omega^2 L^2 + 16M\tilde{L}^2 \right)^{-\frac{1}{2}} \right\}$, $\Lambda = \Delta^{M-1} \cap Q_a^M$, where $Q_a^M = \{ x \in \mathbb{R}^M | 0 \leq x_i \leq \frac{a}{M} \}$. Then, after $K$ iterations of Algorithm 1 with*

*unbiased compressor 1 Q and exact local gradients solving problem* (4) *the following holds:*

$$W^K \leq \mathbb{E}\frac{1}{2\gamma K}\left[\max_{\pi'\in\Lambda}\left(2D_{KL}(\pi',\pi^1) + 2\gamma\langle\overline{F}(z^1) - F(z^0), z'^* - z^1\rangle\right) + \|\theta^1 - \theta^*\|^2\right.$$
$$\left.+ \sum_{k=0}^{1}\left[30\gamma^2\omega\left\|\tilde{f}^k - h^k\right\|^2 + 85\gamma^2\omega^2\sum_{i=1}^{M}\mathbb{E}\left\|\tilde{f}_i^k - h_i^k\right\|^2\right]\right].$$

*Proof.* We begin with the result of Lemma 3.

$$
\begin{aligned}
2\gamma\langle F(z^{k+1}), z^{k+1} - z\rangle \leq\ & \left(2D_{KL}(\pi,\pi^k) - 2D_{KL}(\pi,\pi^{k+1})\right) \\
& + \left(\|\theta^k - \theta\|^2 - \|\theta^{k+1} - \theta\|^2\right) \\
& + \left(2\gamma\langle F(z^k) - F(z^{k-1}), z - z^k\rangle\right. \\
& \left. -2\gamma\langle F(z^{k+1}) - F(z^k), z - z^{k+1}\rangle\right) \\
& - \frac{1}{2}\|\pi^{k+1} - \pi^k\|^2 - \frac{1}{2}\|\theta^{k+1} - \theta^k\|^2 \\
& + 2\gamma^2\|p^k - p^{k-1}\|^2 + 2\gamma^2\|g^k - g^{k-1}\|^2.
\end{aligned}
$$

Proceeding with conditioning on $z^{k+1}$.

$$
\begin{aligned}
&2\gamma\langle\overline{F}(z^{k+1}), z^{k+1} - z\rangle \\
&\leq \left(2D_{KL}(\pi,\pi^k) - 2D_{KL}(\pi,\pi^{k+1})\right) + \left(\|\theta^k - \theta\|^2 - \|\theta^{k+1} - \theta\|^2\right) \\
&\quad + \left(2\gamma\langle F(z^k) - F(z^{k-1}), z - z^k\rangle - 2\gamma\langle\overline{F}(z^{k+1}) - F(z^k), z - z^{k+1}\rangle\right) \\
&\quad - \frac{1}{2}\|\pi^{k+1} - \pi^k\|^2 - \frac{1}{2}\|\theta^{k+1} - \theta^k\|^2 \\
&\quad + 2\gamma^2\|p^k - p^{k-1}\|^2 + 2\gamma^2\|g^k - g^{k-1}\|^2.
\end{aligned}
\tag{60}
$$

We choose $z = z'^*$ in order to apply Lemma 9.

$$
\begin{aligned}
\langle\overline{F}(z^{k+1}), z^{k+1} - z\rangle &\geq \langle\overline{F}(z'^*), z^{k+1} - z'^*\rangle \\
&= \langle p^*, \pi^{k+1} - \pi'\rangle + \langle\sum_{i=1}^{M}\pi_i'\nabla f_i(\theta^*), \theta^{k+1} - \theta^*\rangle,
\end{aligned}
\tag{61}
$$

Where $p^* = (f_i(\theta^*))_{i=1}^{M}$. Then we substitute (61) into (60) and summing over $k = 1$ to $K$.

$$
\begin{aligned}
&2\gamma K\left\langle p^*, \sum_{k=1}^{K}\frac{1}{K}\pi^{k+1} - \pi'\right\rangle + 2\gamma K\left\langle\sum_{i=1}^{M}\pi_i'\nabla f_i(\theta^*), \sum_{k=1}^{K}\frac{1}{K}\theta^{k+1} - \theta^*\right\rangle \\
&\leq \left(2D_{KL}(\pi',\pi^1) - 2D_{KL}(\pi',\pi^{K+1})\right) + \left(\|\theta^1 - \theta^*\|^2 - \|\theta^{K+1} - \theta^*\|^2\right) \\
&\quad + \sum_{k=1}^{K}\left[\left(2\gamma\langle F(z^k) - F(z^{k-1}), z'^* - z^k\rangle - 2\gamma\langle\overline{F}(z^{k+1}) - F(z^k), z'^* - z^{k+1}\rangle\right)\right. \\
&\quad - \frac{1}{2}\|\pi^{k+1} - \pi^k\|^2 - \frac{1}{2}\|\theta^{k+1} - \theta^k\|^2 \\
&\quad \left.+ 2\gamma^2\|p^k - p^{k-1}\|^2 + 2\gamma^2\|g^k - g^{k-1}\|^2\right].
\end{aligned}
\tag{62}
$$

At the next step we evaluate $\|\theta^{k+1} - \theta^k\|^2$ to extract the term for convergence criterion. We utilize (CS) inequality several times.

$$\|\theta^{k+1} - \theta^k\|^2 \overset{16}{=} \gamma^2 \|2g^k - g^{k-1}\|^2$$
$$\overset{\text{(CS)}}{\geq} \gamma^2 \left[ \frac{1}{2}\|g^k\|^2 - \|g^k - g^{k-1}\|^2 \right]$$
$$\overset{\text{(CS)}}{\geq} \gamma^2 \left[ \frac{1}{4}\|\tilde{f}^k\|^2 - \frac{1}{2}\|\tilde{f}^k - g^k\|^2 - \|g^k - g^{k-1}\|^2 \right] \tag{63}$$

After that we splitting the $\frac{1}{2}\|\theta^{k+1} - \theta^k\|$ in (62) into 2 terms with factors $\frac{1}{4}$ and substitute (63) into the one of them.

$$2\gamma K \left\langle p^*, \sum_{k=1}^{K} \frac{1}{K}\pi^{k+1} - \pi' \right\rangle + 2\gamma K \left\langle \sum_{i=1}^{M} \pi_i' \nabla f_i(\theta^*), \sum_{k=1}^{K} \frac{1}{K}\theta^{k+1} - \theta^* \right\rangle$$
$$+ \sum_{k=1}^{K} \left[ \frac{1}{4}\|\pi^{k+1} - \pi^k\|^2 + \frac{\gamma^2}{16}\|\tilde{f}^k\|^2 \right]$$
$$\leq \left( 2D_{KL}(\pi', \pi^1) - 2D_{KL}(\pi', \pi^{K+1}) \right) + \left( \|\theta^1 - \theta^*\|^2 - \|\theta^{K+1} - \theta^*\|^2 \right)$$
$$+ \sum_{k=1}^{K} \left[ \left( 2\gamma \langle F(z^k) - F(z^{k-1}), z'^* - z^k \rangle - 2\gamma \langle \overline{F}(z^{k+1}) - F(z^k), z'^* - z^{k+1} \rangle \right) \right.$$
$$- \frac{1}{4}\|\pi^{k+1} - \pi^k\|^2 - \frac{1}{4}\|\theta^{k+1} - \theta^k\|^2$$
$$+ \frac{\gamma^2}{8}\|\tilde{f}^k - g^k\|^2 + \frac{\gamma^2}{4}\|g^k - g^{k-1}\|^2$$
$$\left. + 2\gamma^2\|p^k - p^{k-1}\|^2 + 2\gamma^2\|g^k - g^{k-1}\|^2 \right]. \tag{64}$$

Then we maximize (64) over $\pi' \in \Lambda$ and take full expectation. Additionally we note, that

$$\mathbb{E}\left[ \sum_{k=1}^{K} \max_{\pi' \in \Lambda} \langle F(z^k) - F(z^{k-1}), z'^* - z^k \rangle \right]$$
$$= \mathbb{E}\left[ \sum_{k=1}^{K} \langle g^k - g^{k-1}, \theta^* - \theta^k \rangle + \max_{\pi' \in \Lambda} \sum_{k=1}^{K} \langle p^k - p^{k-1}, \pi' - \pi^k \rangle \right]$$
$$= \mathbb{E}\left[ \sum_{k=1}^{K} \mathbb{E}\left[ \langle g^k - g^{k-1}, \theta^* - \theta^k \rangle | z^k \right] + \max_{\pi' \in \Lambda} \sum_{k=1}^{K} \langle p^k - p^{k-1}, \pi' - \pi^k \rangle \right]$$
$$= \mathbb{E}\left[ \sum_{k=1}^{K} \langle \tilde{f}^k - g^{k-1}, \theta^* - \theta^k \rangle + \max_{\pi' \in \Lambda} \sum_{k=1}^{K} \langle p^k - p^{k-1}, \pi' - \pi^k \rangle \right],$$

which enable us to recover the telescopic structure of inner products sum.

$$\mathbb{E}\max_{\pi' \in \Lambda} \left[ 2\gamma K \left\langle p^*, \sum_{k=1}^{K} \frac{1}{K}\pi^{k+1} - \pi' \right\rangle + 2\gamma K \left\langle \sum_{i=1}^{M} \pi_i' \nabla f_i(\theta^*), \sum_{k=1}^{K} \frac{1}{K}\theta^{k+1} - \theta^* \right\rangle \right.$$
$$\left. + \sum_{k=1}^{K} \left[ \frac{1}{4}\|\pi^{k+1} - \pi^k\|^2 + \frac{\gamma^2}{16}\|\tilde{f}^k\|^2 \right] \right]$$
$$\leq \mathbb{E}\left[ \max_{\pi' \in \Lambda} \left( 2D_{KL}(\pi', \pi^1) - 2D_{KL}(\pi', \pi^{K+1}) \right) + \left( \|\theta^1 - \theta^*\|^2 - \|\theta^{K+1} - \theta^*\|^2 \right) \right.$$
$$\left. + \max_{\pi' \in \Lambda} \left( 2\gamma \langle \overline{F}(z^1) - F(z^0), z'^* - z^1 \rangle - 2\gamma \langle \overline{F}(z^{K+1}) - F(z^K), z'^* - z^{K+1} \rangle \right) \right.$$

$$+ \sum_{k=1}^{K} \left[ -\frac{1}{4}\|\pi^{k+1} - \pi^k\|^2 - \frac{1}{4}\|\theta^{k+1} - \theta^k\|^2 \right.$$

$$\left. + \frac{\gamma^2}{8}v^k + \frac{9\gamma^2}{4}\|g^k - g^{k-1}\|^2 + 2\gamma^2\|p^k - p^{k-1}\|^2 \right]. \tag{65}$$

We proceed by summing $(65) + \sum_{k=1}^{K}(27) + \sum_{k=1}^{K}\frac{9}{8}(18) + \sum_{k=2}^{K+1}N(14) + \sum_{k=2}^{K+1}H(8) + (40)$:

$$\mathbb{E}\max_{\pi' \in \Lambda}\left[ 2\gamma K\left\langle p^*, \sum_{k=1}^{K}\frac{1}{K}\pi^{k+1} - \pi'\right\rangle + 2\gamma K\left\langle \sum_{i=1}^{M}\pi_i'\nabla f_i(\theta^*), \sum_{k=1}^{K}\frac{1}{K}\theta^{k+1} - \theta^*\right\rangle \right.$$

$$\left. + \sum_{k=1}^{K}\left[\frac{1}{4}\|\pi^{k+1} - \pi^k\|^2 + \frac{\gamma^2}{16}\|\tilde{f}^k\|^2\right]\right] + \sum_{k=1}^{K-1}Nv^{k+1} + \sum_{k=1}^{K-1}Hw^{k+1}$$

$$\leq \mathbb{E}\left[ \max_{\pi' \in \Lambda}\left( 2D_{KL}(\pi', \pi^1) + 2\gamma\langle \overline{F}(z^1) - F(z^0), z'^* - z^1\rangle\right) + \|\theta^1 - \theta^*\|^2 \right.$$

$$+ \sum_{k=1}^{K}\left[ \left( 2H(1 + c_2^{-1})\tilde{L}^2 + 2N(1 + c_1^{-1})M\tilde{L}^2 - \frac{1}{4}\right)\|\pi^{k+1} - \pi^k\|^2 \right.$$

$$+ \left( 6H(1 + c_2^{-1})\frac{aL^2}{M} + 2N(1 + c_1^{-1})L^2 + 4\gamma^2 M\tilde{L}^2 - \frac{1}{4}\right)\|\theta^{k+1} - \theta^k\|^2$$

$$+ \left( \frac{\gamma^2}{8} + \frac{9\gamma^2}{4}N(1 + c_1)(1 + \beta^2 - 2\beta)\right)v^k + \frac{9(1 - \beta^2)\gamma^2}{4}v^{k-1}$$

$$+ \left( \frac{9\gamma^2}{4}(\omega - 1) + N(1 + c_1)\beta^2(\omega - 1) + H(1 + c_2)(1 + \beta^2\omega - 2\beta)\right)w^k$$

$$\left. \left. + \frac{9\gamma^2}{4}(\omega - 1)(1 - \beta^2)w^{k-1}\right]\right]. \tag{66}$$

Considering the respective coefficients of $\|\theta^{k+1} - \theta^k\|^2$, $\|\pi^{k+1} - \pi^k\|_1^2$, $w^k$ and $v^k$, we derive the following restrictions:

$$\begin{cases} \frac{1}{4} \geq 6H(1 + c_2^{-1})\frac{aL^2}{M} + 2N(1 + c_1^{-1})L^2 + 4\gamma^2 M\tilde{L}^2 \\ \frac{1}{4} \geq 2H(1 + c_2^{-1})\tilde{L}^2 + 2N(1 + c_1^{-1})M\tilde{L}^2 \\ H \geq \frac{9\gamma^2}{4}(\omega - 1) + N(1 + c_1)\beta^2(\omega - 1) + H(1 + c_2)(1 + \beta^2\omega - 2\beta) \, . \\ \qquad + \frac{9\gamma^2}{4}(\omega - 1)(1 - \beta^2) \\ N \geq \frac{\gamma^2}{8} + \frac{9\gamma^2}{4} + N(1 + c_1)(1 + \beta^2 - 2\beta) + \frac{9(1 - \beta^2)\gamma^2}{4} \end{cases} \tag{67}$$

We now turn to selecting the free coefficients to satisfy conditions (67). Beginning with the last inequality on $N$, we set

$$c_1 = \beta, \beta \leq 1, N = 6\frac{\gamma^2}{\beta}. \tag{68}$$

Then the choice

$$c_2 = \frac{\beta}{2}, \; \beta = \frac{1}{\omega} \text{ guarantees sufficiency of } H = \frac{34\gamma^2}{\beta^2}. \tag{69}$$

to satisfy third restriction in (67).

Finally, we evaluate the first two inequalities in (67) and obtaining

$$\gamma \leq \tilde{L}^{-1}\left( 48(17\omega^3 - 2M\omega^2)\right)^{-\frac{1}{2}} \tag{70}$$

and

$$\gamma \leq \left( 2448\frac{a}{M}\omega^3 L^2 + 96\omega^2 L^2 + 16M\tilde{L}^2\right)^{-\frac{1}{2}} \tag{71}$$

respectively.

By satisfying constraint (67) via choices (68)-(71), we transform (66) into

$$\mathbb{E} \max_{\pi' \in \Lambda} \left( \left\langle p^*, \sum_{k=1}^{K} \frac{1}{K} \pi^{k+1} - \pi' \right\rangle + \left\langle \sum_{i=1}^{M} \pi_i' \nabla f_i(\theta^*), \sum_{k=1}^{K} \frac{1}{K} \theta^{k+1} - \theta^* \right\rangle \right)$$

$$+ \frac{1}{8\gamma K} \sum_{k=1}^{K} \mathbb{E} \|\pi^{k+1} - \pi^k\|^2 + \frac{\gamma}{32} \sum_{k=1}^{K} \frac{1}{K} \mathbb{E} \|\tilde{f}^k\|^2$$

$$\leq \mathbb{E} \frac{1}{2\gamma K} \left[ \max_{\pi' \in \Lambda} \left( 2D_{KL}(\pi', \pi^1) + 2\gamma \langle \overline{F}(z^1) - F(z^0), z'^* - z^1 \rangle \right) + \|\theta^1 - \theta^*\|^2 \right.$$

$$\left. + \sum_{k=0}^{1} \left[ 30\gamma^2 \omega v^k + 85\gamma^2 \omega^2 w^k \right] \right].$$

It finishes the proof. $\qquad\square$

Proceeding similarly to Corollary 1, we obtain the following bounds on the required number of iterations and the total amount of communicated information.

**Corollary 2.** *In setting of Theorem 3 with $\gamma = \gamma_1$, Algorithm 1 with exact local gradients needs*

$$\mathcal{O}\left( \frac{1}{\varepsilon} \left[ \tilde{L}\omega^{3/2} + \tilde{L}M^{1/2}\omega + L \left( \sqrt{\frac{a\omega^3}{M}} + \omega \right) \right] \right)$$

*iterations in order to reach $\varepsilon$-accuracy with respect to $W^K$. Additionally, it requires*

$$\mathcal{O}\left( \frac{1}{\varepsilon} \left[ \tilde{L}\omega^{1/2} + \tilde{L}M^{1/2} + L \left( \sqrt{\frac{a\omega}{M}} + 1 \right) \right] \right)$$

*bits communicated from nodes to the server.*

## H LLM USAGE

Beyond aiding in the editing process, no large language models (LLMs) were employed in this work. The entire intellectual content – including all facts, claims, arguments, and proofs – remained unaffected by LLM influence.

