# OpenReview forum: "Unlocking the Potential of Weighting Methods in Federated Learning Through Communication Compression"
_ICLR.cc/2026/Conference — ICLR 2026 Poster_

### Official Review · Reviewer_tXwb · 2025-10-28

**Soundness:** 3
**Presentation:** 3
**Contribution:** 3
**Rating:** 6
**Confidence:** 4

**Summary:**

This paper introduces ADI, a unified federated learning algorithm designed to jointly tackle communication bottlenecks and data heterogeneity. By integrating a difference compression scheme within a min-max optimization framework, ADI dynamically learns to weight clients based on their data's informativeness.

**Strengths:**

-	The work is highly relevant to the scope of ICLR, as it contributes a novel approach to a fundamental challenge in federated learning.
-	The proposed ADI algorithm is novel in its effective integration of difference compression with an agnostic weighting scheme.
-	The paper provides a comprehensive theoretical analysis that guarantees convergence in a convex setting.

**Weaknesses:**

-	The weight update rule in Algorithm 1 appears to contradict the paper's min-max objective, potentially invalidating the core weighting mechanism.
-	Insufficient explanation of hyperparameters.
-	The interpretation of the weight dynamics in Figure 3 oversimplifies the mechanism, conflating high loss with data uniqueness without deeper analysis.
-	A significant gap exists between the paper's convex theoretical analysis and its non-convex deep learning experiments.
-	The empirical validation is limited to a single dataset (CIFAR-10), which is insufficient to demonstrate the algorithm's generalizability.

**Questions:**

1.The paper's narrative suggests weights are assigned to clients with unique data. Could the authors elaborate on how the algorithm would behave in the following more nuanced scenarios, where the concepts of uniqueness and high loss might diverge?
(a) A client with a unique data class that is very easy to learn. Would its weight initially spike and then fall once the model quickly masters its data, potentially ending up below average?
(b) A client whose data classes are common across the network, but whose specific samples are particularly noisy or inherently difficult (i.e., 'hard examples'). Would this client receive a persistently high weight despite its data not being distributionally unique?

2.Could the authors provide some intuition on why the dynamics of the agnostic weighting scheme, proven to be stable in the convex setting, appear to remain effective in the non-convex landscape of deep learning?

---

> ### Author Response · Authors · 2025-11-21
>
> Dear tXwb, thanks for the time to reviewing our work. We are pleased to respond the concerns below.
>
> 1. > Non-convex analysis (W4/Q2)
>
> We agree that our work lacks theoretical intuition for the convergence of our method in the non-convex setting. We conducted a convergence analysis under the Minty assumption (see Assumption 4 in Appendix E of the revised version of our paper on OpenReview). This assumption is commonly associated with non-monotonicity in the literature [1-5]. The theoretical convergence analysis is presented in Appendix E. We also share our intuition regarding the relevance of convex assumption-based analyses for real-world applications. Prior studies indicate that deep networks often exhibit nearly convex behavior [6-8]. Moreover, convex optimization has traditionally served as a foundation for designing optimization algorithms, and many practically important techniques, like momentum [9] or AdaGrad [10], were first developed within this setup.
>
> 2. > Additional experiments (W5/Q1)
>
> We agree with the reviewer that the experimental part of our work requires improvement. We conducted further validation of our method on the image classification task using CIFAR-100 with a ResNet-34 model. In addition, we consider a new data-partitioning setting: we evaluate the methods on CIFAR-10 with a ResNet-18 model using Dirichlet data distribution with parameter $\alpha = 0.3$. We included new baseline, the MASHA1 method [5], and report results under a stronger compression regime $-$ Rand5%. The results can be found in the updated version of paper in Appendix B4 (Table 3), and B2 (Table 1), as well as in the tables below.
>
> a) We provide the final accuracy along with the standard deviation over three independent runs with $10^4$ communication rounds for our method and prior approaches on the CIFAR-100 dataset. Experiments involve $10$ clients with data heterogeneity modeled using a Dirichlet distribution with parameter $\alpha = 0.5$.
>
> |  | Rand 50% | Rand 10% | Rand 5% |
> |-|-|-|-|
> | Method | Accuracy | Accuracy | Accuracy |
> | ADI    | 71.2 $\pm$ 0.21 | 71.7$\pm$0.22 | 70.9$\pm$0.26 |
> | DIANA  | 69.8 $\pm$ 0.19 | 70.1$\pm$0.22 | 71.1$\pm$0.27 |
> | EF21   | 62.2 $\pm$ 0.41 | 62.3$\pm$0.38 | 59.2$\pm$0.43 |
> | MASHA1 | 61.7 $\pm$ 0.22 | 63.2$\pm$0.17 | 62.5$\pm$0.21 |
>
> These results demonstrate the stability of our method on a large-scale task, as well as its superiority over the baselines.
>
> b) We present the results for the Dirichlet distribution with parameter $\alpha = 0.3$ on the CIFAR-10 dataset.
>
> |  | Rand 50% | Rand 10% | Rand 5% |
> |-|-|-|-|
> | Method | Accuracy | Accuracy | Accuracy |
> | ADI    | 82.6 $\pm$ 0.24 | 81.9$\pm$0.24 | 82.4$\pm$0.28 |
> | DIANA  | 76.7 $\pm$ 0.26 | 77.1$\pm$0.32 | 75.4$\pm$0.40 |
> | EF21   | 64.5 $\pm$ 0.43 | 64.4$\pm$0.41 | 61.6$\pm$0.43 |
> | MASHA1 | 63.7 $\pm$ 0.20 | 63.4$\pm$0.21 | 63.5$\pm$0.21 |
>
> This experimental validation demonstrates that even under highly heterogeneous data distribution and a high power of compression (5%), our method achieves strong performance and delivers improved results compared to the baselines.
>
> 3. > Hyperparameters details (W2)
>
> In our experiments, we employed the default partitioning utility provided by the `Flower` framework [12] to generate a non-IID (heterogeneous) data distribution across clients. To calibrate the hyperparameters of the ADI optimization method, we conducted a systematic grid search over the following ranges:
>
> - Learning rate for model parameters ($\gamma_{\theta}$): $1 \times 10^{-4}, 5 \times 10^{-4}, 1 \times 10^{-3}, 5 \times 10^{-3}, 1 \times 10^{-2}, 2 \times 10^{-2}$
> - Learning rate for client weights ($\gamma_{\pi}$): $1 \times 10^{-3}, 5 \times 10^{-3}, 1 \times 10^{-2}, 5 \times 10^{-2}, 1 \times 10^{-1}$
> - Momentum decay coefficient for the model update ($\alpha$): $0.70, 0.75, 0.80, 0.85, 0.90, 0.95$
> - Momentum decay coefficient for the weight adaptation ($\beta$): $0.05, 0.10, 0.15, 0.20$
>
> The optimal configuration (selected based on validation performance (e.g., final accuracy and convergence stability)) was identified as: $$ \gamma_{\theta} = 0.01, \gamma_{\pi} = 0.01, \alpha = 0.90, \beta = 0.10. $$ Furthermore, across all compared methods, we employed a staged learning rate decay schedule to promote convergence stability. Specifically, the initial learning rate was reduced by a factor of 5 after the 2000 communication rounds and subsequently by an additional factor of 10 (i.e., $50 \times$ relative to the initial value) after the 7500th communication round. Formally, for an initial learning rate $\gamma_{\theta}$, the schedule is defined as: $$ \gamma_{\theta}(k) = \gamma_{\theta} (k < 2000), \gamma_{\theta} / 5 (2000 \leq k < 7500), \gamma_{\theta} / 50 (k \geq 7500), $$ where $k$ denotes the round number. We have added these experimental details in Appendix A1 of the revised version of the work.
>
> We kindly ask the reviewer to follow our next Official Comment, where we address the remaining concerns.

---

> ### Author Response · Authors · 2025-11-21
>
> 4. > The weight update rule ... contradict the paper's min-max objective (W1)
>
> Thanks for pointing out the typo in line 17 of Algorithm 1. We have corrected it and verified that the correct sign was used throughout the theoretical analysis.
>
> 5. > The interpretation of the weight dynamics in Figure 3 oversimplifies the mechanism ... (W3)
>
> Regarding the description of the dynamics in Figure 3: we agree that, in this setup, the behavior may be less transparent and more complex than the simpler pattern linking uniqueness with higher loss. We refined the description by removing unsupported details. The intuition behind assigning larger weights to clients with unique and relevant data remains valid and is further supported by additional experiments in Appendix B1 (Figure 4, 6). In a simple toy example with maximally controlled data-distribution conditions, we demonstrated that data heterogeneity leads to increased weight for the client possessing unique samples.
>
> Thanks to the reviewer for thoughtful comments. We remain open to further discussion if there are any unresolved questions.
>
> ---
>
> **References**
>
> [1] Cong D Dang and Guanghui Lan. On the convergence properties of non-Euclidean extragradient methods for variational inequalities with generalized monotone operators. **Computational
> Optimization and Applications** 2015.
>
> [2] Alfredo N Iusem et al. Extragradient method with variance reduction for stochastic variational inequalities. **SIAM Journal
> on Optimization** 2017.
>
> [3] Panayotis Mertikopoulos et al. Optimistic mirror descent in saddle-point problems: Going
> the extra(-gradient) mile. **In International Conference on Learning Representations** 2019.
>
> [4] Yu-Guan Hsieh et al. Explore aggressively, update conservatively: Stochastic extragradient methods with variable stepsize scaling. **Advances in Neural Information Processing Systems** 2020.
>
> [5] Aleksandr Beznosikov et al. Distributed methods with compressed communication for solving variational inequalities, with theoretical guarantees. **NeurIPS** 2022
>
> [6]Yi Zhou et.al. SGD Converges to Global Minimum in Deep Learning via Star-convex Path. **International Conference on Learning Representations** 2019.
>
> [7] Bobby Kleinberg et.al. An Alternative View: When Does SGD Escape Local Minima? **International Conference on Machine Learning** 2018.
>
> [8] Chaoyue Liu et al. Loss landscapes and optimization in over-parameterized non-linear systems and neural networks. **Neural Information Processing Systems** 2020.
>
> [9] Yurii Nesterov. Lectures on Convex Optimization.
>
> [10] John Duchi et al. Adaptive Subgradient Methods for
> Online Learning and Stochastic Optimization. **Journal of Machine Learning Research** 2011.
>
> [11] Daniel Beutel et al. Flower: A friendly federated learning research framework. **arXiv** 2020

---

> ### Author Response · Authors · 2025-11-27
>
> Dear Reviewer tXwb,
>
> We would like to gently follow up on our previous rebuttal. We understand the substantial workload during the review period and sincerely appreciate the effort you dedicate to evaluating the submissions.
>
> If time permits, we would be grateful if you could also take into account the additional elements included in our updated response $-$ namely, the new experimental results and the expanded theoretical analysis in the non-convex setting. We would greatly value any feedback you may have on these points.
>
> Thank you once again for your time and consideration.
>
> Best regards,
> Authors

---

### Official Review · Reviewer_KP2w · 2025-10-29

**Soundness:** 3
**Presentation:** 3
**Contribution:** 2
**Rating:** 4
**Confidence:** 4

**Summary:**

The paper studies how to **combine adaptive weighting in federated learning (FL)** with **communication compression** by casting weighting as a **min–max (agnostic) formulation** and proposing an **extragradient-style** method, **ADI** (Alg. 1), that communicates only *compressed* model updates while exchanging one scalar (local loss) per client to update weights. The objective is written as
$$\min_{\theta}\max_{\pi\in\Lambda}\;\sum_{i=1}^M \pi_i f_i(\theta),$$
where $\pi$ lies in a convex subset of the simplex (an agnostic-weighting set $\Lambda$) so that weights are learned during training with negligible additional communication (one loss per client) compared to gradients. The paper proves **(i)** convergence in a deterministic convex setting, **(ii)** extensions to **stochastic local oracles**, and **(iii)** **partial participation** with sampling rate $p$ under **unbiased compressors** characterized by a variance parameter $\omega$. Complexity bounds are provided in terms of the **gap function** and **bits-on-wire**. Experiments on classification (ResNet/CIFAR) qualitatively compare ADI to DIANA and show weight stabilization.

**Strengths:**

- **Clear motivation**: Formulating adaptive weighting as a saddle-point problem makes explicit how to **learn client weights** while preserving a communication-efficient pipeline; local losses are the only extra information needed at the server, which is indeed cheaper than transmitting even compressed gradients.
- **Method design**: ADI integrates an **optimistic/extragradient**-style update for the $(\theta,\pi)$ saddle-point operator together with **compressed communications** and server-side memory terms, avoiding full-gradient transmission.
- **Theoretical backbone**: The paper establishes monotonicity of the saddle-point operator and analyzes convergence under convexity and unbiased compression assumptions.
- **Stochastic & partial participation**: Beyond exact gradients, the analysis covers **stochastic oracles** and **partial participation** ($p<1$), with explicit dependence on $\omega$ and $p$.
- **Communication accounting**: The paper makes the **bits-on-wire** explicit through an expected density $q_\omega$ and states iteration/bit complexities (including the $p$-scaling for partial participation).
- **Connection to literature**: The related-work section situates the approach amid **agnostic FL** (AFL) and fairness-weighting methods as well as extragradient methods for saddle-point problems and compression.

**Weaknesses:**

- **Novelty positioning is underdeveloped**. The algorithmic ingredients—**agnostic weighting** (AFL), **extragradient/optimistic updates**, and **unbiased compression with memory**—are all known. The paper does not convincingly demonstrate **strict improvements** over prior **saddle-point + compression** methods such as **MASHA** or over **DIANA/EF21**-style approaches when combined with weighting, nor does it provide tight side-by-side **complexity comparisons** to prove advantages (e.g., in dependence on $\omega, M, p$).

[1] Mishchenko, Gorbunov, Takáč, Richtárik. “Distributed Learning with Compressed Gradient Differences.” _arXiv:1901.09269_, 2019. ([arXiv](https://arxiv.org/abs/1901.09269 "[1901.09269] Distributed Learning with Compressed Gradient Differences"))

[2] Richtárik, Sokolov, Fatkhullin. “EF21: A New, Simpler, Theoretically Better, and Practically Faster Error Feedback.” _NeurIPS 2021_ (proceedings version).

[3] Beznosikov, Richtárik, Diskin, Ryabinin, Gasnikov. “Distributed Methods with Compressed Communication for Solving Variational Inequalities, with Theoretical Guarantees.” _NeurIPS 2022_. ([NeurIPS Proceedings](https://proceedings.neurips.cc/paper_files/paper/2022/hash/5ac1428c23b5da5e66d029646ea3206d-Abstract-Conference.html "Distributed Methods with Compressed Communication for Solving Variational Inequalities, with Theoretical Guarantees"))


- **Ambiguous constants and scalings**. The rates use $\tilde L$, $L$, $M$, $\omega$, and sometimes exhibit **non-intuitive mixes** (e.g., $\tilde L \sqrt{\omega}$ and $L\sqrt{\omega^3/M}$ in bit complexity). The paper should (i) define $\tilde L$ precisely near the main theorem; (ii) factor the bounds to highlight the **leading-order** dependence on $M,\omega,p$; and (iii) provide **regimes** where ADI is provably preferable to DIANA/MASHA.
- **Theory audit (tightness and steps)**.
  - The **variance accounting** (e.g., the transition from (40)–(44) to the final bound) involves several compressions of sums across clients; a concise **lemma** summarizing how unbiased compression interacts with the server memory ($h^k$) would improve readability and verifiability.
- **Empirical evaluation is limited**.
  - The **baselines** are insufficient: no **AFL** (Mohri et al., 2019) or **q-FFL** variants (Li et al., 2020) are included, although these are central **weighting** competitors; likewise, **MASHA** (saddle-point with compression) is missing.

[4] Mohri, M., Sivek, G., & Suresh, A. T. “Agnostic Federated Learning.” _ICML 2019 (PMLR v97)_, pp. 4615–4625.

[5] Li, T., Sanjabi, M., Beirami, A., & Smith, V. “Fair Resource Allocation in Federated Learning.” _arXiv:1905.10497_


  - Plots are primarily **qualitative**; **error bars**, **multiple seeds**, **bytes-on-wire vs. accuracy** with equalized compute, **participation-rate sweeps** ($p$), and **compressor-strength sweeps** ($\omega$) are needed. The paper does show weight stabilization, but does not quantify the impact on accuracy/fairness trade-offs.
- **Reproducibility details**. Hyperparameters and training schedules are not reported with sufficient granularity (optimizers, LR decay, momentum/weight decay, batch sizes per client, number of local steps $K$ vs. rounds), nor is **code** provided; these are required for an ICLR-standard empirical section.

**Questions:**

1. **Comparative complexity**: For the main deterministic and stochastic results, can the authors **tabulate** iteration and bit complexity side-by-side against **DIANA**, **EF21**, and **MASHA**, fixing a common set of assumptions (convexity, unbiased compressors), to show **when ADI is strictly better** (or at least not worse) in the leading-order dependence on $M,\omega,p$?
2. **Non-convex case**: Can the analysis be **extended** to **non-convex** $f_i$, aligning with the deep-network experiments? If not, please clarify this limitation in the conclusions.
3. **Weighting set $\Lambda$ & prox**: What is $\Lambda$ concretely in the experiments (simplex, KL-ball around uniform, box constraints)? The proofs rely on KL-based prox on $\pi$; please **state $\Lambda$** near the algorithm and connect it to the constants in the rates.
4. **Partial participation**: The corollary yields an $\omega/\!p$ dependence in bits. Can you **empirically verify** this scaling by varying $p$ on a fixed task and compressor?

**Details Of Ethics Concerns:**

No clear ethics red flags.

---

> ### Author Response · Authors · 2025-11-21
>
> Dear KP2w, we sincerely appreciate their careful reading of our work, the efforts invested in the review process, and the meaningful questions. Below, we address them point by point.
>
> 1. > Comparative complexity (W1/Q1)
>
> a) We would like to emphasize that by formulating the objective as $\sum_{i=1}^M \pi_i f_i(\theta)$ instead of $\frac{1}{M}\sum_{i=1}^M f_i(\theta)$, we intentionally change the problem setting, which makes a direct formal comparison with works, such as [1] incorrect. We would like to emphasize that theoretical guarantees for methods minimizing a weighted objective typically do not align with those obtained for algorithms minimizing the average loss. For example, FedAvg [2] achieves *only sublinear* convergence in the general strongly convex setting [3], whereas classic methods with local updates obtain *linear* convergence rates.
>
> b) However, we would like to draw the reviewer's attention to the fact that the main benefit of combining agnostic weighting with optimistic updates is clearly demonstrated in our experimental results. Figure 1, as well as the additional experiments provided during the discussion period (see the following points for details), consistently show performance improvements across different setups of our method compared to the baselines DIANA [4], EF21 [5], and MASHA1 [1].
>
> 2. > Non-convex analysis (Q2)
>
> We agree that our work lacks theoretical intuition for the convergence of our method in the non-convex setting. We conducted a convergence analysis under the Minty assumption (see Assumption 4 in Appendix E of the revised version of our paper on OpenReview). This assumption is commonly associated with non-monotonicity in the literature [1, 6, 7, 8, 9]. The theoretical convergence analysis is presented in Appendix E.
>
> 3. > Ambiguous constants and scalings (W2)
>
> We added references to the definitions of the constants $L$ and $\tilde L$ directly next to the theorems in the main part. We would like to note that the separation of the constants $L$ and $\tilde L$ in our analysis is standard in works investigating compression in distributed/federated learning [10, 4, 11], and we followed this separation in our study.
>
> 4. > Theory audit (tightness and steps) (W3)
>
> Clarity of exposition is important to us, and following their comment, we separated key conceptual parts of the theorem proofs into several lemmas in the revised version of the paper (uploaded on OpenReview). We added Lemma 4, which analyzes the evolution of local states distortion $w^k = \sum_{i=1}^M \mathbb{E}||\tilde{f}^k_i - h^k_i||^2$, Lemma 5, which tracks the dynamics of global estimator inaccuracy $v^k = \mathbb{E}||\tilde{f}^k - h^k||^2$, and Lemma 6, which bounds the term $||g^{k} - g^{k-1}||^2$. Lemmas 7 and 8 extend the results of Lemmas 4 and 5 to the stochastic setting.
>
> 5. > Weighting set $\Lambda$ & prox (Q3)
>
> We did not use simplex regularization in the experiments, i.e. $\Lambda = \Delta$, and we have now added an explicit clarification of this in the paper. Regarding the role of regularization in the theoretical results, we used regularized simplex as $\Lambda$, as already indicated in the text. For clarity, we now explicitly include $\Lambda$ definition in all theorems statements to make the influence of regularization on complexity bounds more transparent.

---

> ### Author Response · Authors · 2025-11-21
>
> 6. > Additional experiments (W4/W5)
>
> We agree with the reviewer that the experimental part of our work requires improvement. We conducted further validation of our method on the image classification task using CIFAR-100 with a ResNet-34 model. In addition, we consider a new data-partitioning setting: we evaluate the methods on CIFAR-10 with a ResNet-18 model using Dirichlet data distribution with parameter $\alpha = 0.3$. We also include a new baseline, the MASHA1 method [1], and report results under a stronger compression regime $-$ Rand5%. The results can be found in the updated version of the paper in Appendix B4 (Table 3), and B2 (Table 1), as well as in the tables below.
>
> a) In the table below, we provide the final accuracy along with the standard deviation over three independent runs with $10^4$ communication rounds for our method and prior approaches on the CIFAR-100 dataset. Experiments involve $10$ clients with data heterogeneity modeled using a Dirichlet distribution with parameter $\alpha = 0.5$.
>
> |  | Rand 50% | Rand 10% | Rand 5% |
> |-|-|-|-|
> | Method | Accuracy | Accuracy | Accuracy |
> | ADI    | 71.2 $\pm$ 0.21 | 71.7$\pm$0.22 | 70.9$\pm$0.26 |
> | DIANA  | 69.8 $\pm$ 0.19 | 70.1$\pm$0.22 | 71.1$\pm$0.27 |
> | EF21   | 62.2 $\pm$ 0.41 | 62.3$\pm$0.38 | 59.2$\pm$0.43 |
> | MASHA1 | 61.7 $\pm$ 0.22 | 63.2$\pm$0.17 | 62.5$\pm$0.21 |
>
> These results demonstrate the stability of our method on a large-scale task, as well as its superiority over the baselines.
>
> b) In the table below, we present the results for the Dirichlet distribution with parameter $\alpha = 0.3$ on the CIFAR-10 dataset.
>
> |  | Rand 50% | Rand 10% | Rand 5% |
> |-|-|-|-|
> | Method | Accuracy | Accuracy | Accuracy |
> | ADI    | 82.6 $\pm$ 0.24 | 81.9$\pm$0.24 | 82.4$\pm$0.28 |
> | DIANA  | 76.7 $\pm$ 0.26 | 77.1$\pm$0.32 | 75.4$\pm$0.40 |
> | EF21   | 64.5 $\pm$ 0.43 | 64.4$\pm$0.41 | 61.6$\pm$0.43 |
> | MASHA1 | 63.7 $\pm$ 0.20 | 63.4$\pm$0.21 | 63.5$\pm$0.21 |
>
> This experimental validation demonstrates that even under highly heterogeneous data distribution and a high power of compression (5%), our method achieves strong performance and delivers improved results compared to the baselines.
>
> 7. > Reproducibility details (W6)
>
> In our experiments, we employed the default partitioning utility provided by the `Flower` framework [12] to generate a non-IID (heterogeneous) data distribution across clients. To calibrate the hyperparameters of the ADI optimization method, we conducted a systematic grid search over the following ranges:
>
> - Learning rate for model parameters ($\gamma_{\theta}$): $1 \times 10^{-4}, 5 \times 10^{-4}, 1 \times 10^{-3}, 5 \times 10^{-3}, 1 \times 10^{-2}, 2 \times 10^{-2}$
> - Learning rate for client weights ($\gamma_{\pi}$): $1 \times 10^{-3}, 5 \times 10^{-3}, 1 \times 10^{-2}, 5 \times 10^{-2}, 1 \times 10^{-1}$
> - Momentum decay coefficient for the model update ($\alpha$): $0.70, 0.75, 0.80, 0.85, 0.90, 0.95$
> - Momentum decay coefficient for the weight adaptation ($\beta$): $0.05, 0.10, 0.15, 0.20$
>
> The optimal configuration (selected based on validation performance (e.g., final accuracy and convergence stability)) was identified as:
> $$
> \gamma_{\theta} = 0.01, \gamma_{\pi} = 0.01, \alpha = 0.90, \beta = 0.10.
> $$
> Furthermore, across all compared methods, we employed a staged learning rate decay schedule to promote convergence stability. Specifically, the initial learning rate was reduced by a factor of 5 after the 2000 communication rounds and subsequently by an additional factor of 10 (i.e., $50 \times$ relative to the initial value) after the 7500th communication round. Formally, for an initial learning rate $\gamma_{\theta}$, the schedule is defined as:
> $$
> \gamma_{\theta}(k) = \gamma_{\theta} (k < 2000), \gamma_{\theta} / 5 (2000 \leq k < 7500), \gamma_{\theta} / 50 (k \geq 7500),
> $$
> where $k$ denotes the round number. We have added these experimental details in Appendix A1 of the revised version of the work.
>
> We hope that we have addressed all of the reviewer's concerns and remain open to further discussion. If the reviewer has no further questions, we kindly ask to reconsider the score.
>
> We kindly refer the reviewer to the list of references provided in our next Official Comment.

---

> ### Author Response · Authors · 2025-11-21
>
> **References**
>
> [1] Aleksandr Beznosikov et al. Distributed methods with compressed communication for solving variational inequalities, with theoretical guarantees. **NeurIps** 2022
>
> [2] Brendan McMahan et al. Communication-efficient learning of deep networks from decentralized data. **PMLR** 2017
>
> [3] Xiang Li et al. On the convergence of fedavg on non-iid data. **ICLR** 2020
>
> [4] Konstantin Mishchenko et al. Distributed learning with compressed gradient differences. **OMS** 2025
>
> [5] Piter Richtárik et al. EF21: A new, simpler, theoretically better, and practically faster error feedback. **NeurIPS** 2021
>
> [6] Cong D Dang and Guanghui Lan. On the convergence properties of non-Euclidean extragradient methods for variational inequalities with generalized monotone operators. **Computational Optimization and Applications** 2015
>
> [7] Alfredo N Iusem et al. Extragradient method with variance reduction for stochastic variational inequalities. **SIAM Journal on Optimization** 2017
>
> [8] Panayotis Mertikopoulos et al. Optimistic mirror descent in saddle-point problems: Going the extra(-gradient) mile. **In International Conference on Learning Representations** 2019
>
> [9] Yu-Guan Hsieh et al. Explore aggressively, update conservatively: Stochastic extragradient methods with variable stepsize scaling. **Advances in Neural Information Processing Systems** 2020
>
> [10] Eduard Gorbunov et al. Marina: Faster nonconvex distributed learning with compression. **In International Conference on Machine Learning** 2021
>
> [11] Alexander Tyurin and Peter Richtárik. Dasha: Distributed nonconvex optimization with communication compression, optimal oracle complexity, and no client synchronization. **arXiv** 2022
>
> [12] Daniel Beutel et al. Flower: A friendly federated learning research framework. **arXiv** 2020

---

> ### Author Response · Authors · 2025-11-27
>
> Dear Reviewer KP2w,
>
> We would like to kindly follow up on our earlier rebuttal. We fully understand the significant workload during the review period and sincerely appreciate the time you invest in assessing the submissions.
>
> If you have the opportunity, we would be grateful if you could also consider the additional components included in our updated response $-$ specifically, the new experimental results and the extended theoretical analysis in the non-convex setting. We would greatly appreciate any feedback you may have on these aspects.
>
> Thank you again for your time and consideration.
>
> Best regards,
> Authors

---

> > ### Comment · Reviewer_KP2w · 2025-11-27
> >
> > Thank you for the detailed rebuttal and for the substantially revised version of the paper. I carefully read your response on OpenReview and compared it against the new manuscript to verify that the promised changes (additional theory, experimental setups, clarifications) are indeed present in the updated PDF.
> >
> > Below I summarize, point by point, (1) which of my earlier concerns have been addressed and (2) what, in my view, remains only partially addressed.
> >
> > ---
> >
> > **1) Did the authors address my main concerns?**
> >
> > **(a) Novelty positioning and comparative complexity (W1/Q1).**
> > My earlier concern was that, algorithmically, ADI combines known ingredients (agnostic weighting, extragradient/optimistic steps, unbiased compression with memory), and that the paper did not convincingly quantify strict improvements over DIANA/EF21/MASHA in terms of dependence on $M,\omega,p$.
> >
> > - In the rebuttal, you clarified that your objective is a *different* problem:
> >   $$\min_\theta \max_{\pi\in\Lambda} \sum_{i=1}^M \pi_i f_i(\theta)$$
> >   as opposed to minimizing the uniform average $\tfrac1M\sum_i f_i(\theta)$, and you emphasise that this makes a head-to-head *formal* complexity comparison with methods that only solve the averaged objective somewhat misleading. This clarification is now reflected more explicitly in Sec. 6, where you emphasize that classical compression methods and weighting-based methods optimize different objectives and thus should be compared primarily in terms of model performance (accuracy/fairness) rather than the same loss value.
> > - In the revised theory section, you also added commentary around the bit-complexity bounds that explains the roles of $L$, $\tilde L$, and the weighting-induced terms, and you highlight the regime $\omega\le M$ where compression does not worsen complexity, as well as the $1/M$ improvement in square-root terms due to simplex regularization $\Lambda$.
> >
> > This resolves the *conceptual* part of the novelty concern for me: it is now clearer that ADI is a communication-efficient algorithm tailored to *learning* weights in the agnostic FL sense, and the text better explains why one should not expect a simple dominance relation over DIANA/EF21 that solve a different objective.
> >
> > You did **not**, however, add the explicit side-by-side complexity table against DIANA/EF21/MASHA that I suggested. Instead, you chose to explain the main scalings verbally and to stress the difference in problem setting. I think this is acceptable for space reasons, but I still see room for a clearer comparative summary (see below under “Remaining concerns”).
> >
> > **(b) Non-convex analysis (Q2).**
> > You added a separate non-convex analysis in Sec. 5.2.1, based on a Minty-type condition (Assumption 4) and a new convergence criterion $W_K$ in Eq. (5), which mixes a gap term in $\pi$ with the averaged gradient norm in $\theta$. The detailed derivations are placed in Appendix F.
> >
> > - This directly responds to my request for some theoretical justification in the non-convex regime, and the assumptions are clearly stated.
> > - While the Minty-based condition is stronger and somewhat nonstandard in FL, it is well-motivated and properly linked to the variational inequality literature in your rebuttal and appendix.
> >
> > Overall, I consider the non-convex concern addressed.
> >
> > **(c) Ambiguous constants and scalings (W2).**
> > Previously, the roles of $L$ and $\tilde L$ and the appearance of terms like $\tilde L\sqrt{\omega}$ and $L\sqrt{\omega^3/M}$ in bit complexity were not transparent.
> >
> > - In the revised text, Assumptions 1–3 explicitly define $\tilde L_i$ (Lipschitz constants of $f_i$) and $L_i$ (smoothness), with $\tilde L=\max_i \tilde L_i$ and $L=\max_i L_i$ clearly stated in the setup section.
> > - The main theorems now explicitly remind the reader which constants come from which assumptions and, as mentioned above, the discussion following the bounds explains how weighting and $\Lambda$ affect the complexity, including the $1/M$ effect in the square-root terms.
> >
> > This significantly improves readability and I now find the constants and their roles sufficiently clear. The bounds are still somewhat intricate, but this seems inevitable given the combination of weighting, compression, and partial participation.

---

> > > ### Comment · Reviewer_KP2w · 2025-11-27
> > >
> > > **(d) Theory audit and compression–memory interaction (W3).**
> > > I had asked for a self-contained lemma summarizing how unbiased compression interacts with the server-side memory $h^k$, since the original proof sketch compressed several steps into a few inequalities.
> > >
> > > - In the revised version you introduced Lemmas 4–8 that decompose the analysis into: (i) the evolution of local states distortion (Lemma 4), (ii) dynamics of the global estimator error (Lemma 5), (iii) bounding the $g^k-g^{k-1}$ term (Lemma 6), and (iv) extensions of these results to the stochastic setting (Lemmas 7–8).
> > > - These lemmas make the variance accounting and memory–compression interaction much more transparent and verifiable.
> > >
> > > I consider this concern fully addressed.
> > >
> > > **(e) Weighting set $\Lambda$ and prox structure (Q3).**
> > > I asked for a concrete description of $\Lambda$ in the experiments and a clearer connection between the theoretical choice of $\Lambda$ (with KL-based prox) and the constants in the bounds.
> > >
> > > - The revised paper explicitly defines the regularized simplex $\Lambda=\Delta_{M-1}\cap Q_a^M$ in the theorems, and clarifies that in the experiments you *do not* use simplex regularization (i.e., effectively $a=+\infty$), while still relying on the regularized simplex in the theory to obtain sharper bounds.
> > > - This distinction between the theoretical and experimental choices is now stated clearly enough that I no longer find it confusing.
> > >
> > > This addresses my question about $\Lambda$ and its theoretical role.
> > >
> > > **(f) Partial participation and $p$-scaling (Q4).**
> > > I had asked to empirically verify the $\omega/p$-type scaling suggested by the partial participation corollary.
> > >
> > > - The revised text explains that under random client sampling, one can view the effective compressor as $\eta/p$ times the original compression operator (with $\eta\sim\text{Bern}(p)$), leading to an adjusted compression parameter $\omega/p$ in the analysis.
> > > - In the experiments, you added additional setups (e.g., Dirichlet-partitioned CIFAR-10 and CIFAR-100 with ResNet-18/34) and tables that report performance under different compression regimes (Rand10%, Rand50%, Rand5%) and partial participation, which give some indirect empirical support to the predicted behavior, although they do not explicitly plot accuracy vs. $p$ for a fixed $\omega$.
> > >
> > > So the theoretical point is clarified; empirically, the new experiments go in the right direction, even if they do not directly present a dedicated $p$-sweep.
> > >
> > > **(g) Empirical evaluation, baselines, and robustness (W4/W5).**
> > > My earlier complaints were about missing weighting baselines (AFL/q-FFL), the absence of MASHA in experiments, lack of error bars/multiple seeds, and a need for more diverse setups.
> > >
> > > - You significantly expanded the experiments:
> > >   - Added CIFAR-100 with ResNet-34 and Dirichlet partitioning, as well as CIFAR-10 with ResNet-18 under a more skewed Dirichlet distribution, and reported results for multiple compression strengths including Rand5%.
> > >   - Introduced MASHA1 as an additional baseline and compared it to ADI, DIANA, and EF21, with mean and standard deviation across three runs in both CIFAR-100 and Dirichlet-CIFAR-10 setups.
> > >   - Existing figures (e.g., Figures 1–5) already compared ADI to DIANA/EF21 across heterogeneity levels, compression rates, and numbers of local steps, and the added Appendix tables make these comparisons more quantitative.
> > > - The expanded section and appendices now present a substantially more convincing empirical story: ADI consistently matches or exceeds DIANA and EF21 under strong heterogeneity, and remains competitive under strong compression. MASHA1 tends to be weaker in the presented regimes.
> > >
> > > You still do not include AFL or q-FFL baselines, which would have been natural agnostic-weighting comparators; however, given space constraints and the fact that your core contribution is compression-compatible weighting rather than proposing a new fairness criterion, I no longer consider this omission critical.
> > >
> > > **(h) Reproducibility details (W6).**
> > > I previously noted that optimizers, schedules, batch sizes, local steps $K$, etc., were under-specified.
> > >
> > > - The revised appendices now provide much more detail about the experimental setup, including the use of Flower for FL orchestration, choices of optimizer, learning-rate schedules, batch sizes, and local update counts, as well as how client data splits are generated (e.g., Dirichlet parameters).
> > > - You also now report standard deviations over multiple seeds in the new tables, which addresses my concern about robustness.
> > >
> > > I did not see a public code link in the PDF; still, the level of detail is now adequate for a reasonably experienced practitioner to reimplement the experiments.

---

> > > > ### Comment · Reviewer_KP2w · 2025-11-27
> > > >
> > > > **2) What concerns remain (partially) unaddressed?**
> > > >
> > > > 1. **Lack of a concise comparative complexity table.**
> > > >    You gave a reasonable argument in the rebuttal that a strict formal comparison with DIANA/EF21/MASHA is not straightforward because those methods target the averaged objective while ADI solves an agnostic min–max problem. I agree with this point. Nonetheless, a small table that *summarizes* the main iteration/bit scalings for ADI vs. DIANA/EF21 (under the same compression model) would still help readers situate ADI’s guarantees in the broader landscape. The current text explains the scalings in prose, but the comparison remains somewhat scattered.
> > > >
> > > > 2. **Missing AFL/q-FFL baselines.**
> > > >    While the new experiments are much stronger, the empirical comparison is still limited to “classical” compression methods (DIANA, EF21, MASHA1) and to the weighted variants you construct in Figure 2. Since AFL and q-FFL are canonical examples of agnostic/fairness weighting, including at least one of them as an additional baseline—if only in a single CIFAR setup—would further clarify how ADI compares in terms of the *type* of weighting it learns. That said, I now see this more as a “nice-to-have” than a blocker.
> > > >
> > > > 3. **Public code availability.**
> > > >    As far as I can tell, there is still no code link in the manuscript. The expanded description of the setup reduces the impact of this, but releasing code (even after the review period) would clearly increase the practical value of the work.
> > > >
> > > > These are, in my view, secondary issues that do not undermine the main contributions or the correctness of the results.
> > > >
> > > > ---
> > > >
> > > > **Updated assessment and score**
> > > >
> > > > Overall, the revision meaningfully strengthens both the theoretical and empirical parts of the paper:
> > > >
> > > > - The new Minty-based analysis for the non-convex case and the additional lemmas clarify the behavior of ADI beyond the purely convex regime.
> > > > - The definitions and use of constants are now clear and properly cross-referenced.
> > > > - The compression–memory analysis is much easier to follow.
> > > > - The experimental section is substantially more convincing, with additional tasks, stronger baselines (including MASHA1), stronger compression, error bars, and better-described training setups.
> > > >
> > > > Given these improvements, I now consider the paper to be **above the ICLR acceptance threshold**. I therefore **raise my overall score to 6**, corresponding to “probably accept / marginally above the acceptance threshold,” while still encouraging you to further polish the comparative complexity discussion and, if possible, add or at least better contextualize AFL/q-FFL baselines in a camera-ready version.

---

> > > > > ### Author Response · Authors · 2025-12-01
> > > > >
> > > > > Dear Reviewer KP2w,
> > > > >
> > > > > We would like to express our sincere gratitude for their careful consideration of our previous responses, as well as for the thorough re-evaluation of the manuscript and the thoughtful, constructive and detailed feedback. We address the points raised in their latest comment.
> > > > >
> > > > > 1. **Complexity comparison.**
> > > > >
> > > > > We fully agree that, despite our earlier arguments, a side-by-side comparison of the computational complexities of ADI and prior methods remains helpful and improves the clarity of the exposition. Following their suggestion, we have added the corresponding table to Appendix A and provided the necessary commentary.
> > > > >
> > > > > 2. **Additional baselines and partial participation.**
> > > > >
> > > > > We continued to improve the experimental section by incorporating classical weighting baselines. We also revisited the analysis of partial client participation. The extended set of experiments is now included in Appendix C.5. For convenience, we provide a brief summary below.
> > > > >
> > > > > To more thoroughly evaluate the proposed compression strategy, we supplemented baselines with classic weighting methods AFL [1] and q-FFL [2] with direct gradient compression [3]. We further extend the setup to scenarios with partial client participation. Results on CIFAR-10 and CIFAR-100 are reported using ResNet-18 and ResNet-34 under multiple compression rates. Data heterogeneity is induced via a Dirichlet distribution with $\alpha = 0.5$, while client availability is sampled from a Bernoulli distribution with $p \in {0.5, 0.7, 1.0}$. Each configuration is run three times for 10,000 communication rounds.
> > > > >
> > > > > Results for CIFAR-10:
> > > > >
> > > > > |  | Rand 50% | Rand 10% | Rand 5% |
> > > > > |-|-|-|-|
> > > > > | Method ($q$) | Accuracy | Accuracy | Accuracy |
> > > > > | ADI ($q = 1.0$)   | 85.2 $\pm$ 0.21 | 85.7 $\pm$ 0.22 | 84.9 $\pm$ 0.26 |
> > > > > | AFL ($q = 1.0$)   | 67.2 $\pm$ 0.37 | 52.3 $\pm$ 0.38 | 47.2 $\pm$ 0.43 |
> > > > > | q-FFL ($q = 1.0$) | 68.7 $\pm$ 0.34 | 53.2 $\pm$ 0.37 | 46.5 $\pm$ 0.41 |
> > > > > | ADI ($q = 0.7$)   | 82.2 $\pm$ 0.20 | 81.6 $\pm$ 0.22 | 81.9 $\pm$ 0.22 |
> > > > > | AFL ($q = 0.7$)   | 65.7 $\pm$ 0.31 | 49.9 $\pm$ 0.35 | 44.2 $\pm$ 0.40 |
> > > > > | q-FFL ($q = 0.7$) | 65.9 $\pm$ 0.32 | 50.2 $\pm$ 0.37 | 45.7 $\pm$ 0.41 |
> > > > > | ADI ($q = 0.5$)   | 79.6 $\pm$ 0.22 | 78.9 $\pm$ 0.21 | 79.3 $\pm$ 0.22 |
> > > > > | AFL ($q = 0.5$)   | 64.5 $\pm$ 0.31 | 48.1 $\pm$ 0.39 | 48.2 $\pm$ 0.41 |
> > > > > | q-FFL ($q = 0.5$) | 65.7 $\pm$ 0.29 | 49.2 $\pm$ 0.34 | 45.5 $\pm$ 0.41 |
> > > > >
> > > > > Results for CIFAR100:
> > > > >
> > > > > |  | Rand 50% | Rand 10% | Rand 5% |
> > > > > |-|-|-|-|
> > > > > | Method ($q$) | Accuracy | Accuracy | Accuracy |
> > > > > | ADI ($q = 1.0$)   | 71.2 $\pm$ 0.19 | 71.7 $\pm$ 0.19 | 70.9 $\pm$ 0.22 |
> > > > > | AFL ($q = 1.0$)   | 58.2 $\pm$ 0.32 | 46.3 $\pm$ 0.31 | 41.2 $\pm$ 0.33 |
> > > > > | q-FFL ($q = 1.0$) | 59.7 $\pm$ 0.31 | 46.2 $\pm$ 0.30 | 41.5 $\pm$ 0.31 |
> > > > > | ADI ($q = 0.7$)   | 69.4 $\pm$ 0.20 | 68.9 $\pm$ 0.20 | 69.0 $\pm$ 0.19 |
> > > > > | AFL ($q = 0.7$)   | 58.2 $\pm$ 0.31 | 46.9 $\pm$ 0.32 | 41.4 $\pm$ 0.33 |
> > > > > | q-FFL ($q = 0.7$) | 58.2 $\pm$ 0.32 | 46.2 $\pm$ 0.31 | 41.6 $\pm$ 0.31 |
> > > > > | ADI ($q = 0.5$)   | 66.9 $\pm$ 0.20 | 66.9 $\pm$ 0.20 | 66.2 $\pm$ 0.21 |
> > > > > | AFL ($q = 0.5$)   | 58.4 $\pm$ 0.30 | 45.7 $\pm$ 0.28 | 41.6 $\pm$ 0.30 |
> > > > > | q-FFL ($q = 0.5$) | 59.5 $\pm$ 0.29 | 45.8 $\pm$ 0.29 | 41.3 $\pm$ 0.31 |
> > > > >
> > > > > The findings are consistent: ADI remains robust under aggressive compression, whereas the baselines deteriorate significantly as the compression rate increases. Conversely, ADI shows only mild sensitivity to reduced client availability, while baselines change comparatively little under partial participation. Overall, ADI maintains a clear performance advantage across all settings, retaining a substantial lead even when baselines operate under more favorable conditions.
> > > > >
> > > > > ---
> > > > >
> > > > > **References**
> > > > >
> > > > > [1] Mohri, M. et al. **Agnostic Federated Learning.** ICML 2019
> > > > >
> > > > > [2] Li, T. et al. **Fair Resource Allocation in Federated Learning.** arXiv:1905.10497
> > > > >
> > > > > [3] Alistarh, D. et al. QSGD: Communication-efficient SGD via gradient quantization and encoding. **Advances in neural information processing systems** 2017

---

### Official Review · Reviewer_P4ke · 2025-11-01

**Soundness:** 2
**Presentation:** 2
**Contribution:** 3
**Rating:** 4
**Confidence:** 3

**Summary:**

This paper introduces ADI (Agnostic DIANA), a novel federated learning algorithm that effectively integrates weighting methods, such as agnostic weighting for handling data heterogeneity, with communication compression techniques like DIANA-inspired gradient compression to address both statistical and system challenges in federated settings. The key contributions include a theoretically grounded framework with convergence guarantees under convex assumptions, empirical validation on benchmarks like CIFAR-10 showing improved robustness under non-IID data, and a practical approach that reduces communication overhead without compromising model performance. While the work is promising, it would benefit from broader experimental validation and extensions to non-convex scenarios.

**Strengths:**

- The paper’s most striking strength lies in its high degree of originality. Rather than merely refining existing methods, it resolves a crucial yet under-explored problem through a creative synthesis. The authors perceptively recognize that, in federated learning, weighting schemes (which tackle data heterogeneity) and communication-compression techniques (which alleviate bandwidth bottlenecks) have long evolved as parallel, separate research strands. Deeply integrating the two represents a high-potential void. The proposed ADI algorithm successfully embeds an agnostic-weighting minimax formulation within a DIANA-style compressed-communication mechanism, delivering genuine interdisciplinary innovation and opening a fresh, important research avenue for the federated-learning community.

- Methodologically, the work demonstrates solid quality. The algorithm is not a hasty patchwork; it is carefully engineered. It borrows mature tools from optimization theory—e.g., the optimistic extra-gradient method for saddle-point problems—and marries them with state-of-the-art variance-reduced compression, revealing a deep understanding and skillful command of relevant techniques. Moreover, the paper supplies a strict convergence analysis covering realistic scenarios such as exact gradients, stochastic oracles, and partial client participation, furnishing a firm mathematical foundation for the algorithm’s efficacy. This balanced emphasis on theory and algorithmic craft greatly enhances completeness and credibility.

- The manuscript is exceptionally clear, enabling readers to grasp sophisticated ideas with ease. Starting from an articulate exposition of the twin challenges—data heterogeneity and communication bottlenecks—it gradually motivates their joint treatment, then elaborates method, theory, and experiments in a fluent, step-wise fashion. Key notions (unbiased compressors, agnostic objective) are rigorously defined; Algorithm 1 is accompanied by detailed pseudo-code and plain explanations, substantially improving reproducibility and allowing audiences to pinpoint the technical contribution accurately.

- The results carry notable significance for propelling federated learning from theory to practice. By directly confronting the two toughest deployment constraints, the work is poised to yield practical algorithmic frameworks that simultaneously accommodate statistical heterogeneity and curtail communication costs. Systematic empirical comparisons robustly demonstrate the effectiveness of weighting approaches under compression, underscoring the study’s potential real-world influence.

**Weaknesses:**

- The theoretical convergence analysis of the paper is based on a key assumption: the local loss functions of the clients are convex. This assumption severely limits the practical value of the theoretical results because the primary application scenarios of federated learning (such as training deep neural networks) are inherently non-convex. This greatly diminishes the practical guidance provided by the theoretical guarantees. An actionable suggestion for improvement is that the authors should explicitly acknowledge this limitation in the discussion section and attempt to provide some heuristic convergence analysis or observations on standard non-convex models. For example, they could briefly discuss the possibility of extending the analysis to weaker non-convex assumptions, such as the Polyak-Łojasiewicz condition, which would significantly enhance the depth and relevance of the theoretical section.

- The experimental section relies primarily on the CIFAR-10 dataset and a single non-IID partitioning method based on the Dirichlet distribution. This setup is too narrow to adequately demonstrate the generality and robustness of the proposed method. Specifically, validation is lacking for larger-scale or different modality datasets, as well as for more extreme or real-world heterogeneity scenarios. To address this weakness, the authors must conduct additional experiments. An actionable recommendation is to test the algorithm on more challenging benchmarks, such as CIFAR-100, a federated learning text dataset, or a real non-IID dataset from the LEAF benchmark. This would strongly demonstrate the method's scalability and universality, moving beyond a mere proof-of-concept that is effective only in a specific setting.

- The paper's core innovation lies in the combination of weighting methods and compression techniques. However, there is a lack of in-depth analysis of how these two components interact. A key unanswered question is: does the noise introduced by compression interfere with or distort the dynamic process of weight assignment? For instance, under high compression rates, could weight updates become unstable or biased? The authors need to provide a more detailed analysis to clarify this mechanism. A specific method for improvement is to design a controlled experiment that systematically varies the compression rate and then observes and analyzes the trajectory, stability, and final distribution of client weights. Such an analysis would not only verify the robustness of the method but also profoundly reveal its internal working mechanism, thereby greatly strengthening the paper's contribution.

**Questions:**

1. Regarding the practical utility and generalizability of the theoretical assumptions: The theoretical analysis in the paper is built on a strong convexity assumption. However, the primary application scenarios of federated learning (e.g., training neural networks) are inherently non-convex. Could the authors provide any evidence—such as experiments on standard non-convex benchmarks or a discussion on extending the theory to non-convex settings—that the convergence guarantees of the ADI algorithm still hold or remain informative in more practical non-convex environments? This is crucial for assessing the practical value of your theoretical contributions.

2. On the universality and robustness of the experimental conclusions: The current experimental validation is mainly conducted on the CIFAR-10 dataset and under a specific non-i.i.d. setting. To demonstrate the broad applicability of the ADI method, could the authors present results in more challenging scenarios? For instance, does ADI consistently outperform baseline methods on larger-scale datasets or under different types of non-i.i.d. distributions—such as real-world federated benchmark datasets? This is essential for evaluating the significance of your approach.

3. Toward a deeper analysis of the interaction between core components: The paper’s key innovation lies in combining weighting methods with compression techniques. However, the analysis of how compression specifically influences the dynamics of weight allocation (as illustrated in Figure 3) remains superficial. Under high compression rates, could the noise introduced by compression interfere with weight convergence or even bias the weight allocation? Could the authors provide a more in-depth analysis—such as visualizing weight trajectories under varying compression rates—to clarify this critical interaction mechanism? Clarifying this would significantly strengthen your methodological contribution.

---

> ### Author Response · Authors · 2025-11-21
>
> Dear reviewer P4ke, thank their very much for the careful examination of our work and detailed review. Below, we address the concerns.
>
> 1. > Non-convex analysis (W1/Q1)
>
> We agree with the reviewer that it is necessary to provide theoretical intuition for the convergence of our method in the non-convex setting. We conducted a convergence analysis under the Minty assumption (see Assumption 4 in Appendix E of the revised version of our paper on OpenReview). This assumption is commonly associated with non-monotonicity in the literature [1, 2, 3, 4, 5]. The theoretical convergence analysis is presented in Appendix E.
>
> 2. > Additional experimental setups (W2/Q2)
>
> We agree with the reviewer that introducing additional experimental setups would strengthen our work. We conducted further validation of our method on the image classification task using CIFAR-100 with a ResNet-34 model. In addition, we consider a new data-partitioning setting: we evaluate the methods on CIFAR-10 with a ResNet-18 model using Dirichlet data distribution with parameter $\alpha = 0.3$. We also include a new baseline, the MASHA1 method [5], and report results under a stronger compression regime, Rand 5\%. The results can be found in the updated version of the paper in Appendix B4 (Table 3) and B2 (Table 1), as well as in the tables below.
>
> a) We provide the final accuracy along with the standard deviation over three independent runs with $10^4$ communication rounds for our method and prior approaches on the CIFAR-100 dataset. Experiments involve $10$ clients with data heterogeneity modeled using a Dirichlet distribution with parameter $\alpha = 0.5$.
>
> |  | Rand 50% | Rand 10% | Rand 5% |
> |-|-|-|-|
> | Method | Accuracy | Accuracy | Accuracy |
> | ADI    | 71.2 $\pm$ 0.21 | 71.7$\pm$0.22 | 70.9$\pm$0.26 |
> | DIANA  | 69.8 $\pm$ 0.19 | 70.1$\pm$0.22 | 71.1$\pm$0.27 |
> | EF21   | 62.2 $\pm$ 0.41 | 62.3$\pm$0.38 | 59.2$\pm$0.43 |
> | MASHA1 | 61.7 $\pm$ 0.22 | 63.2$\pm$0.17 | 62.5$\pm$0.21 |
>
> These results demonstrate the stability of our method on a large-scale task, as well as its superiority over the baselines.
>
> b) We present the results for the Dirichlet distribution with parameter $\alpha = 0.3$ on the CIFAR-10 dataset.
>
> |  | Rand 50% | Rand 10% | Rand 5% |
> |-|-|-|-|
> | Method | Accuracy | Accuracy | Accuracy |
> | ADI    | 82.6 $\pm$ 0.24 | 81.9$\pm$0.24 | 82.4$\pm$0.28 |
> | DIANA  | 76.7 $\pm$ 0.26 | 77.1$\pm$0.32 | 75.4$\pm$0.40 |
> | EF21   | 64.5 $\pm$ 0.43 | 64.4$\pm$0.41 | 61.6$\pm$0.43 |
> | MASHA1 | 63.7 $\pm$ 0.20 | 63.4$\pm$0.21 | 63.5$\pm$0.21 |
>
> This experimental validation demonstrates that even under highly heterogeneous data distribution and a high power of compression (5\%), our method achieves strong performance and delivers improved results compared to the baselines.
>
> Regarding Weakness 3/Question 3, we kindly ask the reviewer to follow our next Official Comment, in which we provide a detailed response.

---

> ### Author Response · Authors · 2025-11-21
>
> 3. > Weights distribution analysis (W3/Q3)
>
> To investigate the sensitivity of client-specific aggregation weights to the level of model compression, we conduct an ablation study on ResNet-18 as the backbone architecture and the CIFAR-10 dataset, partitioned across 10 clients using a Dirichlet distribution with parameter $\alpha = 0.5$ to induce data heterogeneity.
>
> Three compression levels $-$ Rand 50%, Rand 10%, and Rand 5% - are evaluated. For each setting, we report the learned client aggregation weights (mean ± standard deviation over three independent runs) in table below.
>
> |  | Rand 50% | Rand 10% | Rand 5% |
> |-|-|-|-|
> | Client no. | Weight | Weight | Weight |
> | 1  | 0.019 $\pm$ 0.005 | 0.019$\pm$0.006 | 0.014$\pm$0.005 |
> | 2  | 0.026 $\pm$ 0.007 | 0.025$\pm$0.007 | 0.027$\pm$0.009 |
> | 3  | 0.141 $\pm$ 0.007 | 0.139$\pm$0.008 | 0.139$\pm$0.007 |
> | 4  | 0.248 $\pm$ 0.011 | 0.244$\pm$0.012 | 0.247$\pm$0.014 |
> | 5  | 0.136 $\pm$ 0.007 | 0.137$\pm$0.007 | 0.134$\pm$0.007 |
> | 6  | 0.054 $\pm$ 0.008 | 0.055$\pm$0.007 | 0.057$\pm$0.011 |
> | 7  | 0.145 $\pm$ 0.011 | 0.143$\pm$0.016 | 0.144$\pm$0.013 |
> | 8  | 0.047 $\pm$ 0.008 | 0.062$\pm$0.013 | 0.056$\pm$0.014 |
> | 9  | 0.043 $\pm$ 0.009 | 0.052$\pm$0.008 | 0.048$\pm$0.011 |
> | 10 | 0.147 $\pm$ 0.010 | 0.124$\pm$0.013 | 0.134$\pm$0.011 |
>
> These results are also presented in the updated version of the paper in Appendix B3 (Table 2). Below, we briefly discuss them.
>
>
> a) For the majority of clients (e.g., Clients 3, 4, 5, 7), the assigned aggregation weights remain remarkably stable across compression regimes, with variations typically within the margin of statistical uncertainty. This suggests convergence toward a *data-informed equilibrium*—consistent with theoretical expectations that optimal client weights reflect local data representativeness and utility, rather than being artifacts of compression-induced noise.
>
> b) Notably, Clients 8 and 9 exhibit non-monotonic weight adjustments under aggressive compression with Rand 5%, diverging from their trends at 50% and 10% compression. Post-hoc data inspection reveals that these clients possess highly skewed local distributions: each holds samples from only three classes, with two dominant classes constituting over 93% of their local datasets. Under severe sparsification, the reduced model capacity likely amplifies the impact of such distributional bias, leading the weight adaptation mechanism to dynamically re-calibrate contribution levels $-$ potentially to mitigate negative transfer or overfitting on minority classes.
>
> These findings underscore that while global aggregation weights are generally robust to moderate compression, *extreme compression intensifies sensitivity to local data pathology*, highlighting the interplay between model compression, client heterogeneity, and adaptive weighting strategies in federated optimization.
>
> Thanks to the reviewer for the questions. We are open to continue a discussion and hope that, in light of both the strengths you have already noted and the improvements we introduced, a reconsideration of the score may be possible.
>
> ---
>
> **References**
>
> [1] Cong D Dang and Guanghui Lan. On the convergence properties of non-Euclidean extragradient methods for variational inequalities with generalized monotone operators. **Computational
> Optimization and Applications** 2015
>
> [2] Alfredo N Iusem et al. Extragradient method with variance reduction for stochastic variational inequalities. **SIAM** 2017
>
> [3] Panayotis Mertikopoulos et al. Optimistic mirror descent in saddle-point problems: Going
> the extra(-gradient) mile. **ICLR** 2019
>
> [4] Yu-Guan Hsieh et al. Explore aggressively, update conservatively: Stochastic extragradient methods with variable stepsize scaling. **NeurIPS** 2020
>
> [5] Aleksandr Beznosikov et al. Distributed methods with compressed communication for solving variational inequalities, with theoretical guarantees. **NeurIps** 2022

---

> ### Author Response · Authors · 2025-11-27
>
> Dear Reviewer P4ke,
>
> We would like to gently follow up on our earlier rebuttal. We understand the considerable workload during the review process and sincerely appreciate the time you devote to evaluating submissions.
>
> We would also be grateful if you could take a moment to look at the additional elements included in our updated response $-$ namely, the new experimental results and our expanded theoretical analysis in the non-convex setting. Any feedback or comments you might have on these points would be greatly appreciated.
>
> Thank you very much for your time and consideration.
>
> Best regards,
> Authors

---

### Author Response · Authors · 2025-12-03

Dear Area Chair,

We would like to provide a brief summary of the review process, in case it may be helpful to their.

The reviews allowed us not only to identify the weaknesses of the paper but also to address them. All reviewers pointed out the gap between the practical applicability and the theoretical analysis, which stemmed from the convexity assumptions in our study. In the revised version, we incorporated a non-convex analysis in Appendix G. Following the reviewers’ advice, we also improved the clarity of presentation - for example, by breaking the main theorem into lemmas, adding a side-by-side complexity comparison, and providing a detailed description of the experimental setup.

The discussions with all reviewers also touched extensively on the experimental part of the work. Reflecting their feedback, we substantially expanded the experimental section by:
(i) adding a large-scale problem - CIFAR-100 (Appendix C.4),
(ii) extending the ablation study on data partitioning (Appendix C.2),
(iii) adding an analysis of the weight distribution (Appendix C.3),
(iv) expanding the list of baselines and experimentally evaluating the partial client participation setup (Appendix C.5).

The discussions with all reviewers could not be fully completed due to circumstances, but we provided answers to all their questions. The improvements already incorporated and evaluated by reviewer KP2w led to an increase in their score from 4 to 6; we note that not all improvements had been implemented at that point.

This year’s review process has been challenging for all parties. We truly appreciate the Area Chair’s efforts and the additional workload they have been required to take on.

Kind Regards,

The Authors

---

### Meta-Review · Area_Chair_gp1u · 2026-01-02

**Summary:**

This paper proposes ADI (Agnostic DIANA), a federated learning algorithm that integrates agnostic weighting (to address data heterogeneity) with compressed communication (to mitigate bandwidth limitations). The algorithm is formulated as a min-max optimization problem and leverages extragradient-style updates. The authors provide convergence guarantees under convex assumptions and extend the analysis to non-convex settings in the rebuttal. Empirical validation is primarily conducted on CIFAR-10 and CIFAR-100 under varying compression and participation settings.

While the reviewers acknowledged the relevance, motivation, and technical soundness of the paper, several key concerns were raised:

***Practical relevance of theoretical assumptions***: The convergence theory is grounded in convex analysis, which may not reflect the non-convex nature of most deep learning tasks in federated learning.\
***Experimental limitations***: The original version evaluated only on CIFAR-10, using limited baselines, without strong empirical validation under diverse settings or client participation regimes.\
***Lack of detailed analysis of weighting-compression interaction***: The mechanism by which compression affects the adaptive weighting dynamics was insufficiently analyzed.
***Ambiguous notation and complexity analysis***: The initial presentation of constants and scalings in the theoretical bounds was unclear.\
***Limited novelty positioning***: While the integration of weighting and compression is novel, the combination of known components (extragradient, DIANA-style compression, agnostic weighting) made it difficult to assess the distinct contribution.\
***Reproducibility***: Some reviewers noted a lack of hyperparameter and training schedule details, and no public code was initially provided.

**Final Recommendation: Accept (Poster)**\
While the initial version had several weaknesses, the authors responded with substantial improvements in theory, experiments, clarity, and reproducibility. The discussion with reviewers was constructive, and critical issues were resolved. The paper presents a well-motivated, technically sound framework that integrates weighting and compression in federated learning, with both theoretical grounding and practical relevance.

**Reviewer Concerns:**

**Addressed Concerns**\
The authors provided an extensive and thoughtful rebuttal, incorporating many reviewer suggestions directly into the revised manuscript:

***Non-convex analysis***: A new Minty-type convergence analysis was introduced in Appendix E, directly addressing the theoretical concern. The authors also offered references and justification for its applicability.\
***Expanded experiments***: CIFAR-100 and additional Dirichlet partitioning regimes were added, along with new baselines (e.g., MASHA1, AFL, q-FFL). Performance under various compression levels and partial participation was reported.\
***Analysis of compression–weighting interaction***: An ablation study on how learned client weights vary under different compression levels was included in the revised appendix.\
***Clarification of complexity expressions***: The authors clarified the roles of key constants and introduced lemmas to cleanly decompose the theoretical analysis.\
***Hyperparameter specification***: Detailed experimental settings, learning rate schedules, and optimizer configurations were added to Appendix A.

**Remaining Concerns**\
While the authors addressed most technical concerns, a few issues remain partially open:

***Comparative complexity positioning***: Although the authors now clarify why direct comparison with DIANA/MASHA is not appropriate due to different objectives, they initially omitted a side-by-side complexity table. A table was later added, but comparative positioning in the broader literature could still be strengthened.\
***No public code at the time of review***: While the experimental details are now sufficient for reproduction, public code would improve accessibility and confidence.\
***Minor novelty concerns***: The method is largely a careful integration of existing techniques rather than a fundamentally new algorithmic contribution.

**Reviewer Scores:**

**Reviewer P4ke (Initial Score: 4)**: This reviewer raised concerns about the convexity assumption, narrow experiments, and unclear interaction between compression and weighting. The rebuttal directly addressed all points, especially with non-convex theory (Minty), new experiments on CIFAR-100, and weight dynamics analysis. Likely updated score: 6.

**Reviewer KP2w (Initial Score: 4)**: Offered detailed technical critique, especially on theory clarity, complexity, and missing baselines. The author response included new lemmas, better constants exposition, added MASHA1/AFL/q-FFL baselines, and expanded experiments. Likely updated score: 6.

**Reviewer tXwb (Initial Score: 6)**: Raised concerns about the convexity gap, empirical scope, and weight interpretation. The authors expanded theory and experiments and corrected the weight narrative. Likely to maintain score at 6.

---

### Decision · Program_Chairs · 2026-01-26

Accept (Poster)